# STARD3 regulates lysosome positioning and contacts via a GSK3-controlled phosphorylation switch

Julie Eichler[1,2,3,4], Corinne Wendling[1,2,3,4], Sophie Huver[1,2,3,4], Mehdi Zouiouich [1,2,3,4], Victor Hanss [1,2,3,4], Anna Cardinal[5], Victoria Fimbel[1,2,3,4], Catherine Birck [1,2,3,4], Alastair G McEwen [1,2,3,4], Céline Knorr[1,2,3,4], Catherine Fromental-Ramain[1,2,3,4], Maxime Boutry[1,2,3,4], Marie-Pierre Chenard[1,2,3,4], Guillaume Drin[5], Catherine Tomasetto [1,2,3,4]✉ & Fabien Alpy [1,2,3,4]✉

## Abstract

Membrane contact sites (MCS) are dynamic regions where the membranes of two organelles come into close apposition. MCSs play many roles in cellular homeostasis by facilitating inter-organelle lipid exchange and organelle positioning. The late endosome/lysosome (LE/Lys) cholesterol transfer protein STARD3 forms reversible contacts between LE/Lys and the endoplasmic reticulum (ER). This tether protein contains a Phospho-FFAT motif (two phenylalanines in an acidic tract) whose interaction with ER-resident VAPs (vesicle-associated membrane protein-associated proteins) is phosphorylation-dependent. In this study, we identify GSK3α and GSK3β as the kinases responsible for phosphorylating serine 209 within the Phospho-FFAT motif of STARD3. This phosphorylation event is both necessary and sufficient to activate STARD3's tethering activity, thereby promoting ER-LE/Lys contacts. Furthermore, we show that when ER-LE/Lys tethering is prevented, STARD3 triggers LE/Lys homotypic interactions, revealing an additional function for STARD3 on endosome biology. Our findings establish a direct and critical role for GSK3 in regulating MCS via STARD3 phosphorylation, and expand our understanding of the molecular basis of inter-organelle communication.

**Keywords** Membrane Contact Site; Endoplasmic Reticulum; Endosome; Lipid Transfer Protein; Phosphorylation
**Subject Categories** Membranes & Trafficking; Organelles

## Introduction

In eukaryotic cells, organelles are physically connected to each other via dynamic structures known as membrane contact sites (MCS). MCSs are characterized by the close apposition, without fusion, of the membranes of two organelles at a distance around 10-30 nm (Levine and Loewen, 2006; Scorrano et al, 2019). This proximity enables the non-vesicular exchange of lipids and ions, and also contributes to the regulation of organelle positioning (Gatta and Levine, 2017; Prinz et al, 2020). The formation of MCSs is mediated by tether proteins that bring membranes close to each other via protein-protein and/or protein-membrane interactions (Levine and Loewen, 2006; Scorrano et al, 2019).

The endoplasmic reticulum (ER), a branched organelle extending throughout the cytosol, forms many contacts with other organelles. On its surface, the ER presents three major receptors belonging to the Major Sperm protein (MSP) domain-containing family: vesicle-associated membrane protein-associated proteins (VAP)-A and B (Loewen et al, 2003), and motile sperm domain-containing protein 2 (MOSPD2) (Di Mattia et al, 2018). These proteins expose their MSP domain to the cytosol, allowing them to interact with proteins bearing an FFAT motif [two phenylalanines (FF) in an acidic tract (AT)] (Di Mattia et al, 2018; Loewen et al, 2003; Murphy and Levine, 2016). The consensus sequence of conventional FFAT motifs is: $E_1F_2F_3D_4A_5X_6E_7$ (Mikitova and Levine, 2012). Positions 2 and 4 of the motif are crucial for the association of FFAT motif-containing proteins with VAPs (Di Mattia et al, 2020a; Furuita et al, 2010; Kaiser et al, 2005; Mikitova and Levine, 2012). Specifically, the aromatic residue at position 2 of FFAT motifs inserts into a hydrophobic pocket within the MSP domain of VAPs, while the acidic residue at position 4 interacts with positively charged residues of this domain. However, our recent studies on VAP-binding proteins have uncovered the existence of unconventional FFAT motifs in which the 4th position is occupied by a serine or a threonine (Di Mattia et al, 2020a). Importantly, we have shown that phosphorylation of this serine/threonine is essential for the recognition of these unconventional FFAT motifs by the MSP domain. Consequently, these motifs were termed Phospho-FFAT (Di Mattia et al, 2020a). Through bioinformatic analyses, we and others have identified both conventional and Phospho-FFAT motifs throughout the human proteome (Di Mattia et al, 2020a; Mikitova and Levine, 2012; Murphy and Levine, 2016), and interestingly, an equivalent number of proteins (~ 200) are predicted to contain

[1]Institut de Génétique et de Biologie Moléculaire et Cellulaire (IGBMC), Illkirch, France. [2]Institut National de la Santé et de la Recherche Médicale (INSERM), U 1258, Illkirch, France. [3]Centre National de la Recherche Scientifique (CNRS), UMR 7104, Illkirch, France. [4]Université de Strasbourg, Illkirch, France. [5]Université Côte d'Azur, Centre National de la Recherche Scientifique, Institut de Pharmacologie Moléculaire et Cellulaire, Valbonne, France. ✉E-mail: Catherine-Laure.Tomasetto@igbmc.fr; Fabien.Alpy@inserm.fr

either a conventional or a Phospho-FFAT motif (Di Mattia et al, 2020a).

We and others have further established that phosphorylation within the Phospho-FFAT motif of proteins can trigger functions that depend on their interaction with VAPs (Di Mattia et al, 2020a; Ende et al, 2022; Guillén-Samander et al, 2021; James et al, 2024). Intriguingly, phosphorylation can exert additional regulatory effects on FFAT motifs. For instance, some conventional FFAT motifs bear a phosphorylatable residue at the 5$^{th}$ position, such as that in Acyl-coenzyme A-binding domain protein 5 (ACBD5); in this case, phosphorylation inhibits interaction with the MSP domain of VAP proteins (Kors et al, 2022). Phosphorylation also occurs outside the FFAT core sequence, within the so-called acidic tract, which increases the binding affinity of the MSP domain for the FFAT motif (Furuita et al, 2010; Kumagai et al, 2014; Milanini et al, 2022). These observations support the idea that a phosphorylation code governs the formation of MCSs by providing a temporary mark either activating, reinforcing or inhibiting the binding of FFAT-containing proteins to VAPs (Alpy et al, 2013; Di Mattia et al, 2020a). STARD3 (StAR related lipid transfer (START) domain containing protein 3) is a protein with an NH2-terminal transmembrane MENTAL domain that anchors the protein to late endosomes/lysosomes (LE/Lys) (Alpy et al, 2005, 2001), and a COOH-terminal START domain that transports cholesterol at ER-LE/Lys contacts (Wilhelm et al, 2017). Interestingly, the non-structured region between these two domains contains a Phospho-FFAT motif, with the sequence $Q_1F_2Y_3S_4P_5P_6E_7$, where serine 209 in the protein (corresponding to position 4 of the motif) can be phosphorylated. The phosphorylation on the central serine ($S_{209}$) is necessary and sufficient to activate the interaction between STARD3 and VAP proteins, thereby triggering the tethering of LE/Lys to the ER and the cholesterol transport activity of STARD3 (Di Mattia et al, 2020a). While phospho-proteomic analyses of STARD3 in various biological samples revealed that $S_{209}$ is phosphorylated (Di Mattia et al, 2020a; Hornbeck et al, 2015), the identity of the kinase responsible for this modification remained unknown.

LE/Lys are central trafficking hubs of the endosomal system, involved in sorting and degrading extracellular material and plasma-membrane bound molecules internalized by endocytosis, as well as intracellular components such as organelles via autophagy (Scott et al, 2014). These processes rely on highly dynamic membrane trafficking events, including the formation of tubular structures, fission, fusion, and the movement of vesicles/tubes between the cell center and periphery. Interestingly, the dynamics of the endocytic pathway are regulated in part by MCSs with other organelles, such as the ER (Boutry et al, 2023; Cabukusta and Neefjes, 2018; Di Mattia et al, 2020b). For example, anterograde transport of LE/Lys is controlled by the ER protein Protrudin that interacts with LE/Lys and allows the recruitment of the kinesin interacting protein FYCO1 (Raiborg et al, 2015), and retrograde transport by ORP1L, which adopts two distinct conformations, one allowing its interaction with VAPs in ER-LE/Lys contacts, and the other the recruitment of the molecular motor dynein (Rocha et al, 2009).

In this study, a combination of in vitro and in cellulo analyses demonstrates that Glycogen synthase kinase 3 (GSK3) directly phosphorylates the Phospho-FFAT motif of STARD3, thereby regulating the formation of ER-LE/Lys contacts. Furthermore, this study uncovers a previously unrecognized function of the START domain: by binding membranes, it promotes homotypic interactions between LE/Lys, thereby regulating their positioning.

# Results

## STARD3 phosphorylation on serine 209 depends on GSK3α and GSK3β

To identify the kinase responsible for STARD3 $S_{209}$ phosphorylation, we analyzed the primary sequence of the motif using the PhosphoSitePlus Kinase Prediction tool (Johnson et al, 2023), which suggested Glycogen Synthase Kinase 3α and β (GSK3α and GSK3β) as candidates. The GSK3 recognition site fits with the Phospho-FFAT sequence with the presence of consensus amino acids around $S_{209}$, including a proline ($P_{210}$), an acidic residue ($E_{212}$) and a serine ($S_{213}$) (Fig. 1A). We also noted that GSK3α and GSK3β are priming-dependent kinases recognizing substrates with a pre-existing phosphoserine present 4 residues downstream of their target serine (ter Haar et al, 2001). We also knew that in STARD3, $S_{213}$, as well as two other serines at position 217 and 221 downstream of $S_{209}$, can be phosphorylated (Di Mattia et al, 2020a; Hornbeck et al, 2015). Together, these analyses prompted us to determine whether GSK3 is responsible for the phosphorylation of STARD3 in cells.

STARD3 is a ubiquitous protein expressed at low basal levels, except in cancer cells with genomic alterations in the HER2 (Human Epidermal Growth Factor Receptor 2) locus that lead to high expression levels (Lodi et al, 2023; Tomasetto et al, 1995; Voilquin et al, 2019). To study the regulation of STARD3 phosphorylation, we selected two breast cancer cell lines, HCC1954 and MCF7, which naturally express high and low endogenous levels of STARD3, respectively. MCF7 cells served as a model for ectopic STARD3 expression. To test whether GSK3 can phosphorylate STARD3, we treated HCC1954 cells with the GSK3 inhibitor CHIR99021 and examined STARD3 phosphorylation (Fig. 1B). The efficacy of CHIR99021 was confirmed by assessing the phosphorylation of a known GSK3 target, the acetyltransferase Tip60 (Fig. EV1A) (Charvet et al, 2011). Under these conditions, we measured the level of STARD3 phosphorylation at $S_{209}$ (STARD3-p$S_{209}$) using a phospho-specific antibody (Di Mattia et al, 2020a). Compared to untreated control cells, CHIR99021-treated cells showed a dramatic reduction in STARD3-p$S_{209}$ levels, while the total amount of STARD3 remained unchanged (Fig. 1B). To test the effect of GSK3 inhibition on ectopically expressed STARD3, we stably expressed STARD3 in MCF7 cells, thereby generating the MCF7/STARD3 cell line (Fig. 1C). These cells were also treated with CHIR99021. We observed a significant decrease in p$S_{209}$ STARD3 levels, with no major reduction in total STARD3 (Fig. 1D). Based on dose-response and time-course experiments (Fig. EV1B,Ca,b), we chose to treat cells overnight at a concentration of 5 μM of CHIR99021 for subsequent experiments.

As mentioned above, GSK3 exists as two isoforms, encoded by distinct genes; we therefore wondered which isoform was responsible for STARD3 $S_{209}$ phosphorylation. To determine the relative contribution of each isoform, GSK3α and GSK3β were silenced, either individually or together using pools of siRNAs in HCC1954 (Figs. 1E and EV1D) and in MCF7/STARD3

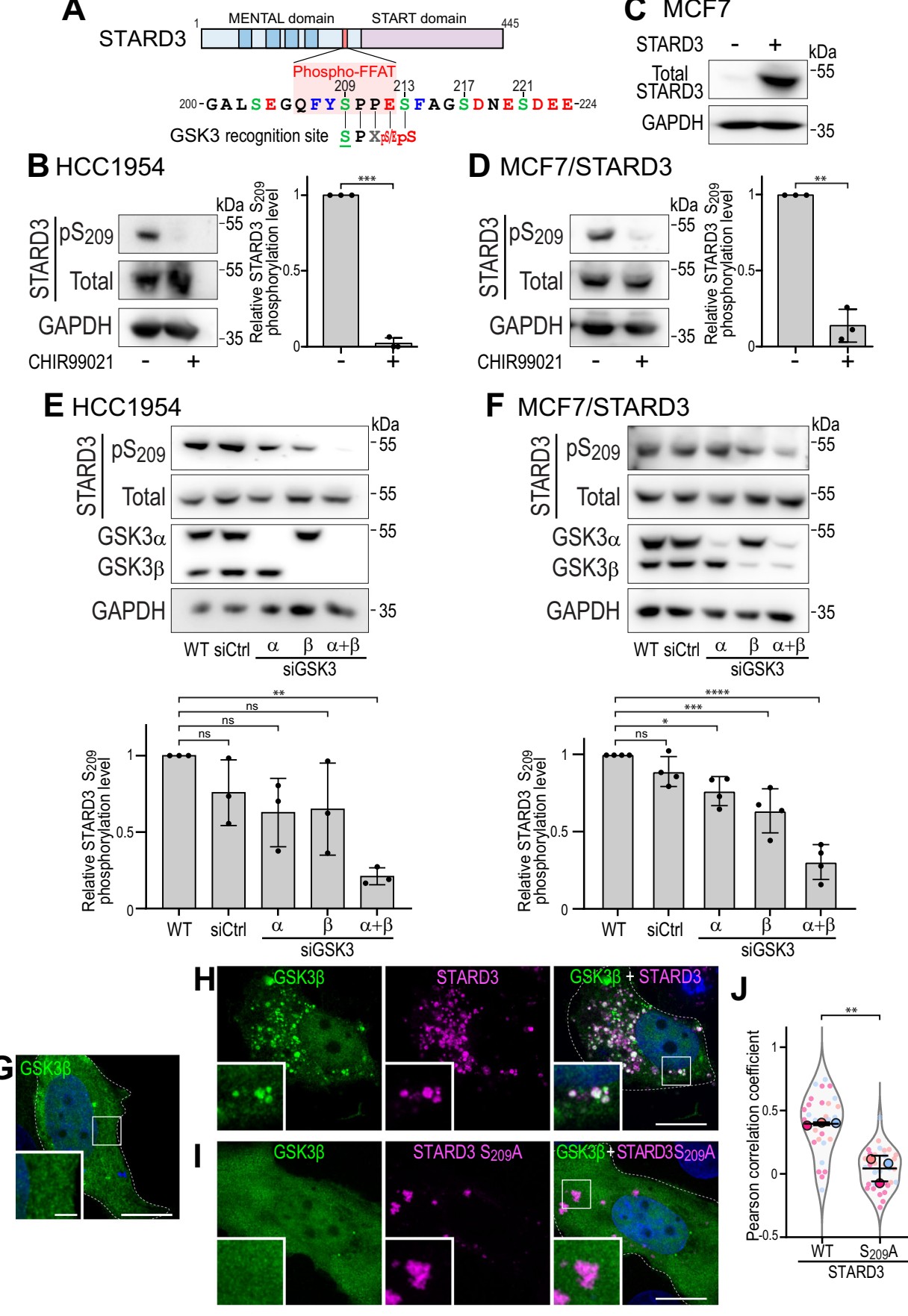

Figure 1. **STARD3 phosphorylation at $S_{209}$ requires GSK3α and GSK3β activity.**

(A) Schematic representation of STARD3. The MENTAL domain in blue presents four transmembrane helices (dark blue). The sequence of the Phospho-FFAT motif is highlighted in a red box. The four serine residues phosphorylated within and around the FFAT motif are numbered. The consensus sequence for the GSK3 recognition site is also shown. (B, D) Western blot analysis of HCC1954 (B) and MCF7/STARD3 (D) cells treated or not with the GSK3 inhibitor CHIR99021 (5 μM; overnight). STARD3 protein levels (Total) and $S_{209}$ phosphorylation ($pS_{209}$) were analyzed. Right: quantification of relative $S_{209}$ phosphorylation levels. Means ± SD. Student's $t$-test (***, $P < 0.001$; **, $P < 0.01$; $n = 3$ independent experiments; HCC1954: $P = 5 \times 10^{-4}$; MCF7/STARD3: $P = 5 \times 10^{-3}$). (C) Western blot analysis of control MCF7 cells (left) and MCF7 cells stably overexpressing STARD3 (right). (E,F) Western blot analysis of HCC1954 (E) and MCF7/STARD3 (F) cells transfected with control siRNAs (siCtrl) or siRNAs targeting GSK3α (siGSK3α), GSK3β (siGSK3β), or both (siGSK3α + siGSK3β). Bottom: quantification of relative $S_{209}$ phosphorylation levels. Means ± SD. One-way ANOVA with Dunnett's multiple comparison test (ns: not significant; *, $P < 0.05$; **, $P < 0.01$; ***, $P < 0.001$, $n = 3–4$ independent experiments: HCC1954: WT vs siCtrl $P = 0.4$; WT vs siGSK3α $P = 0.12$; WT vs siGSK3β $P = 0.15$; WT vs siGSK3α + siGSK3β $P = 1.9 \times 10^{-3}$; MCF7: WT vs siCtrl $P = 0.37$; WT vs siGSK3α $P = 1.5 \times 10^{-2}$; WT vs siGSK3β $P = 5 \times 10^{-4}$; WT vs siGSK3α + siGSK3β $P < 10^{-4}$). (G–I) HA-tagged GSK3β (green) was expressed alone (G) or together with WT (H) and $S_{209}A$ mutant (I) STARD3 (magenta) in MCF7 cells. GSK3β and STARD3 were labeled with anti-HA and anti-STARD3 antibodies, respectively. The subpanels are higher magnification images of the area outlined in white. The overlay panel shows merged images of green, magenta and blue (nuclei labeled with Hoechst) images. Scale bars: 10 μm. Inset scale bars: 2 μm. (J) Pearson's correlation coefficients between GSK3β-HA and STARD3 (WT or S209A) staining. Data are displayed as Superplots with Pearson's correlation coefficient for individual cells (small dots) and the mean per independent experiment (large dots). Number of cells: GSK3β-HA–STARD3: 29; GSK3β-HA–STARD3 S209A: 33, from three independent experiments. Independent experiments are color-coded. Mean values with error bars (SD) are shown. Unpaired t-test (**$P = 3.9 \times 10^{-3}$). Source data are available online for this figure.

(Figs. 1F and EV1E) cells. Silencing of either GSK3α or GSK3β individually did not markedly affect the level of STARD3 phosphorylation, while silencing both isoforms reduced STARD3 phosphorylation to a level comparable to that observed with pharmacological inhibition of GSK3 (Fig. 1E,F).

GSK3 shuttles between the cytosol and the nucleus, while STARD3 is exclusively found attached to LE/Lys with its Phospho-FFAT exposed in the cytosol (Alpy et al, 2013; Beurel et al, 2015). To determine whether the two proteins could occupy the same cellular location, allowing GSK3 to potentially directly phosphorylate STARD3, we performed immunofluorescence on ectopically expressed STARD3 and GSK3β in MCF7 cells. When expressed alone, GSK3β was localized diffusely in the cytosol and the nucleoplasm (Fig. 1G). However, when co-expressed with STARD3, in addition to this diffuse localization, GSK3β accumulated on punctiform structures corresponding to STARD3-positive LE/Lys (Fig. 1H), as shown by the co-localization of the two signals (Fig. 1J). Noteworthy, when a STARD3 mutant in which the potential GSK3 target serine residue was substituted with a non-phosphorylatable one (STARD3 $S_{209}A$) was co-expressed with GSK3β, GSK3β remained exclusively diffuse (Fig. 1I,J). These experiments show that GSK3β, present in the cytosol, could potentially phosphorylate STARD3, and be enriched on STARD3-positive LE/Lys, suggesting that a stable association might occur.

Together, these data show that GSK3 is implicated in STARD3 phosphorylation on $S_{209}$, and that GSK3α and GSK3β have a redundant activity.

## STARD3 is directly phosphorylated on serine 209 by GSK3

Since GSK3α and GSK3β are priming-dependent kinases (ter Haar et al, 2001), their identification as responsible for $S_{209}$ Phospho-FFAT phosphorylation suggested a mechanism in which phosphorylation of $S_{213}$ must occur to prime GSK3 to phosphorylate $S_{209}$ (Fig. 2A). To test this hypothesis, we mutated $S_{213}$, $S_{217}$ and $S_{221}$ into alanine, either individually or in combination. After expression of these mutants in MCF7 cells, we examined phosphorylation of $S_{209}$ by Western blot (Fig. 2B). As expected, the $S_{209}A$ mutant was not detected by the STARD3-$pS_{209}$-specific antibody (Fig. 2B,C). Interestingly, the $S_{213}A$ mutant was not detected either, indicating

that the absence of phosphorylation on $S_{213}$ prevents $S_{209}$ phosphorylation. On the other hand, mutation of $S_{217}$ and $S_{221}$ did not affect $S_{209}$ phosphorylation (Fig. 2B,C), suggesting that phosphorylation does not occur in cascade from $S_{221}$ to $S_{209}$, and that only $S_{213}$ phosphorylation is required for $S_{209}$ phosphorylation.

To fully assess the ability of GSK3 to directly phosphorylate STARD3, we performed in vitro kinase assays using purified recombinant STARD3 substrates, recombinant GSK3β, and ATP (Fig. 2D,E). For the STARD3 substrate, we produced the cytosolic part of STARD3, hereafter termed $_C$STD3, that includes the Phospho-FFAT motif and the START domain (Fig. 2D) (Wilhelm et al, 2017). To account for GSK3's dependence on priming, we generated $_C$STD3 WT but also two constitutively phosphorylated forms at either $S_{209}$ or $S_{213}$, using a genomically recoded E. coli strain engineered to allow phosphoserine incorporation into recombinant proteins (Park et al, 2011; Pirman et al, 2015). Because we do not have phosphospecific antibodies against $pS_{213}$, the presence of a phosphate on $S_{213}$ in $_C$STD3 $pS_{213}$ was verified by mass spectrometry (Appendix Fig. S1). As expected, in the absence of the kinase, constitutively phosphorylated $_C$STD3 $pS_{209}$ was efficiently detected with the phospho-$S_{209}$ specific antibody, while $_C$STD3 WT and $_C$STD3 $pS_{213}$ were not (Fig. 2F). In the presence of GSK3β, $_C$STD3 WT was not phosphorylated while the constitutively phosphorylated $c_C$STD3 $pS_{213}$ was phosphorylated on $S_{209}$ by GSK3β (Fig. 2F). This experiment shows that GSK3β directly phosphorylates STARD3 on $S_{209}$, but only when the protein is already phosphorylated on $S_{213}$. To test whether replacing $S_{213}$ by an acidic residue could lure GSK3β by mimicking phosphorylation, we produced and purified the $_C$STD3 $S_{213}E$ mutant; however, this mutant was not phosphorylated by GSK3β (Fig. 2F).

Altogether, these data show that STARD3 is directly phosphorylated on $S_{209}$ by GSK3 and that this event requires prior phosphorylation of $S_{213}$ by another kinase.

## GSK3 modulates STARD3 tethering activity in vivo

STARD3 is a tether protein that interacts with VAP-A, VAP-B and MOSPD2 to build contacts between the ER and LE/Lys (Alpy et al, 2013; Di Mattia et al, 2018). Phosphorylation of the Phospho-FFAT motif of STARD3 is necessary for the interaction with VAP proteins and MCS formation (Di Mattia et al, 2020a). We thus

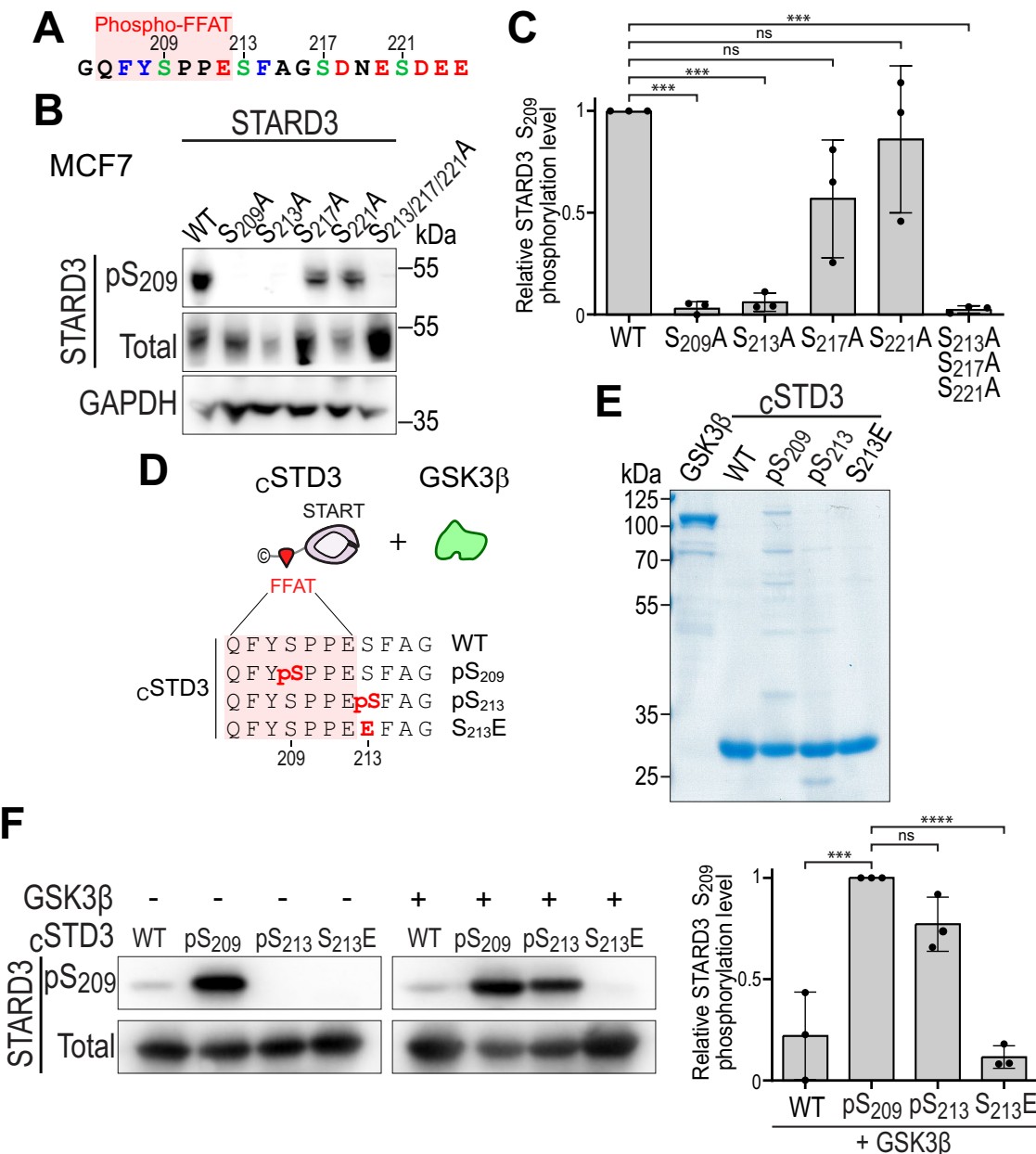

**Figure 2. GSK3 phosphorylates STARD3 S$_{209}$ in a priming-dependent manner.**

(A) Schematic representation of the Phospho-FFAT sequence of STARD3, highlighting the different phosphorylated serines. (B,C) Western blot analysis of MCF7 cells expressing WT and mutant (S$_{209}$A; S$_{213}$A; S$_{217}$A; S$_{221}$A; S$_{213}$A + S$_{217}$A + S$_{221}$A) STARD3. (C) quantification of relative S$_{209}$ phosphorylation levels. Means ± SD. One-way ANOVA with Dunnett's multiple comparison test (***, $P < 0.001$; $n = 3$ independent experiments; WT vs S$_{209}$A, $P = 2 \times 10^{-4}$; WT vs S$_{213}$A, $P = 3 \times 10^{-4}$; WT vs S$_{217}$A, $P = 0.06$; WT vs S$_{221}$A, $P = 0.84$; WT vs S$_{213}$A + S$_{217}$A + S$_{221}$A, $P = 2 \times 10^{-4}$). (D) Schematic representation of the recombinant proteins used in the in vitro kinase assay. (E) Coomassie blue staining of recombinant GSK3β, $_c$STD3 WT, $_c$STD3 pS$_{209}$, $_c$STD3 pS$_{213}$ and $_c$STD3 S$_{213}$E after SDS-PAGE. (F) Western blot analysis of in vitro kinase assay for $_c$STD3 WT, $_c$STD3 pS$_{209}$, $_c$STD3 pS$_{213}$, and $_c$STD3 S$_{213}$E incubated with recombinant GSK3β (right) or without it (left). Right: quantification of the relative S$_{209}$ phosphorylation levels. Means ± SD. One-way ANOVA with Dunnett's multiple comparison test (***, $P < 0.001$; ****, $P < 0.0001$; $n = 3$ independent experiments; $_c$STD3 WT vs $_c$STD3 pS$_{209}$, $P = 2 \times 10^{-4}$; $_c$STD3 pS$_{209}$ vs $_c$STD3 pS$_{213}$, $P = 0.15$; $_c$STD3 pS$_{209}$ vs $_c$STD3 pS$_{213}$E, $P < 10^{-4}$). Source data are available online for this figure.

reasoned that GSK3 inhibition should also prevent STARD3 from interacting with VAP proteins and the formation of ER-LE/Lys contacts.

First, we studied the interaction of STARD3 with VAPs under conditions of GSK3 inhibition by co-immunoprecipitation. For this, GFP-tagged VAP-A was co-expressed in MCF7 cells with wild-type STARD3, and GSK3 was inhibited with CHIR99021. Proteins were then immunoprecipitated using anti-GFP antibodies, and the presence of the STARD3-VAP-A complex was detected by Western blot (Fig. 3A). As previously described (Alpy et al, 2013; Di Mattia et al, 2020a), STARD3 was co-immunoprecipitated with VAP-A in untreated cells, while a mutant form of VAP-A (KD/

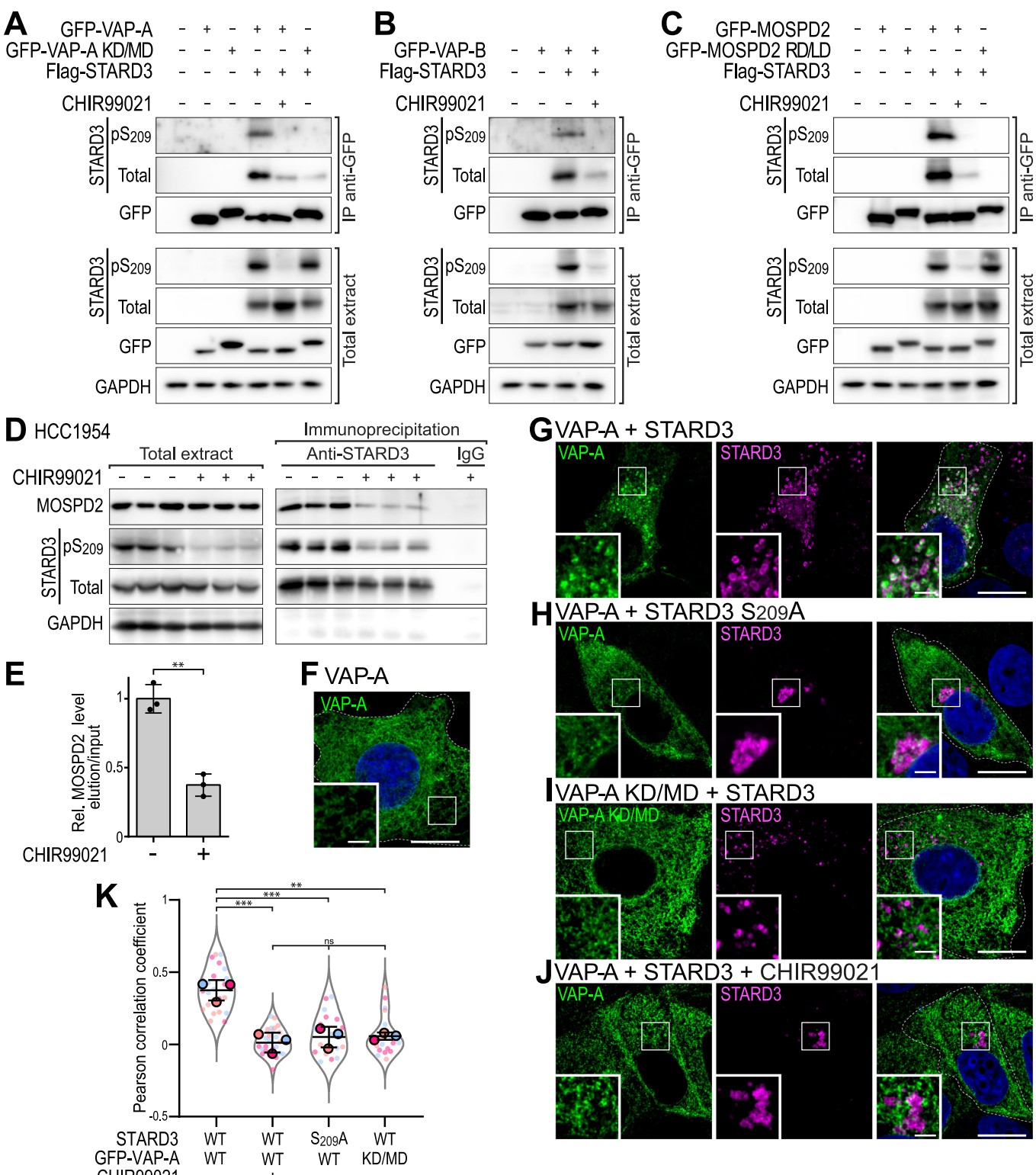

MD) unable to bind FFAT motifs showed only a faint residual STARD3 band, indicating that the binding was largely abolished. Likewise, after CHIR99021 treatment, only a weak STARD3 signal was detected (Fig. 3A), suggesting that GSK3 activity is required for the association of STARD3 with VAP-A. Similar experiments were then performed with VAP-B (Fig. 3B) and MOSPD2 (Fig. 3C) which showed that the binding of STARD3 to VAP-B and to MOSPD2 also requires GSK3 activity. Similar results were obtained in HeLa cells, showing that GSK3 activity is required across different cell types (Appendix Fig. S2A–C).

Figure 3. In vivo, GSK3 activity regulates the interaction between STARD3 and VAPs and the establishment of ER-endosome contacts.

(A–C) Immunoprecipitation (GFP-Trap) experiments between GFP-VAP-A (A), GFP-VAP-B (B) or GFP-MOSPD2 (C) and Flag-tagged STARD3 in MCF7 cells. Approximately 5 μg of total protein extract was analyzed by Western blot using anti-STARD3, anti-pS$_{209}$-STARD3, anti-GFP and anti-GAPDH antibodies. Immunoprecipitated proteins were analyzed using anti-STARD3, anti-pS$_{209}$-STARD3 and anti-GFP antibodies. (D) Immunoprecipitation experiment between endogenous STARD3 and MOSPD2 in HCC1954 cells. Cells were treated or not with CHIR99021, and proteins extracted. Immunoprecipitation was performed using control IgG or anti-STARD3 antibodies. Total protein extract and immunoprecipitated proteins were analyzed by Western blot using anti-STARD3, anti-pS$_{209}$-STARD3, anti-MOSPD2, and anti-GAPDH antibodies. (E) Quantification of MOSPD2 co-immunoprecipitated with STARD3 from the data shown in (D). The graph represents the ratio of MOSPD2 signal intensity in the eluted fraction relative to the input fraction. Means ± SD. Student's t-test (**, $P = 1.2 \times 10^{-3}$; $n = 3$ independent experiments). (F–J) MCF7 cells expressing GFP-VAP-A (F–H, J) and GFP-VAP-A KD/MD (I) (green) were either untransfected (F) or transfected with STARD3 WT (G, I, J) or STARD3 S$_{209}$A (H). Cells were left untreated (F, G–I) or treated with CHIR99021 (J). STARD3 was labeled using anti-STARD3 antibodies (G–J, magenta), and nuclei stained with Hoechst (blue). Insets show higher magnification images of the areas outlined in white. Scale bars: 10 μm. Inset scale bars: 2 μm. The overlay panels show merged green, magenta and blue images. (K) Pearson's correlation coefficients between VAP-A (WT or KD/MD mutant) and STARD3 (WT or STARD3 S$_{209}$A) in cells treated or not with CHIR99021. Data are displayed as Superplots with Pearson's correlation coefficient for individual cells (small dots) and the mean per independent experiment (large dots). Number of cells: VAP-A–STARD3: 24; VAP-A–STARD3 treated with CHIR99021: 28; VAP-A–STARD3 S$_{209}$A 19, VAP-A KD/MD–STARD3: 21, from three independent experiments. Independent experiments are color-coded. Means ± SD. ANOVA with Tukey's multiple comparison test ($P < 0.01$; ***, $P < 0.001$; VAP-A–STARD3 vs VAP-A–STARD3-CHIR99021, $P = 4 \times 10^{-4}$; VAP-A–STARD3 vs VAP-A–STARD3 S$_{209}$A, $P = 10^{-3}$; VAP-A–STARD3 vs VAP-A KD/MD–STARD3, $P = 1.1 \times 10^{-3}$; VAP-A–STARD3-CHIR99021 vs VAP-A–STARD3 S$_{209}$A, $P = 0.87$; VAP-A–STARD3-CHIR99021 vs VAP-A KD/MD–STARD3, $P = 0.82$; VAP-A–STARD3 S$_{209}$A vs VAP-A KD/MD–STARD3, $P = 0.99$). Source data are available online for this figure.

We then further characterized the interaction between STARD3 and VAP proteins at the endogenous level (Fig. 3D,E). To this end, HCC1954 cells were treated or not with CHIR99021, and proteins were extracted. Immunoprecipitation was performed using anti-STARD3 antibodies, followed by detection of endogenous STARD3 and MOSPD2. In untreated cells, MOSPD2 co-immunoprecipitated with STARD3. However, in CHIR99021-treated cells, where STARD3 phosphorylation was reduced, MOSPD2 was poorly immunoprecipitated with STARD3 (Fig. 3D,E). These results indicate that the interaction between endogenous STARD3 and MOSPD2 depends on STARD3 phosphorylation by GSK3.

Finally, we assessed the impact of GSK3 activity on MCSs built by STARD3 in cells. We previously observed that the formation of ER-LE/Lys contacts resulting from the interaction of STARD3 with VAPs induces the enrichment of VAPs in the ER subdomains in contact with STARD3-positive LE/Lys (Alpy et al, 2013; Di Mattia et al, 2020a, 2018; Wilhelm et al, 2017). We repeated this experiment under GSK3 inhibition conditions in MCF7 cells and observed that VAP-A was evenly distributed in the ER when expressed alone (Fig. 3F), while in the presence of STARD3, it accumulated around STARD3-positive LE/Lys (Fig. 3G,K). Preventing VAP-A-STARD3 interaction by mutagenesis using, on the one hand, a STARD3 mutant (S$_{209}$A) devoid of a functional FFAT motif (Fig. 3H), and, on the other hand, a VAP-A mutant (KD/MD) unable to bind FFAT motifs (Fig. 3I), prevented VAP-A accumulation in cells expressing STARD3 (Fig. 3K). We then investigated whether GSK3 inhibition produced a similar phenotype to that observed when VAP-A-STARD3 interaction was disrupted by mutagenesis. CHIR99021 treatment prevented the accumulation of VAP-A around STARD3-positive LE/Lys in cells expressing STARD3 (Fig. 3J). Pearson's correlation coefficient indicated that GSK3 inhibition led to a loss of colocalization between STARD3 and VAP-A, as seen with mutants unable to interact (Fig. 3K). Similar experiments performed with VAP-B (Appendix Fig. S3A–F) showed that the inhibition of GSK3 activity also impairs the association between STARD3 and VAP-B. To verify that inhibition of GSK3 on its own does not alter VAP protein localization, control cells expressing VAP-A only were treated with CHIR99021. GSK3 inhibition did not affect the even distribution of VAP-A in the ER in the absence of STARD3

overexpression (Appendix Fig. S3G,H). Similarly, in HeLa cells, the recruitment of VAP-A and VAP-B around STARD3-positive LE/Lys was dependent on GSK3 activity (Appendix Fig. S2D–P).

To directly assess the proximity between the ER and LE/Lys without relying on STARD3-VAP co-localization, we expressed fluorescent markers of the ER (mScarlet-ER) and LE/Lys (EGFP-TMEM192) (Fig. EV2). We first expressed STARD3 in cells co-expressing mScarlet-ER and EGFP-TMEM192, and evaluated ER-LE/Lys co-localization by measuring the Pearson correlation coefficient. Compared with control cells (WT MCF7 with or without CHIR99021 treatment), STARD3-expressing cells displayed increased ER-LE/Lys co-localization, as evidenced by a higher Pearson correlation coefficient (Fig. EV2A–C,E). This indicated that STARD3-dependent ER-LE/Lys contact formation is detectable with these fluorescent markers. When GSK3 was inhibited, the Pearson correlation coefficient decreased to basal levels in cells not expressing STARD3 (Fig. EV2D,E), indicating that the ER shifted away from LE/Lys under these conditions. Jointly, these findings confirm that GSK3 activity is required for the ER-LE/Lys tethering function of STARD3.

Together, these data show that in vivo, GSK3 activity is not compensated by other kinases and is required for the assembly of ER-LE/Lys contacts made by a complex between STARD3 and VAPs.

## STARD3 promotes the aggregation of late endosomes/lysosomes

While investigating the effect of CHIR99021 on STARD3 phosphorylation in HCC1954 cells, we unexpectedly observed a striking enrichment of LE/Lys to the perinuclear region (Fig. EV1Cc). Specifically, as STARD3 phosphorylation progressively decreased following CHIR99021 treatment, LE/Lys shifted from a dispersed cytosolic distribution to a concentrated perinuclear localization (Fig. EV1Ca-c). To further investigate this, HCC1954 cells were treated with CHIR99021, LE/Lys were labeled with anti-LAMP1 antibodies, and LE/Lys position analyzed (Fig. 4A). In control cells, LE/Lys were scattered throughout the cytosol. In contrast, CHIR99021-treated cells displayed a clear accumulation of LE/Lys in the perinuclear region (Fig. 4A). This

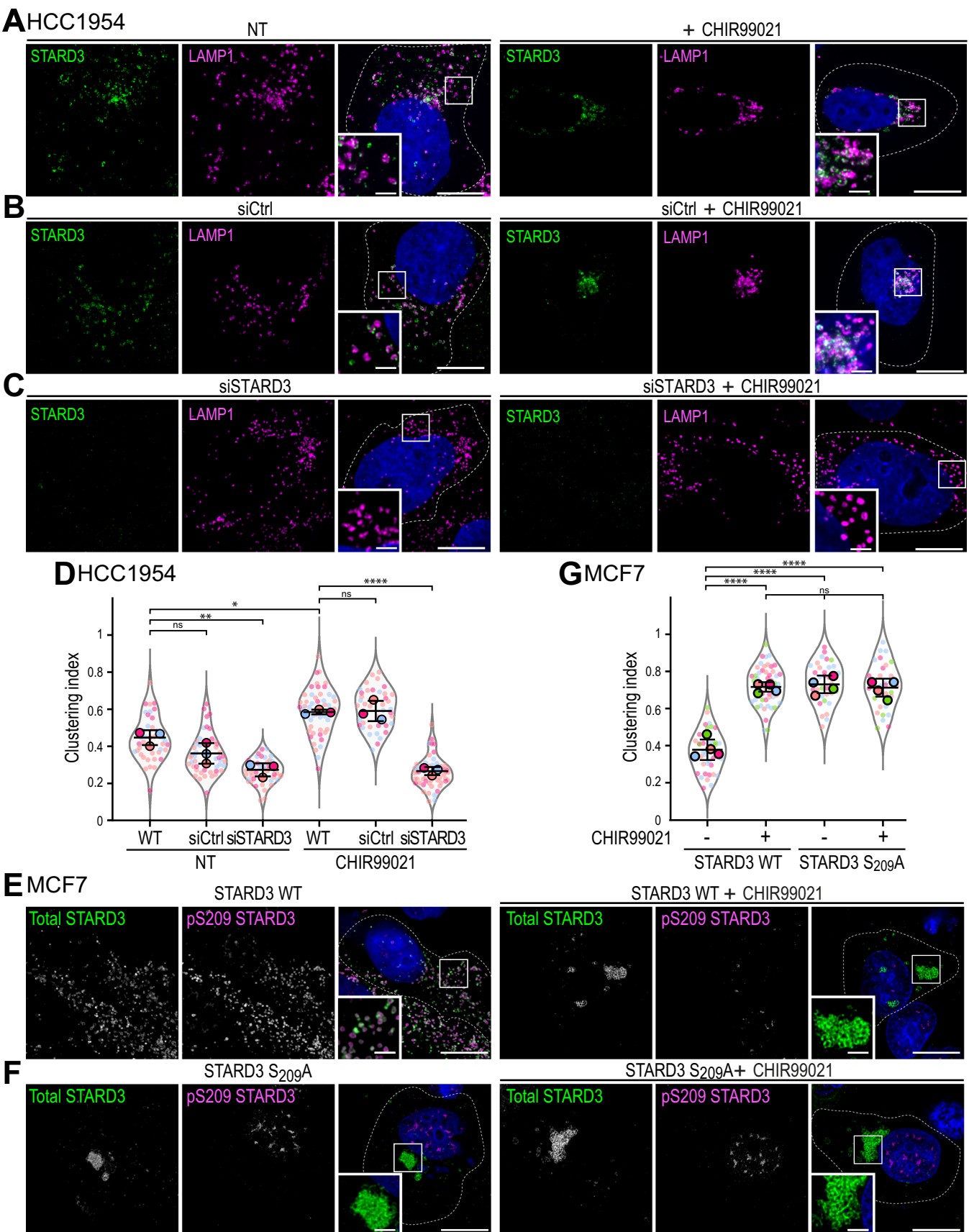

**Figure 4.  Inactivation of STARD3 phosphorylation induces clustering of LE/Lys.**

(A–C) HCC1954 cells either non-transfected (NT) (A) or transfected with control siRNAs (siCtrl) (B), or siRNAs targeting STARD3 (siSTARD3) (C) were treated or not with CHIR99021 (5 μM, overnight). STARD3 (green) and LAMP1 (magenta) were labeled with antibodies, and nuclei (blue) stained with Hoechst. The subpanels are higher magnification images of the area outlined in white. Scale bars: 10 μm. Inset scale bars: 2 μm. (D) Clustering index of LAMP1-positive vesicles of samples shown in (A–C). Data are displayed as Superplots showing the clustering index per cell (small dots) and its mean per independent experiment (large dots). Number of cells: WT-NT: 41, siCtrl-NT: 47, siSTARD3-NT: 45, WT-CHIR99021: 48, siCtrl-CHIR99021: 39, siSTARD3-CHIR99021: 51; from three independent experiments. Independent experiments are color-coded. Means and error bars (SD) are shown as black bars. One-way ANOVA with Tukey's multiple comparison test (*, $P < 0.05$; **, $P < 0.01$; ****, $P < 0.0001$; $n = 3$ independent experiments; WT-NT vs siCtrl-NT, $P = 0.17$; WT-NT vs siSTARD3-NT, $P = 1.9 \times 10^{-3}$; WT-NT vs WT-CHIR99021, $P = 1.1 \times 10^{-2}$; WT-CHIR99021 vs siCtrl-CHIR99021, $P = 0.99$; WT-CHIR99021 vs siSTARD3-CHIR99021, $P < 10^{-4}$). Scale bars: 10 μm. Inset scale bars: 2 μm. (E, F) MCF7 cells expressing STARD3 (C) and STARD3 $S_{209}$A (D) were left untreated (NT; left) or treated with CHIR99021 (right). Cells were labeled with anti-STARD3 antibodies (total STARD3; green) and phospho-specific antibodies (pS$_{209}$ STARD3; magenta). Nuclei were stained with Hoechst (blue). Subpanels show higher magnification images of the area outlined in white. Scale bars: 10 μm. Inset scale bars: 2 μm. (G) Clustering index of STARD3-positive vesicles of samples shown in (E, F). Data are displayed as Superplots showing the clustering index per cell (small dots) and its mean per independent experiment (large dots). Number of cells: STARD3-NT: 41, STARD3-CHIR99021: 52, STARD3 $S_{209}$A-NT: 33, STARD3 $S_{209}$A-CHIR99021: 32, from four independent experiments. Independent experiments are color-coded. Means and error bars (SD) are shown as black bars. One-way ANOVA with Tukey's multiple comparison test (****, $P < 0.0001$; $n = 4$ independent experiments; STARD3-NT vs STARD3-CHIR99021, $P < 10^{-4}$; STARD3-NT vs STARD3 $S_{209}$A-NT, $P < 10^{-4}$; STARD3-NT vs STARD3 $S_{209}$A-CHIR99021, $P < 10^{-4}$; STARD3-CHIR99021 vs STARD3 $S_{209}$A-NT, $P = 0.97$; STARD3-CHIR99021 vs STARD3 $S_{209}$A-CHIR99021, $P = 0.99$; STARD3 $S_{209}$A-NT vs STARD3 $S_{209}$A-CHIR99021, $P = 0.95$). Scale bars: 10 μm. Inset scale bars: 2 μm. Source data are available online for this figure.

effect was quantified using an unbiased automated image analysis (Appendix Fig. S4). Specifically, cells were imaged at high resolution and we defined a clustering index as the proportion of LE/Lys vesicles in direct contact with at least one other vesicle. This index was significantly higher in CHIR99021-treated cells compared to control cells, confirming increased LE/Lys clustering upon GSK3 inhibition (Fig. 4A,D). We next tested whether the effect of GSK3 on LE/Lys positioning was dependent on STARD3 by knocking-down its expression using siRNAs (Fig. 4C). Cells transfected with control siRNAs behaved similarly as WT cells (Fig. 4B,D). In contrast, STARD3 silencing prevented LE/Lys clustering following CHIR99021 treatment (Fig. 4C,D). These data suggest that GSK3 activity regulates LE/Lys positioning in a STARD3-dependent manner, and that unphosphorylated STARD3 favours LE/Lys clustering in the perinuclear region.

To further investigate the molecular mechanism of LE/Lys clustering, we turned to the overexpression model of STARD3 in MCF7 cells. First, we examined whether GSK3 inhibition induces LE/Lys clustering in the presence of STARD3 (Fig. 4E). In cells expressing STARD3 WT, STARD3-positive vesicles were dispersed throughout the cytosol and stained for both the pan-STARD3 antibody and the pS$_{209}$-specific antibody. Following CHIR99021 treatment, STARD3 staining became limited to one or a few large perinuclear structures that appeared as a cluster of vesicles, which was negative for the pS$_{209}$-specific antibody (Fig. 4E); consistently, the clustering index significantly increased (from ~0.4 to ~0.7) (Fig. 4G). To further characterize these vesicles and to determine whether other organelles were affected, we performed co-staining experiments with different organelle markers (Fig. EV3). MCF7/STARD3 cells were co-labeled with anti-STARD3 antibodies and late endosome/lysosome (LAMP1; Fig. EV3B), early endosome (EEA1; Fig. EV3C), Golgi (GM130; Fig. EV3D), and ER (Calnexin; Fig. EV3E) specific markers. Under condition of GSK3 inhibition, STARD3-positive vesicle clusters were positive for LE/Lys markers such as LAMP1 and negative for early endosomes, Golgi and ER markers (Fig. EV3B–E). While LE/Lys were clustered next to the nucleus, the positioning of early endosomes, Golgi and ER was not affected by GSK3 inhibition (Fig. EV3C–E). In control MCF7 cells (Fig. EV3A), GSK3 inhibition did not lead to any noticeable changes in the localization of LAMP1-positive endosomes, consistent with the notion that these cells have low levels of

STARD3 (Fig. 1C). To test whether this phenotype was common to other cell types, we conducted similar experiments in HeLa, U2OS, and COS-7 cells. We found that, in the presence of STARD3, CHIR99021 triggered the clustering of STARD3-positive vesicles in all these cell lines (Fig. EV4A–C). However, while in MCF7 cells GSK3 inhibition typically resulted in a unique cluster of endosomes, in the other cell types, LE/Lys aggregated into a few distinct perinuclear structures.

To determine whether the phosphorylation at S$_{209}$ of STARD3 controls this phenotype, we expressed a non-phosphorylatable STARD3 S$_{209}$A mutant. Consistent with this hypothesis, cells expressing STARD3 S$_{209}$A displayed clusters of STARD3-positive vesicles both in the presence and absence of CHIR99021 treatment (Fig. 4F). Accordingly, the clustering index of STARD3-positive vesicles remained high (~0.7) in STARD3 S$_{209}$A expressing cells, irrespective of the treatment (Fig. 4G).

Together, these data indicate that STARD3 promotes the perinuclear aggregation of LE/Lys vesicles when it is not phosphorylated by GSK3.

## Preventing STARD3 interaction with VAP proteins results in the clustering of late endosomes/lysosomes

Given that the formation of LE/Lys clusters occurs when VAP and STARD3 binding is prevented, we speculated that endosome positioning may be an additional function for this complex. To investigate this, we expressed in MCF7 cells two STARD3 mutants that are defective in VAP binding: one with a non-functional Phospho-FFAT motif (STARD3 F207A/Y208A, referred to as FA/YA) and one lacking the complete Phospho-FFAT motif (STARD3 ΔFFAT) (Alpy et al, 2013; Di Mattia et al, 2020a). Even in the presence of active GSK3 (without CHIR99021 treatment), expression of each mutant triggered the clustering of LE/Lys in the perinuclear area of MCF7 cells (Fig. EV3F,G). The same phenotype was observed following GSK3 inhibition. Thus, expression of a VAP-binding-deficient STARD3 mutant results in LE/Lys clustering.

Given that the Phospho-FFAT motif mediates binding to VAPs, we reasoned that silencing these proteins would trigger the clustering phenotype. To address this, we knocked-down VAP-A, VAP-B, and MOSPD2 individually or simultaneously in MCF7/

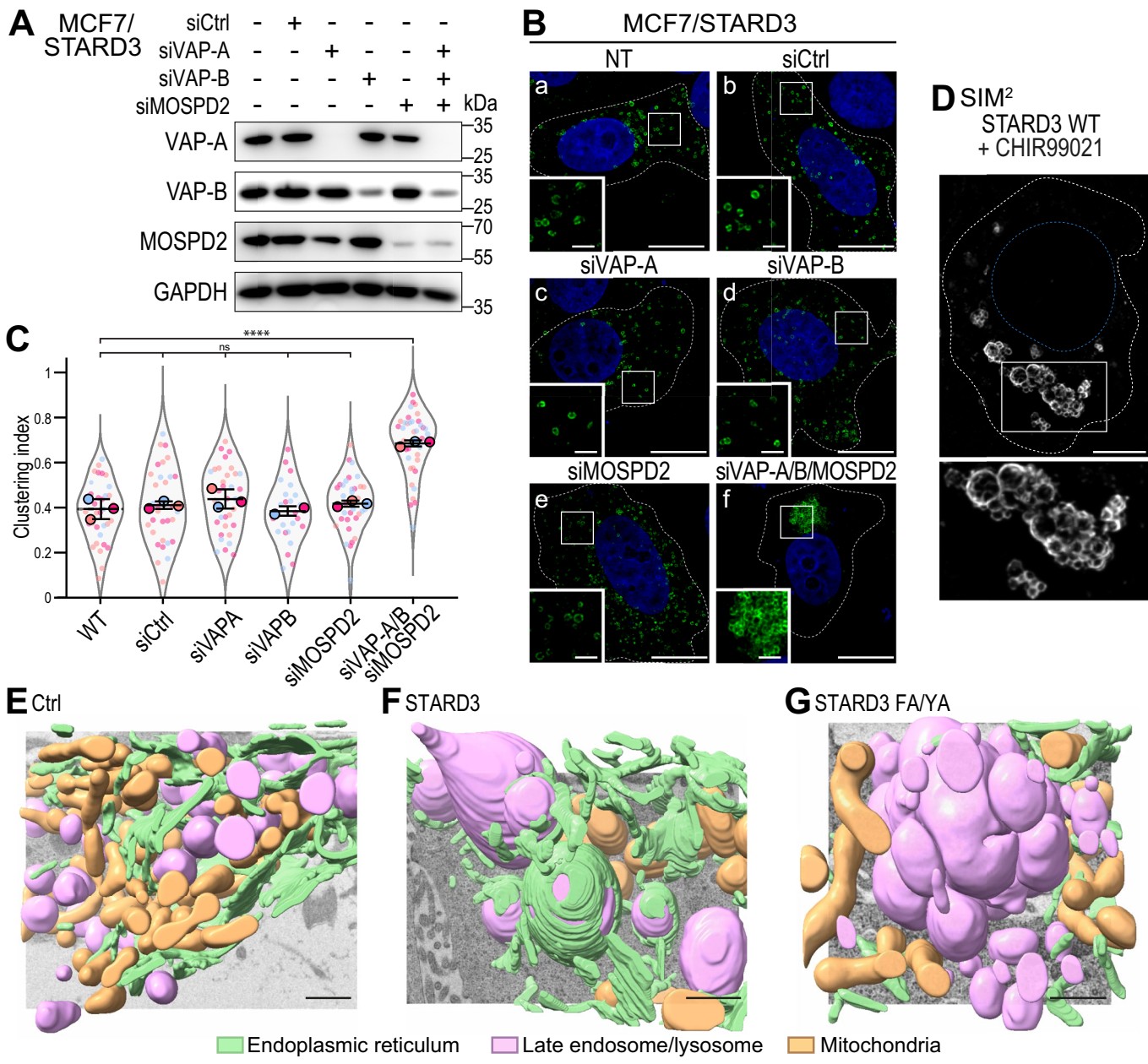

**Figure 5. ER-endosome contacts formed by the binding of STARD3 with VAP proteins are involved in LE/Lys positioning.**

(**A**) Western blot analysis of MCF7/STARD3 cells transfected with control siRNAs (siCtrl), siRNAs targeting VAP-A (siVAP-A), VAP-B (siVAP-B), MOSPD2 (siMOSPD2), and the three together. (**B**) MCF7/STARD3 cells either non-transfected (a) or transfected with control siRNAs (b), or siRNAs targeting VAP-A (c), VAP-B (d), MOSPD2 (e) or the three together (f). STARD3 (green) was labeled using anti-STARD3 antibodies, and nuclei (blue) stained with Hoechst. The subpanels are higher magnification images of the area outlined in white. Scale bars: 10 μm. Inset scale bars: 2 μm. (**C**) Clustering index of STARD3-positive vesicles of samples shown in (**B**). Data are displayed as Superplots showing the clustering index per cell (small dots) and its mean per independent experiment (large dots). Number of cells: NT: 35, siCtrl: 33, siVAP-A: 35, siVAP-B: 25, siMOSPD2: 36, siVAP-A/B/MOSPD2: 39, from three independent experiments. Independent experiments are color-coded. Means and error bars (SD) are shown as black bars. One-way ANOVA with Tukey's multiple comparison test (****, $P < 0.0001$; $n = 3$ independent experiments; WT vs siCtrl, $P = 0.91$; WT vs siVAPA, $P = 0.29$; WT vs siVAPB, $P = 0.99$; WT vs siMOSPD2, $P = 0.76$; WT vs siVAP-A/B/MOSPD2, $P < 10^{-4}$). (**D**) Super-resolution imaging (SIM²) of an MCF7 cell expressing STARD3 and treated with CHIR99021. The subpanel on the bottom shows a higher magnification (2×) image of the area outlined in white. Scale bar: 5 μm. (**E–G**) 3D rendering of LE/Lys (magenta), ER (green), mitochondria (brown) in control HeLa cells (**E**), cells expressing WT STARD3 (**F**), and cells expressing the STARD3 FA/YA mutant (**G**), imaged by FIB-SEM. Scale bar: 1 μm. Source data are available online for this figure.

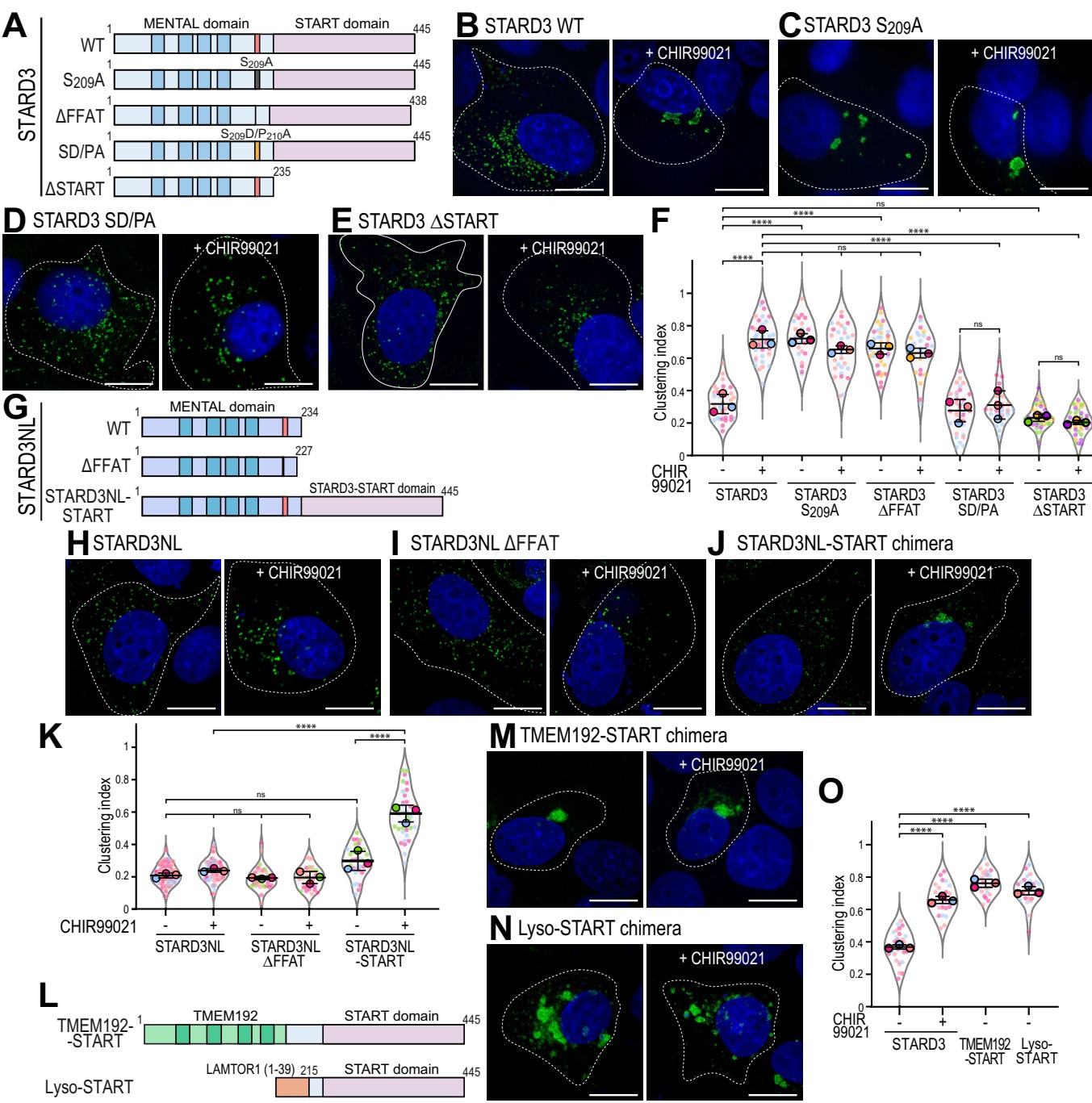

STARD3 cells (Fig. 5A). In control cells, STARD3-positive vesicles were dispersed throughout the cytosol (clustering index ~ 0.4) (Fig. 5Ba,b,C). Similar results were obtained with cells in which VAP-A, VAP-B, or MOSPD2 were individually knocked-down (Fig. 5Bc–e,C). In contrast, when all three VAPs were knocked-down simultaneously, STARD3-positive endosomes became clustered, as shown by a significant increase in the clustering index ( ~ 0.7; Fig. 5Bf,C). Thus, in the absence of VAPs, STARD3 induces the clustering of endosomes similar to what was observed in MCF7/STARD3 cells treated with CHIR99021 (Fig. 4E). These results support the idea that LE/Lys positioning in cells expressing

STARD3 relies on ER-LE/Lys contacts made by STARD3 and VAPs.

Next, we observed the LE/Lys clusters using super-resolution microscopy. Indeed, owing to the limited resolution of conventional light microscopy, the question arose as to whether these structures resulted from the clustering of individual LE/Lys and/or their subsequent fusion. To discriminate between the two possibilities, we performed super-resolution microscopy on cells expressing STARD3 and treated with CHIR99021. STARD3 was immunolabeled and imaged by structured illumination microscopy (SIM). We clearly observed that STARD3-positive clusters

◀

**Figure 6. The START domain of STARD3 mediates endosome clustering.**

(A) Schematic representation of the different STARD3 mutants used. (B–E) Representative images of MCF7 cells expressing WT STARD3 (B), STARD3 $S_{209}A$ (C), STARD3 SD/PA (D), STARD3 ΔSTART (E) left untreated (left) or treated with CHIR99021 (right), and labeled with an anti-STARD3 antibody (green) and with Hoechst (nuclei; blue). Scale bars: 10 μm. (F) Clustering index of STARD3-positive vesicles of samples shown in (B–E) and Fig. EV4H. Data are displayed as Superplots showing the clustering index per cell (small dots) and its mean per independent experiment (large dots). Number of cells: STARD3-NT: 37, STARD3-CHIR99021: 37, STARD3 $S_{209}A$-NT: 25, STARD3 $S_{209}A$-CHIR99021: 32, STARD3 ΔFFAT-NT: 40, STARD3 ΔFFAT-CHIR99021: 29, STARD3 SD/PA-NT: 35, STARD3 SD/PA-CHIR99021: 28, STARD3 ΔSTART-NT: 43, STARD3 ΔSTART-CHIR99021: 44, from five independent experiments. Independent experiments are color-coded. Means and error bars (SD) are shown as black bars. One-way ANOVA with Tukey's multiple comparison test (****, $P < 0.0001$; $n = 5$ independent experiments; STARD3-NT vs STARD3-CHIR99021, $P < 10^{-4}$; STARD3-NT vs STARD3 $S_{209}A$-NT, $P < 10^{-4}$; STARD3-NT vs STARD3 ΔFFAT-NT, $P < 10^{-4}$; STARD3-NT vs STARD3 SD/PA-NT, $P = 0.99$; STARD3-NT vs STARD3 ΔSTART-NT, $P = 0.51$; STARD3 SD/PA-NT vs STARD3 ΔSTART-NT, $P = 0.97$; STARD3-CHIR99021 vs STARD3 $S_{209}A$-NT, $P = 0.99$; STARD3-CHIR99021 vs STARD3 $S_{209}A$-CHIR99021, $P = 0.83$; STARD3-CHIR99021 vs STARD3 ΔFFAT-NT, $P = 0.90$; STARD3-CHIR99021 vs STARD3 ΔFFAT-CHIR99021, $P = 0.51$; STARD3-CHIR99021 vs STARD3 SD/PA-CHIR99021, $P < 10^{-4}$; STARD3-CHIR99021 vs STARD3 ΔSTART-CHIR99021, $P < 10^{-4}$). (G) Schematic representation of STARD3NL mutants used in the study. The MENTAL domain in light blue contains 4 transmembrane helices (dark blue) and a Phospho-FFAT motif (red). The chimeric construct STARD3NL-START domain is composed of STARD3NL fused to the START domain of STARD3. (H–J) Representative images of MCF7 cells expressing WT STARD3NL (H), STARD3NL ΔFFAT (I), and the STARD3NL-START chimeric construct (J) left untreated (left) or treated with CHIR99021 (right), and labeled with an anti-STARD3NL antibody (green) and with Hoechst (nuclei; blue). Scale bars: 10 μm. (K) Clustering index of STARD3NL-positive vesicles of samples shown in (H–J). Data are displayed as Superplots showing the clustering index per cell (small dots) and its mean per independent experiment (large dots). Number of cells: STARD3NL-NT: 78, STARD3NL-CHIR99021: 66, STARD3NL ΔFFAT-NT: 32, STARD3NL ΔFFAT-CHIR99021: 32, STARD3NL-START-NT: 33, STARD3NL-START-CHIR99021: 37, from four independent experiments). Independent experiments are color-coded. Means and error bars (SD) are shown as black bars. One-way ANOVA with Tukey's multiple comparison test (*, $P < 0.05$; ****, $P < 0.0001$; $n = 4$ independent experiments; STARD3NL-NT vs STARD3NL-CHIR99021, $P = 0.89$; STARD3NL-NT vs STARD3NL ΔFFAT-NT, $P = 0.99$; STARD3NL-NT vs STARD3NL ΔFFAT-CHIR99021, $P = 0.99$; STARD3NL-CHIR99021 vs STARD3NL ΔFFAT-NT, $P = 0.63$; STARD3NL-CHIR99021 vs STARD3NL ΔFFAT-CHIR99021, $P = 0.69$; STARD3NL ΔFFAT-NT vs STARD3NL ΔFFAT-CHIR99021, $P = 0.99$; STARD3NL-NT vs STARD3NL-START-NT, $P = 0.89$; STARD3NL-CHIR99021 vs STARD3NL-START-CHIR99021, $P < 10^{-4}$; STARD3NL-START-NT vs STARD3NL-START-CHIR99021, $P < 10^{-4}$). (L) Schematic representation of chimeric constructs: one consisting of the TMEM192 transmembrane fragment fused to the START domain of STARD3, and another combining the amino-terminal region of LAMTOR1 with the START domain of STARD3 (referred to as Lyso-START). (M, N) Representative images of MCF7 cells expressing TMEM192-START chimera (M), and Lyso-START (N), left untreated (left) or treated with CHIR99021 (right), and labeled with an anti-STARD3 antibody (green) and with Hoechst (nuclei; blue). Scale bars: 10 μm. (O) Clustering index of STARD3-positive vesicles of samples shown in (M, N). Data are displayed as Superplots showing the clustering index for individual cell (small dots) and the mean per independent experiment (large dots). Number of cells: STARD3-NT: 29, STARD3-CHIR99021: 24, TMEM192-START chimera-NT: 27, Lyso-START-NT: 30 from three independent experiments. Independent experiments are color-coded. Means and error bars (SD) are shown as black bars. One-way ANOVA with Dunett's multiple comparison test (****, $P < 0.0001$; $n = 3$ independent experiments; STARD3-NT vs STARD3-CHIR99021, $P < 10^{-4}$; STARD3-NT vs TMEM192-START, $P < 10^{-4}$; STARD3-NT vs Lyso-START, $P < 10^{-4}$). Source data are available online for this figure.

consisted of small vesicles that were aggregated (Fig. 5D). To complement these data, we performed focused-ion beam scanning electron microscopy (FIB-SEM) to visualize the three-dimensional organization of organelles (Fig. 5E–G and Movie EV1–3), as well as transmission electron microscopy (Appendix Fig. S5A–C) on control HeLa cells and on cells expressing either STARD3 WT or the FFAT-defective STARD3 FA/YA mutant. In agreement with the role of STARD3 in promoting ER-LE/Lys contacts (Alpy et al, 2013; Wilhelm et al, 2017), cells expressing STARD3 had increased ER in contact with endosomes compared to control cells (Fig. 5E,F, Movie EV1–2, and Appendix Fig. S5A,B). In contrast, cells expressing the FFAT-defective STARD3 FY/AA mutant displayed very few ER-endosome contacts and most endosomes were clustered and exhibited direct endosome-endosome contacts; notably, no endosome fusion was observed (Fig. 5G, Movie EV3, and Appendix Fig. S5C).

To conclude, in cells expressing STARD3, GSK3 inhibition results in the absence of phosphorylation of its FFAT motif, which impairs its tethering function and leads to the clustering without fusion of LE/Lys in the perinuclear region in all the different cells tested. No evidence of alteration of other organelles such as early endosome and Golgi was observed.

## The START domain of STARD3 is involved in the clustering of late endosomes/lysosomes

Next, we sought to understand the molecular mechanism by which STARD3 contributes to the positioning of LE/Lys by analyzing deletion and point mutants of the protein. WT and mutant STARD3 constructs, either lacking the START domain or carrying

mutations in the Phospho-FFAT motif (Fig. 6A), were expressed in MCF7 cells. Cells were treated or not with CHIR99021, STARD3 was imaged (Fig. 6B–E and EV4D–F), and the clustering index of STARD3-positive LE/Lys measured (Fig. 6F). Consistent with the data obtained before, expression of STARD3 WT resulted in the presence of STARD3-positive vesicles dispersed in the cytosol which clustered upon GSK3 inhibition (Figs. 6B and EV4D). As additional controls, we expressed STARD3 mutants with either a non-phosphorylatable (STARD3 $S_{209}A$; Figs. 6C,F and EV4E) or a deleted (STARD3 ΔFFAT; Fig. EV4F, Fig. 6F) FFAT motif. In these cells, STARD3-positive endosomes were clustered, regardless of GSK3 inhibition (Fig. 6F). To substantiate these data, we expressed the STARD3 $S_{209}D/P_{210}A$ mutant (hereafter named SD/PA) which carries phosphomimetic mutations rendering the FFAT motif always active and independent of phosphorylation (Di Mattia et al, 2020a) (Fig. 6D,F). Similar to cells expressing STARD3 WT, cells expressing the STARD3 SD/PA mutant displayed LE/Lys scattered throughout the cytosol; however, upon CHIR99021 treatment, LE/Lys remained dispersed.

As shown before, the phosphorylation of $S_{209}$ by GSK3 requires prior priming at $S_{213}$ (Fig. 2). To verify that the phenotype observed with the non-phosphorylatable STARD3 $S_{209}A$ mutant could be recapitulated when priming is impaired, we expressed the STARD3 $S_{213}A$ mutant in MCF7 cells. As control, we also expressed the STARD3 $S_{217}A$, $S_{221}A$, and $S_{213}A/S_{217}A/S_{221}A$ mutants and quantified the clustering index (Fig. EV4I,J). Consistent with the results presented in Fig. 2, LE/Lys were clustered in cells expressing the STARD3 $S_{213}A$ and STARD3 $S_{213}A/S_{217}A/S_{221}A$ mutants (clustering index: ~0.7) compared to the control condition. To ensure that the $S_{213}A$ mutation itself was not directly responsible

for LE/Lys clustering, we expressed the STARD3 SD/PA $S_{213}A$ mutant (Fig. EV4G). Consistent with this, LE/Lys remained dispersed in STARD3 SD/PA $S_{213}A$-expressing cells, confirming that the clustering observed with $S_{213}A$ alone was due to the loss of priming and not an independent effect of the mutation. Together, these data show that when the Phospho-FFAT motif of STARD3 cannot undergo phosphorylation, STARD3-positive LE/Lys cluster in the perinuclear region, and moreover, that a constitutively active STARD3 mutant does not induce clustering and remains unresponsive to GSK3 inhibition.

Having confirmed the involvement of STARD3's Phospho-FFAT motif in endosome positioning, we next investigated the role of the START domain. To this end, we imaged MCF7 cells expressing a mutant lacking the entire START domain (STARD3 ΔSTART) (Fig. 6A). In these cells, STARD3 ΔSTART-positive endosomes remained dispersed throughout the cytosol, regardless of CHIR99021 treatment (Fig. 6E,F). Similar results were obtained with the STARD3 $S_{209}A$ mutant lacking the START domain (STARD3 $S_{209}A$ ΔSTART) (Fig. EV4H). These findings demonstrate that the START domain is essential for the clustering of STARD3-positive endosomes.

STARD3 has a paralog named STARD3NL (STARD3 N-terminal like, previously named MENTHO) (Alpy et al, 2002), which encodes a protein containing a MENTAL domain and a Phospho-FFAT motif but lacking a START domain (Fig. 6G) (Alpy et al, 2005; Alpy and Tomasetto, 2006; Di Mattia et al, 2020a). Similarly to STARD3, STARD3NL mediates the formation of ER-LE contacts by interacting with VAP-A/VAP-B/MOSPD2 (Alpy et al, 2013; Di Mattia et al, 2018). To further investigate the role of the START domain in endosome clustering, we leveraged this paralog. First, we examined whether STARD3NL behaves similarly to the MENTAL domain of STARD3 with or without Phospho-FFAT motif. Wild-type STARD3NL (STARD3NL WT) and a mutant lacking its FFAT motif (STARD3NL ΔFFAT) were expressed. STARD3NL-positive vesicles remained dispersed in the cytosol, even after GSK3 inhibition (Fig. 6H,K). Consistently, expression of STARD3NL ΔFFAT had no detectable effect on the positioning of STARD3NL-positive endosomes under both untreated and CHIR99021-treated conditions (Fig. 6I,K), indicating that the MENTAL domain does not influence endosome positioning. To further explore the role of the START domain of STARD3, we generated a chimeric protein composed of STARD3NL fused to the START domain of STARD3 (Fig. 6G). Similarly to STARD3, this STARD3NL-START chimera localized to LE/Lys, where it colocalized with LAMP1 (Appendix Fig. S6A,B). Under basal conditions, cells expressing the chimera exhibited dispersed LE/Lysosomes, but LE/Lys clustering was observed when GSK3 was inhibited (Fig. 6J,K). These data further confirm that the START domain is responsible for LE/Lys clustering.

To definitively determine that the START domain of STARD3 is responsible for the clustering phenotype, independently of the rest of the protein and of GSK3/Phospho-FFAT regulation, we designed two synthetic proteins consisting of a LE/Lys anchoring sequence fused to the START domain (Fig. 6L). First, we created a chimeric protein composed of the four transmembrane helices of the LE/Lys-resident protein TMEM192, and the START domain of STARD3. This TMEM192-START chimera co-localized with LAMP1 (Appendix Fig. S6C) and induced LE/Lys clustering, confirming that the MENTAL domain is not involved in the clustering of LE/

Lys (Fig. 6M,O). Second, we designed a protein consisting of the 39-residue N-terminal region of p18/LAMTOR1 that is myristoylated and palmitoylated, and confers LE/Lys localization (Nada et al, 2009), fused to the START domain (Fig. 6L). This protein, referred to as Lyso-START, colocalized with LAMP1 to LE/Lys (Appendix Fig. S6D), and induced LE/Lys clustering in a GSK3-independent manner (Fig. 6N,O). These data show that the START domain, when artificially attached to LE/Lys, is sufficient to drive LE/Lys clustering.

Together, these findings demonstrate that the unphosphorylated form of STARD3 drives the clustering of STARD3-positive LE/Lys and that the START domain is the key driver of this clustering.

## The membrane-binding ability of the START domain of STARD3 promotes late endosomes/lysosomes clustering

To understand how the START domain promotes the clustering of STARD3-positive endosomes when the formation of ER-endosome contacts is prevented, we first hypothesized that the START domain dimerizes in trans to connect LE/Lys. To test this, we purified a 6His-tagged version of this domain (Fig. EV5A,B) and assessed its propensity to dimerize using analytical ultracentrifugation experiments (Fig. EV5C). We found that it exhibited a sedimentation coefficient ($S_{20,W} = 2.35$ S) that closely matches the theoretical value for a monomer in solution ($S_{theoretical} = 2.41$ S) (Fig. EV5B). This result indicates that the START domain of STARD3 behaves as a monomer, suggesting that trans-dimerization is not the mechanism involved in endosomal clustering.

Knowing that the START domain of STARD4 is able to bind negatively charged membranes (Iaea et al, 2015; Mesmin et al, 2011; Zhang et al, 2022), we then hypothesized that the START domain of STARD3 may likewise promote LE/Lys clustering by directly tethering LE/Lys which are known to be negatively-charged (Kobayashi et al, 2002). To test this hypothesis, we first examined whether the $_C$STD3 protein, which encompasses the START domain (Fig. EV5A,B), can bind liposomes using flotation assays (Fig. 7A). We incubated $_C$STD3 with neutrally-charged liposomes only composed of phosphatidylcholine (PC) or with negatively-charged liposomes composed of 70 mol% PC and 30 mol% PS, all doped with a small amount of fluorescent lipid (Fig. 7A). As a positive control, liposomes were doped with 3 mol% MPB-PE to enforce $_C$STD3 binding. MPB-PE contains a maleimide group that reacts covalently with thiols. Since $_C$STD3 possesses an N-terminal cysteine, it becomes covalently anchored to MPB-PE-containing liposomes, irrespective of any intrinsic membrane affinity, thereby mimicking the anchoring of the START domain of STARD3 at the surface of LE/Lys. Each sample was subjected to ultracentrifugation over a sucrose gradient; liposomes, tracked via their fluorescence, were recovered in the top fraction. SDS-PAGE analysis of the bottom and top fractions indicated that $_C$STD3 did not substantially associate with neutrally or negatively-charged liposomes (membrane-bound fraction < 20%) unless they contained MPB-PE ($\geq$ 50%). These data suggested that the isolated START domain cannot associate with membranes under these conditions.

Then, we reasoned that the interaction of the START domain with membranes might be too weak to be detected by flotation assays, but could be revealed under experimental conditions where the local surface density of STARD3 is higher. We therefore performed liposome pull-down assays (Fig. 7C) using magnetic

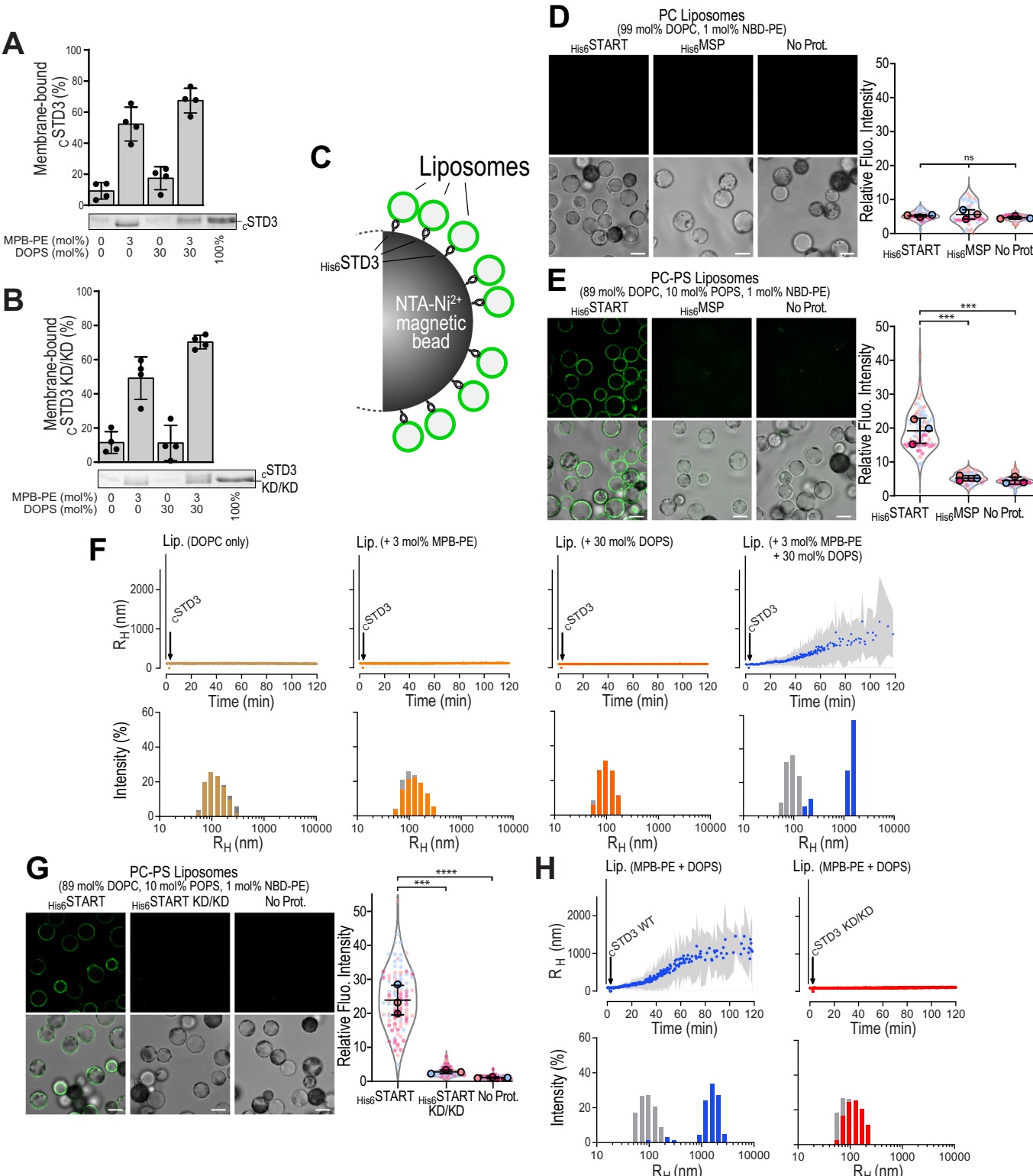

NTA-Ni²⁺ beads covered by a 6His-tagged version of the START domain (Fig. EV5B). Fluorescently labeled liposomes were then incubated with the beads and after washing away unbound liposomes, fluorescence was imaged and quantified (Fig. 7D,E).

The MSP domain of VAP-B was used as a negative control (Figs. EV5B and 7D,E). We incubated the proteins with liposomes only made of PC, and did not observe any detectable binding (Fig. 7D). In contrast, when PS-containing liposomes were used, a

**Figure 7. The START domain of STARD3 interacts with negatively charged liposomes.**

**(A,B)** Flotation assay. Liposomes (750 µM total lipids) composed of DOPC, DOPC/MPB-PE (97:3), DOPC/DOPS (70:30), or DOPC/DOPS/MPB-PE (67:30:3) and doped with 0.2% NBD-PA were mixed with DTT-free $_C$STD3 **(A)** or $_C$STD3 KD/KD **(B)** (0.75 µM) for 1 h in HK buffer at 25 °C. After centrifugation, the liposomes were collected at the top of sucrose cushions and analyzed by SDS-PAGE. The amount of membrane-bound protein was determined using the content of lane 5 (100% total) as a reference based on the SYPRO Orange intensity. Data are represented as mean ± s.e.m. ($n = 4$) with single data points. **(C)** Principle of liposome pull-down assays. Proteins were immobilized on magnetic NTA-Ni$^{2+}$ beads thanks to their 6His tag and incubated with fluorescent liposomes. After washing, the bound liposomes were imaged. **(D, E, G)** Representative confocal images of NTA-Ni$^{2+}$ beads either not bound to recombinant proteins (no Prot.) or bound to recombinant wild-type **(D, E, G)** or KD/KD mutant **(G)** START domain or MSP domain **(D, E)**. Beads were incubated with fluorescent neutral liposomes **(D)** or fluorescent negatively charged liposomes **(E, G)** (green). Top: Confocal sections showing liposome fluorescence; bottom: merged fluorescence and brightfield images of the beads. Spinning-disk confocal microscope (Nikon CSU-X1, 100× NA 1.4) images. Scale bars: 10 µm. Quantification of liposome recruitment on NTA-Ni$^{2+}$ beads. NBD fluorescence was measured around the beads. Data are displayed as Superplots, showing the relative fluorescence intensity per bead (small dots) and the mean per independent experiment (large dots). Number of beads: START WT **(D)** 72, **(E)** 75, **(G)** 104; MSP: **(D)** 47, **(E)** 58; START KD/KD: **(G)** 104; no Prot.: **(D)** 52, **(E)** 78, **(G)** 114; from three independent experiments). Independent experiments are color-coded. Means and error bars (SD) are represented by black bars. One-way ANOVA with Tukey's multiple comparison test (***, $P < 0.001$; ****, $P < 0.0001$; $n = 3$ independent experiments; **(D)**: START WT vs MSP, $P = 0.96$; START WT vs No Prot., $P = 0.71$; MSP vs No Prot., $P = 0.55$; **(E)**: START WT vs MSP, $P = 5 \times 10^{-4}$; START WT vs No Prot., $P = 4 \times 10^{-4}$; **(G)**: START WT vs MSP, $P = 10^{-4}$; START WT vs No Prot., $P < 10^{-4}$). **(F)** DLS experiments. DTT-free $_C$STD3 (0.5 µM) was mixed with liposomes (50 µM lipids) composed of DOPC, DOPC/MPB-PE (97:3), DOPC/DOPS (70:30), or DOPC/DOPS/MPB-PE (67:30:3) in HK buffer at 25 °C. The mean radius (dots) and polydispersity (shaded area) of the liposome suspension were measured for 2 h. Bottom panel: size distribution before (gray bars) and after the reaction (colored bars). The data are representative of 4 independent experiments. **(H)** DLS experiments. DTT-free $_C$STD3 or $_C$STD3 KD/KD (0.5 µM) was mixed with liposomes (50 µM lipids) composed of DOPC/DOPS/MPB-PE (67:30:3) in HK buffer at 25 °C. The mean radius (dots) and polydispersity (shaded area) of the liposome suspension were measured for 2 h. Bottom panel: size distribution before (gray bars) and after the reaction (colored bars). The data are representative of 4 independent experiments. Source data are available online for this figure.

strong fluorescence was observed in the presence of the START domain, but not with the control protein or in the absence of protein (Fig. 7E). These results suggest that the START domain of STARD3 can bind negatively charged membranes, when immobilized on a surface, possibly at a high local concentration.

We then examined whether the $_C$STD3 construct, covalently attached via its N-terminus to negatively charged liposomes made of PC, PS and MPB-PE, could induce liposome clustering, using dynamic light scattering (DLS, Fig. 7F). This was indeed the case: upon mixing the protein with liposomes, we measured a progressive increase in the hydrodynamic radius size of particles, from approximately 100 nm to around 1000 nm, along with an increase in polydispersity, indicating the formation of liposome clusters (Fig. 7F). As a control, we repeated this experiment with liposomes devoid of PS or, MPB-PE or both. In all cases, no clustering was seen. Thus, when bound to the surface of negatively charged liposomes, akin to LE/Lys, the START domain of STARD3 can promote their clustering.

The requirement for negatively charged lipids in the interaction between the START domain of STARD3 and membranes suggests that the binding is mediated by electrostatic interactions, likely involving positively charged residues. The START domain adopts a helix-grip fold, consisting of a central nine-stranded antiparallel β-sheet flanked by two α-helices (Fig. EV5D) (Tsujishita and Hurley, 2000). Previous research on STARD4 identified critical positive surface patches, especially around β1 and β3, with R46 residue (mouse STARD4) playing a key role (Iaea et al, 2015; Mesmin et al, 2011; Talandashti et al, 2024; Zhang et al, 2022). R46 in STARD4 corresponds to K260 in STARD3, representing a conservative substitution (Fig. EV5D). K260 is located on β1 near another positively charged residue, K281 on β3. To evaluate their role in the binding of negatively charged membranes, we generated a double mutant in which K260 and K281 were changed to aspartic acid ($K_{260}D$ $K_{281}D$, hereafter referred to as KD/KD). Recombinant KD/KD mutant versions of $_C$STD3 and $_{6His}$START domains were produced (Fig. EV5A,B). To verify that the mutations did not disrupt the protein's secondary structure, we performed circular dichroism spectroscopy (Fig. EV5E). The far-UV CD spectra of the

$_C$STD3 WT and KD/KD mutant were highly similar, indicating that the overall fold of the START domain was preserved. Next, we assessed membrane binding using liposome flotation assays (Fig. 7B), which showed that the KD/KD mutant behaved similarly to wild-type $_C$STD3 in this context. However, in liposome pull-down (Fig. 7G) and liposome aggregation assays (Fig. 7H), the KD/KD mutant failed to bind negatively charged liposomes and induce aggregation, in contrast to the wild-type protein. These results demonstrate that the conserved positively charged residues K260 and K281 are critical for the interaction of the START domain with negatively charged membranes.

Finally, we examined whether interactions between the START domain and negatively charged membranes drives LE/Lys clustering by expressing the STARD3 KD/KD mutant in MCF7 cells (Fig. 8A). Cells were treated or not with CHIR99021, STARD3 was imaged, and the clustering index of STARD3-positive LE/Lys was measured (Fig. 8F). Wild-type STARD3 induced LE/Lys clustering after CHIR99021 treatment (Fig. 8B,F), while LE/Lys remained dispersed in STARD3 KD/KD-expressing cells regardless of treatment (Fig. 8C,F). To confirm these results, we expressed a STARD3 double mutant ($S_{209}A$ KD/KD), combining a non-phosphorylatable Phospho-FFAT ($S_{209}A$) mutation with KD/KD. Unlike STARD3 $S_{209}A$ alone, which clustered LE/Lys (Fig. 8D,F), STARD3 $S_{209}A$ KD/KD-expressing cells exhibited dispersed LE/Lys (Fig. 8E,F).

Together, these data show that the START domain of STARD3 binds negatively charged membranes, furthermore they suggest that this property mediates the homotypic clustering of LE/Lys when STARD3 is free from ER-LE/Lys contacts.

## Discussion

Organelles are not isolated entities; instead, they communicate with one another through membrane contact sites to maintain proper cellular function (Prinz et al, 2020; Scorrano et al, 2019). A crucial mechanism in the formation of MCSs involves the interaction between proteins located on different organelles. The ER-resident

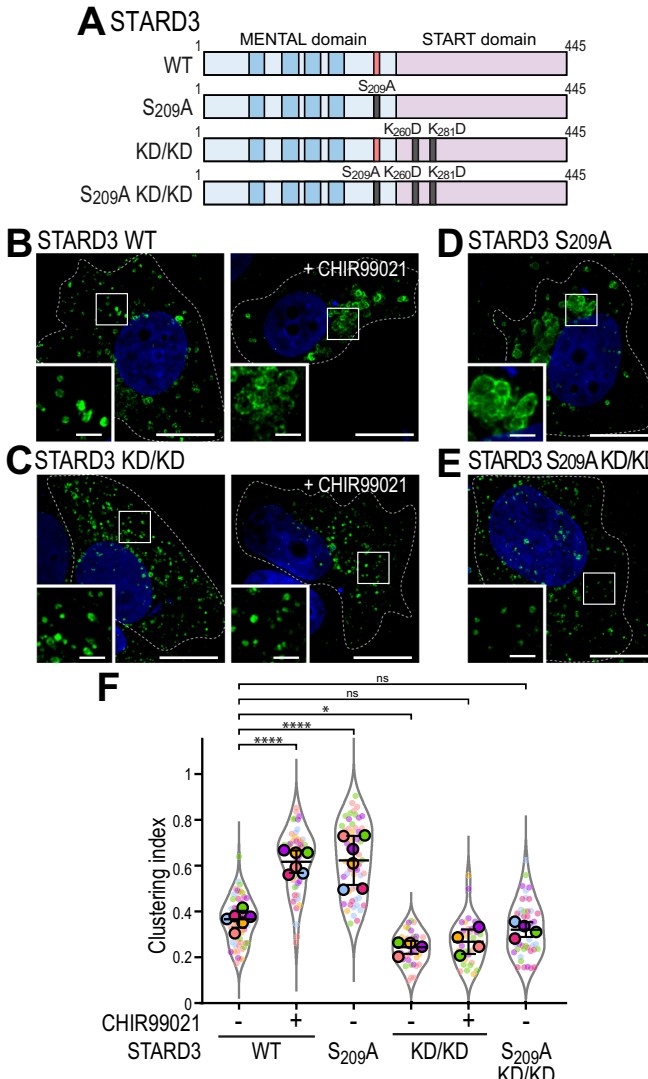

**Figure 8. The ability of the START domain to bind membranes is required for LE/Lys clustering induced by STARD3.**

(A) Schematic representation of the different STARD3 mutants used. (B–E) Representative images of MCF7 cells expressing WT STARD3 (B), STARD3 KD/KD (C), STARD3 $S_{209}$A (D), and STARD3 $S_{209}$A KD/KD (E). In (B) and (C), cells were left untreated (left) or treated with CHIR99021 (right). Cells were labeled with an anti-STARD3 antibody (green) and with Hoechst (nuclei; blue). Scale bars: 10 µm. (F) Clustering index of STARD3-positive vesicles of samples shown in (B–E). Data are displayed as Superplots showing the clustering index per cell (small dots) and its mean per independent experiment (large dots). Number of cells: STARD3 WT-NT: 65, STARD3 WT-CHIR99021: 59, STARD3 KD/KD-NT: 37, STARD3 KD/KD-CHIR99021: 32, STARD3 $S_{209}$A: 57, STARD3 S209A KD/KD: 47, from six independent experiments). Independent experiments are color-coded. Means and error bars (SD) are shown as black bars. One-way ANOVA with Tukey's multiple comparison test (*, $P < 0.05$; ****, $P < 0.0001$; $n = 6$ independent experiments; STARD3 WT-NT vs STARD3 WT-CHIR99021, $P < 10^{-4}$; STARD3 WT-NT vs STARD3 $S_{209}$A-NT, $P < 10^{-4}$; STARD3 WT-NT vs STARD3 KD/KD-NT, $P = 4.9 \times 10^{-2}$; STARD3 WT-NT vs STARD3 KD/KD-CHIR99021, $P = 0.17$; STARD3 WT-NT vs STARD3 $S_{209}$A KD/KD-NT, $P = 0.86$). Source data are available online for this figure.

VAPs (VAP-A, VAP-B, and MOSPD2) build MCSs through their MSP domain, which binds to FFAT motifs present in partner proteins on other organelles. FFAT motifs fall into two main categories: conventional FFAT motifs with the sequence $E_1F_2F_3$-$D_4A_5X_6E_7$ (Loewen et al, 2003; Mikitova and Levine, 2012) and Phospho-FFAT motifs, which are activated by phosphorylation (Di Mattia et al, 2020a; Ende et al, 2022; Guillén-Samander et al, 2021; James et al, 2024). The late endosome/lysosome (LE/Lys)-associated protein STARD3 contains a Phospho-FFAT motif with the sequence $Q_1F_2Y_3S_4P_5P_6E_7$. Our group previously demonstrated that phosphorylation of serine 209 (corresponding to $S_4$ in the motif) regulates STARD3's interaction with VAP proteins (Di Mattia et al, 2020a). In this study, we demonstrate that the kinase GSK3 directly phosphorylates the Phospho-FFAT motif of STARD3 and allows STARD3's interaction with VAPs and the formation of ER-LE/Lys contacts. Additionally, we identified a novel function of the unphosphorylated STARD3 protein, which directly binds to membranes and mediates homotypic interactions between LE/Lys (Fig. 9).

Our findings highlight the pivotal role of GSK3 in phosphorylating STARD3's Phospho-FFAT motif, which enables its interaction with VAP proteins and promotes the formation of ER–LE/Lys contact sites. Beyond this function, GSK3 has emerged as a broader regulator of inter-organelle contacts. For instance, GSK3β disrupts ER–mitochondria connectivity by interfering with the tethering complex formed between VAP-B and the mitochondrial protein PTPIP51 (protein tyrosine phosphatase-interacting protein 51, also known as RMDN3) (Stoica et al, 2014). Although the precise molecular mechanism remains unclear, this regulation involves TDP-43 and FUS, two proteins linked to amyotrophic lateral sclerosis (ALS) and frontotemporal dementia (FTD). Conversely, GSK3β inhibition by the mTORC2/Akt pathway, leads to an increase in ER-mitochondria contacts (Kalarikkal et al, 2024). In addition to its role in ER-mitochondria contact regulation, GSK3 also impairs the ACBD5-VAP-B complex that tethers peroxisomes to the ER (Costello et al, 2017; Kors et al, 2022). Specifically, GSK3β directly phosphorylates the conventional FFAT motif of ACBD5 at position 5, preventing its interaction with VAP-B and thereby reducing ER-peroxisome contacts (Kors et al, 2022). Altogether, this suggests that GSK3 coordinates the formation of contacts among at least 4 organelles: the ER, mitochondria, peroxisomes and LE/Lys.

The positioning of LE/Lys is shaped by their interaction with the ER (Jongsma et al, 2022, 2016; Rocha et al, 2009; Voeltz et al, 2024). Specifically, perinuclear ER forms contacts with LE/Lys via a protein complex made by the ER-resident E3 ubiquitin ligase RNF26, which associates with the LE/Lys protein TOLLIP in a ubiquitin-dependent manner, to promote the retention of LE/Lys near the nucleus (Jongsma et al, 2016). Moreover, the bidirectional transport of LE/Lys is regulated by ER-LE/Lys tethers, with Protrudin facilitating anterograde movement and ORP1L mediating retrograde transport (Raiborg et al, 2015; Rocha et al, 2009). Our findings reveal that STARD3 functions as a molecular switch controlling the balance between homotypic LE/Lys clustering and heterotypic ER-LE/Lys contacts. Disruption of its phosphorylation on $S_{209}$, either through GSK3 inhibition or mutation of the Phospho-FFAT motif, shifts the balance toward clustering. Likewise, STARD3-mediated LE/Lys clustering is recapitulated when all three VAPs (VAP-A, VAP-B, and MOSPD2) are simultaneously

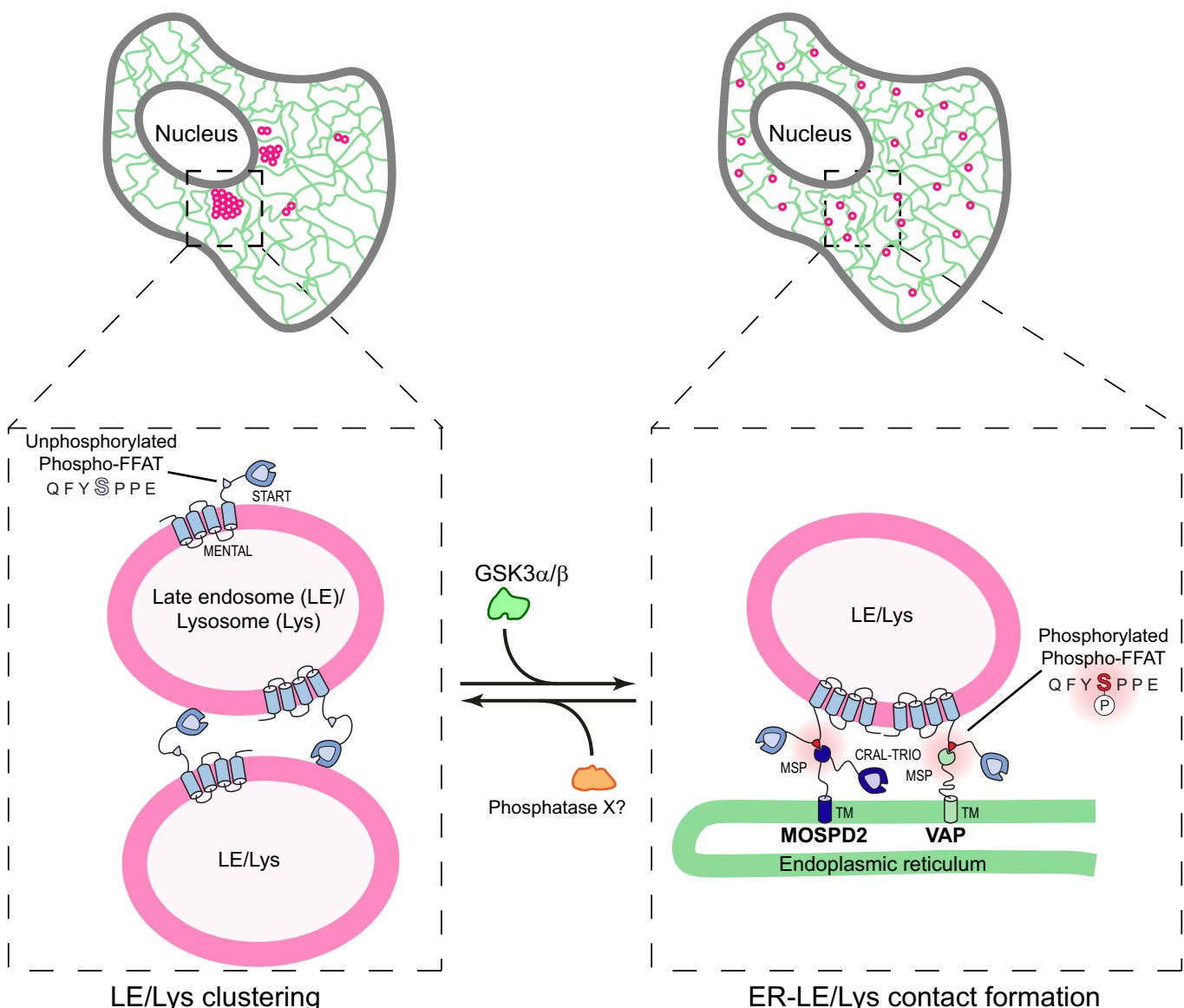

**Figure 9.  Schematic representation of ER-LE/Lys MCS regulation by GSK3.**

STARD3 regulates LE/Lys organization through phosphorylation-dependent mechanisms. When the Phospho-FFAT motif is unphosphorylated, STARD3 promotes LE/Lys clustering through electrostatic interactions between its START domain and LE/Lys membranes. Upon GSK3-mediated phosphorylation of the Phospho-FFAT motif, STARD3 binds to VAP proteins, promoting the formation of ER-LE/Lys MCSs and resulting in a more dispersed LE/Lys distribution.

depleted, highlighting both the essential role of ER tethers and the redundancy among VAPs. Interestingly, we previously observed that the establishment of ER-LE/Lys contacts occurs at the expense of homotypic LE/Lys contacts, suggesting the existence of a balance between these two types of interactions. Specifically, silencing the tethering protein MOSPD2 reduces ER-LE/Lys contacts, while simultaneously increasing homotypic LE/Lys interactions, as revealed by electron microscopy (Di Mattia et al, 2018). Thus, STARD3 appears to sit at the center of a tug-of-war between LE/Lys homotypic interactions and their association with the ER.

Our investigations into the molecular mechanism of STARD3-mediated LE/Lys clustering revealed that the START domain plays a key role through its ability to interact with negatively charged membranes. Fifteen human proteins possess a START domain,

which features an internal cavity that accommodates various lipids, including glycerolipids, sphingolipids, and sterols (Alpy and Tomasetto, 2014, 2005). Notably, STARD3 specifically binds cholesterol and mediates its transfer between the ER and LE/Lys (Tsujishita and Hurley, 2000; Wilhelm et al, 2017). This process requires its START domain to associate transiently with membranes to load and unload the sterol molecule, suggesting that this domain has some affinity for membranes, in line with our data. Consistently, the START domain of other proteins, such as STARD2, STARD11 and STARD4 has been shown to associate with membranes (Feng et al, 2000; Iaea et al, 2015; Kudo et al, 2008; Mesmin et al, 2011). This suggests that the interaction of the START domain of STARD3 with membranes is involved in the transport of sterol when in ER-LE/Lys contacts. However, in the

absence of phosphorylation of its Phospho-FFAT, STARD3 appears to engage in alternative homotypic membrane interactions via its START domain, leading to LE/Lys aggregation. The inability of phosphorylated STARD3 to generate homotypic interactions likely reflects differences in binding affinity, with its stronger interaction with VAPs prevailing over its weaker interaction with membranes. A comparable competitive mechanism likely operates for the mitochondria-bound protein MIGA2, which forms mitochondria-ER contacts via VAP binding when its Phospho-FFAT is active, but, when not phosphorylated, engages mitochondria-LD contacts by directly binding to the LD surface (Freyre et al, 2019). Homotypic interactions have been described for other organelles, such as peroxisomes and lipid droplets (Gong et al, 2011; Schrader et al, 2000). Although poorly characterized, these interactions likely play similar roles as contacts between organelles of different identities. The specific role of STARD3-mediated LE/Lys homotypic interactions remains to be elucidated. Given that STARD3 is a sterol transporter, it is plausible that these interactions facilitate sterol exchange and equilibrate its level in LE/Lys. Alternatively, homotypic interactions might retain LE/Lys in a sequestered state, thereby preventing their involvement in vesicular or non-vesicular trafficking.

An increasing number of studies show that VAP proteins interact with their partners in a phosphorylation-dependent manner. Proteomics studies have identified a large repertoire of VAP-A/VAP-B/MOSPD2 partners (Cabukusta et al, 2020; Di Mattia et al, 2018; Oughtred et al, 2019) and we found in an updated analysis that over 300 of these proteins contain a potential Phospho-FFAT motif (Dataset EV1 and Appendix Fig. S7) (Di Mattia et al, 2020a). Using the PhosphoSitePlus Kinase Prediction tool to identify kinases potentially phosphorylating the serine/threonine in position 4 highlighted consensus motifs for several kinases, supporting the idea that Phospho-FFAT-containing proteins are likely regulated by different upstream kinases depending on their sequence context.

To assess whether some of these motifs could be activated by GSK3, which requires a priming phosphorylation event, we searched for Phospho-FFAT-containing proteins with a serine or threonine at position 8 in addition to the core serine/threonine at position 4. This analysis identified 100 proteins, including the expected STARD3 and STARD3NL (Dataset EV1 and Appendix Fig. S7). Assuming phosphorylation at position 8, GSK3α or GSK3β consistently emerged as the top predicted kinases indicating they are strong candidates for activating the Phospho-FFAT motif in these proteins. Among these, VPS13D contains a Phospho-FFAT with a threonine at position 8, supporting a potential role for GSK3 in its activation and interaction with VAPs (Guillén-Samander et al, 2021). Similarly, SNX2 engages VAPs through a potentially GSK3-dependent Phospho-FFAT motif (Da Graça et al, 2022; Dong et al, 2016). Together, these findings suggest that GSK3 may broadly contribute to the activation of Phospho-FFAT-containing proteins.

In conclusion, our study identifies GSK3 as a key regulator of STARD3's ability to form ER-LE/Lys contacts. When unphosphorylated, STARD3 promotes homotypic clustering of LE/Lys through protein-membrane interactions. Phosphorylation thus acts as a molecular switch that controls a dual activity of STARD3, regulating both organelle positioning through homotypic interactions and the assembly of inter-organelle contact sites. We speculate that the balance between phosphorylated and unphosphorylated STARD3 modulates the spatial distribution of LE/Lys, shifting from intermingled with the ER to favor lipid transfer and inter-organelle communication, to homotypic LE/Lys interaction, which may support LE/Lys maturation. This balance is likely influenced by STARD3 expression levels and GSK3 activity. STARD3 is expressed ubiquitously at basal levels and is overexpressed in HER2-positive breast cancers (Lodi et al, 2023; Tomasetto et al, 1995). GSK3 activity is subject to complex regulation such as through inhibitory phosphorylation mediated by multiple kinases, and through its subcellular localization (Beurel et al, 2015). An important remaining question is to determine the relative abundance of phosphorylated versus unphosphorylated STARD3 in different cell types and in cancer, and how this distribution shapes the balance between heterotypic and homotypic interactions. Notably, GSK3 activity is linked to lysosome biology: GSK3 regulates mammalian target of rapamycin (mTOR) activity, a central kinase that integrates nutrient availability with cell growth. mTOR is localized to lysosomes when inactive, and its activation is associated with changes in lysosome positioning (Jia and Bonifacino, 2019; Korolchuk et al, 2011). The connection between these pathways and STARD3 function deserves further investigations.

Our data indicate that phosphorylation of STARD3 at serine 213 is a prerequisite for subsequent GSK3-mediated phosphorylation at serine 209. This mechanism is consistent with the well-established model of GSK3 substrate recognition, which requires a priming phosphorylation event four residues downstream of the target serine (ter Haar et al, 2001). Identifying the kinase responsible for this priming event, which initiates the hierarchical phosphorylation cascade, is crucial for a comprehensive understanding of the regulatory mechanisms governing STARD3 phosphorylation and its capacity to mediate membrane contact site formation. To conclude, this work highlights how dynamic phosphorylation events can shape organelle network organization and inter-organelle communication.

# Methods

**Reagents and tools table**

| Reagent/Resource | Reference or Source | Identifier or Catalog Number |
|---|---|---|
| **Experimental models** | | |
| MCF7 | ATCC | HTB-22 |
| HCC1954 | ATCC | CRL-2338 |
| HeLa | ATCC | CCL-2 |
| U2OS | ATCC | HTB-296 |
| COS7 | ATCC | CRL-1651 |
| **Recombinant DNA** | | |
| pQCXIP STARD3 | (Alpy et al, 2013) | |
| pQCXIP STARD3 ΔSTART | (Alpy et al, 2013) | |
| pQCXIP STARD3 S209A | (Di Mattia et al, 2020a) | |
| pQCXIP STARD3 ΔFFAT | (Wilhelm et al, 2017) | |
| pQCXIP STARD3 S209D/P210A | (Di Mattia et al, 2020a) | |
| pQCXIP STARD3NL | (Alpy et al, 2013) | |

| Reagent/Resource | Reference or Source | Identifier or Catalog Number |
|---|---|---|
| pQCXIP STARD3NL ΔFFAT | (Alpy et al, 2013) | |
| pQCXIP GFP-MOSPD2 | (Di Mattia et al, 2018) | Addgene # 186467 |
| pQCXIP GFP-MOSPD2 RD/LD | (Di Mattia et al, 2018) | Addgene # 186468 |
| pQCXIP mScarlet-ER [TM(SAC1)] | (Zouiouich et al, 2022) | Addgene # 186572 |
| pRK7N Flag-STARD3 | (Alpy et al, 2013) | |
| pQCXIP GFP-VAP-A | This study | |
| pQCXIP GFP-VAP-A KD/MD | This study | |
| pQCXIP GFP-VAP-B | This study | |
| pQCXIP GFP-VAP-B KD/MD | This study | |
| pQCXIP STARD3 S213A | This study | |
| pQCXIP STARD3 S217A | This study | |
| pQCXIP STARD3 S221A | This study | |
| pQCXIP STARD3 S213A/S217A/S221A | This study | |
| pQCXIP STARD3 KD/KD | This study | |
| pQCXIP STARD3 $S_{209}$A/ΔSTART | This study | |
| pLenti PGK Blast DEST (w524-1) | (Campeau et al, 2009) | Addgene # 19065 |
| pLenti PGK Blast$^R$ | This study | |
| pLenti PGK Puro$^R$ | This study | |
| pLenti PGK Puro$^R$ STARD3 | This study | |
| pQCXIP STARD3NL-START | This study | |
| pLJC5-Tmem192-3xHA | (Abu-Remaileh et al, 2017) | Addgene # 102930 |
| pQCXIP TMEM192-START | This study | |
| pEGFP-C2 | Clontech | |
| Lyso$_{LAMTOR1}$-EGFP | This study | |
| pQCXIP Lyso-START | This study | |
| pGEX4T STARD3 (195-445) | (Di Mattia et al, 2020a) | |
| pGEX4T STARD3 (195-445) S209Amber | (Di Mattia et al, 2020a) | |
| pGEX4T STARD3 (195-445) S213Amber | This study | |
| pGEX4T STARD3 (195-445) S213E | This study | |
| pGEX4T STARD3 (195-445) KD/KD | This study | |
| pGEX4T STARD3 (195-445) S209D/P210A | This study | |
| pGEX4T STARD3 (195-445) S209D/P210A/K260D/K281D | This study | |
| pET22b-START-STARD3 | (Tsujishita and Hurley, 2000) | |
| pET22b-START-STARD3 V2 (216-445) | This study | |
| pET22b-START-STARD3 KD/KD V2 (216-445) | This study | |
| pLJC5 EGFP-TMEM192 | This study | |
| pES002 MBP-GSK3β S9A-HA | (Gavagan et al, 2023) | Addgene # 196184 |
| pcDNA3 HA-GSK3β | (He et al, 1995) | Addgene # 14753 |
| pET15b VAP-B [1-210] | (Di Mattia et al, 2018) | |
| pLP1, pLP2, and pLP/VSVG | Invitrogen | |
| pCL-Ampho vector | Imgenex | |
| **Antibodies** | | |
| Rabbit anti-phospho-STARD3-pS$_{209}$ | (Di Mattia et al, 2020a) | 3144 |
| Rabbit anti-GFP | Torrey Pine Biolabs | TP401 |
| Mouse anti-Lamp1 | DSHB | H4A3 |

| Reagent/Resource | Reference or Source | Identifier or Catalog Number |
|---|---|---|
| Rabbit anti-GOLGA2/GM130 | Proteintech | 11308-1-AP |
| Mouse anti-EEA1 | BD Biosciences | 610457 |
| Mouse anti-GSK3α/β | Santa Cruz Biotechnology | sc-7291 |
| Rabbit anti-calnexin | Proteintech | 10427-2-AP |
| Rabbit anti-Tip60 | Cell Signaling Technology | 12058 |
| Rabbit anti-pS$_{86}$ Tip60 | Abcam | ab73207 |
| Mouse anti-VAP-A | Santa Cruz Biotechnology | 4C12 |
| Rabbit anti-VAP-B | (Kabashi et al, 2013) | |
| Rabbit anti-GAPDH | Sigma-Aldrich | G9545 |
| Rabbit anti-STARD3 | (Alpy et al, 2001) | 1611 |
| Mouse anti-STARD3 | (Wilhelm et al, 2017) | 3G11 |
| Mouse anti-STARD3 | (Di Mattia et al, 2018) | 1STAR-2G5 |
| Rabbit anti-STARD3NL | (Alpy et al, 2002) | 1545 |
| Mouse anti-MOSPD2 | (Di Mattia et al, 2018) | 1MOS-4E10 |
| AlexaFluor 488 | ThermoFisher Scientific | AB_2535792 and AB_141607 |
| AlexaFluor 555 | ThermoFisher Scientific | AB_2762848 and AB_162543 |
| AlexaFluor 647 | ThermoFisher Scientific | AB_2536183 and AB_162542 |
| Peroxidase-conjugated AffiniPure goat anti-rabbit | Jackson ImmunoResearch | 111-035-003 |
| Peroxidase-conjugated AffiniPure goat anti-mouse | Jackson ImmunoResearch | 115-035-003 |
| **Oligonucleotides and other sequence-based reagents** | | |
| PCR primers | Eurofins genomics | See Methods |
| ON-TARGETplus Non-targeting Control Pool | Horizon Discovery | D-001810-10-20 |
| SMARTpool ON-TARGETplus siRNAs targeting GSK3α | Horizon Discovery | L-003009-00-0005 |
| SMARTpool ON-TARGETplus siRNAs targeting GSK3β | Horizon Discovery | L-003010-00-0005 |
| SMARTpool ON-TARGETplus siRNAs targeting STARD3 | Horizon Discovery | L-017665-00-0010 |
| SMARTpool ON-TARGETplus siRNAs targeting MOSPD2 | Horizon Discovery | J-017039-09-0010 |
| SMARTpool ON-TARGETplus siRNAs targeting VAP-A | Horizon Discovery | L-021382-00-0010 |
| SMARTpool ON-TARGETplus siRNAs targeting VAPB | Horizon Discovery | L-017795-00-0010 |
| **Chemicals, Enzymes and other reagents** | | |
| Restriction enzymes | New England Biolabs | |
| AIM Terrific Broth including Trace elements | Formedium | AIMTB0260 |
| Isopropyl β-D-1-thiogalactopyranoside (IPTG) | Euromedex | EU0008 |
| Ampicillin | Euromedex | EU0400-E |
| Ampicillin | Sigma-Aldrich | A9518 |
| Kanamycin | Euromedex | UK0010 |
| Puromycine | Invivogen | ANT-PR-1 |
| Blasticidine | Invivogen | ANT-BL1 |
| Imidazole | Sigma-Aldrich | I2399 |
| EDTA-free protease inhibitor tablets cOmplete | Roche | 05056489001 |
| Phosphatase inhibitor tablets PhosSTOP | Roche | 4906845001 |

| Reagent/Resource | Reference or Source | Identifier or Catalog Number |
|---|---|---|
| Polyethylene glycol powder (Mn 20000) | Sigma-Aldrich | 81300 |
| HisPur Ni-NTA Chromatography Cartridges | Thermo Fisher Scientific | PI90098 |
| Amicon Ultra15 30 kDa and 10 kDa | Millipore | UFC903024 and UFC901024 |
| HiLoad 16/60 Superdex 200 | Cytiva | 9612066 |
| XK-16/70 column | Cytiva | GE28-9889-46 |
| Sephacryl S-200 HR | Cytiva | 17058401 |
| Glutathione Sepharose 4B | Cytiva | 17-0756 |
| SulfoLink Gel | Thermo Fisher Scientific | 20402 |
| GFP-Trap Magnetic Agarose | ChromoTek | GTMA-20 |
| NTA-Ni$^{2+}$ beads (PureProteome Nickel Magnetic Beads) | Millipore | LSKMAGH10 |
| Thrombin | Cytiva | GE27-0846-01 |
| Thrombin | Sigma-Aldrich | T7572 |
| O-phospho-L-serine | Sigma-Aldrich | P0878 |
| Slide-A-Lyzer MINI dialysis device (MWCO 3.5 kDa) | ThermoFisher Scientific | 69550 |
| SnakeSkin Dialysis Tubing | Thermo Fischer Scientific | 68035 |
| DMEM (1 g/L glucose) | Gibco | 31885 |
| DMEM (4.5 g/L glucose) | Gibco | 41966 |
| RPMI without HEPES | Gibco | 21875 |
| Fetal calf serum (FCS) | Pan Biotech | |
| Gentamicin | UGAP | 3636425 |
| Streptomycin/Penicillin solution | Fischer Scientific | 15140122 |
| Insulin | Sigma-Aldrich | I9278 |
| X-tremeGENE 9 DNA Transfection Reagent | Roche | 06365809001 |
| Lipofectamine RNAiMAX | Invitrogen | 13778-150 |
| CHIR99021 | Axon Medchem BV | Axon 1386 |
| Millex-HV Filter Unit (Sterile; 0.45 µm; 33 mm; PVDF membrane, hydrophilic) | Millipore | SLHV033RB |
| Polybrene | Sigma-Aldrich | TR-1003 |
| HEPES | Gibco | 15630 |
| HEPES | Sigma-Aldrich | H3375 |
| Potassium Acetate Gpr Rectapur | Prolabo | 26664.293 |
| M-PER buffer | ThermoFisher Scientific | 78501 |
| BCA protein assay | ThermoFisher Scientific | 23228 and 1859078 |
| PageBlue Protein Staining Solution | Thermo Fisher Scientific | 24620 |
| Protran nitrocellulose membranes, 0.45 µm | Amersham | 10600033 |
| Tran-Blot SD Semi-Dry Transfer Cell | BIO-RAD | |
| PageRuler Plus Prestained Protein Ladder, 10 to 250 kDa | Thermo Fisher Scientific | 26620 |
| Acrylamide/Bis-Acrylamide 40% Solution Ratio 29/1 | Euromedex | EU0077-B |
| N,N,N',N'-Tetramethyl-ethylenediamine | SERVA | 35930.01 |
| Ammonium persulfate | Sigma-Aldrich | A3678 |
| Tris | MP Biomedicals | 819623 |
| Sodium chloride | Carlo Erba Reagents | 479687 |
| Sodium fluoride | Sigma-Aldrich | 201154 |
| Sucrose | Sigma-Aldrich | S0389 |
| Glycerol | Sigma-Aldrich | G5516 |

| Reagent/Resource | Reference or Source | Identifier or Catalog Number |
|---|---|---|
| Phenylmethanesulfonyl fluoride | Sigma-Aldrich | P7626 |
| HyClone Dulbecco's Phosphate Buffered Saline | Cytiva | SH30013.03 |
| Glass coverslips | Marienfield | 0117520 |
| Glass bottom dish | MatTek | P35G-1.5-14-C |
| Paraformaldehyde | Electron Microscopy Sciences | 15710 |
| Triton X-100 | Sigma | T8787 |
| Tween 20 | Sigma | P2287 |
| Filipin | Sigma-Aldrich | F9765 |
| Bovine serum albumin | MP Biomedicals | 160069 |
| Hoechst-33258 | Sigma-Aldrich | B2261 |
| ProLong Gold | Invitrogen | PR36934 |
| Lowicryl HM20 | Electron Microscopy Sciences | 14345 |
| Amersham ECL start Western Blotting detection reagent | GE Healthcare | RPN3244 |
| SuperSignal West Pico Plus Chemiluminescent Substrate | ThermoFischer Scientific | 34580 |
| 18:1-PC (1,2-dioleoyl-sn-glycero-3-phosphocholine) | Avanti Polar Lipids | 850375 |
| 18:1-PS (1,2-dioleoyl-sn-glycero-3-phospho-L-serine) | Avanti Polar Lipids | 840035 |
| 16:0-18:1-PS (1-palmitoyl-2-oleoyl-sn-glycero-3-phospho-L-serine) | Avanti Polar Lipids | 840034 |
| 18:1 NBD-PE (1,2-dioleoyl-sn-glycero-3-phosphoethanolamine-N-(7-nitro-21,3-benzoxadiazol-4-yl) | Avanti Polar Lipids | 810156 |
| NBD-PA (1-oleoyl-2-{12-[(7-nitro-2-1,3-benzoxadiazol-4-yl)amino]dodecanoyl}-sn-glycero-3-phosphate) | Avanti Polar Lipids | 810176 |
| 18:1 MPB-PE 1,2-dioleoyl-sn-glycero-3-phosphoethanolamine-N-[4-(p-maleimidophenyl)butyramide] | Avanti Polar Lipids | 870012 |
| Nuclepore Track-Etched Membrane | Whatman | 800319, 800309 and 10417004 |
| Filter Supports | Avanti Polar Lipids | 610014-1Ea |
| Mini-extruder | Avanti Polar Lipids | 610023 |
| SYPRO Orange | ThermoFischer Scientific | S6650 |
| SYPRO Orange | Sigma-Aldrich | S5692 |
| Dithiothreitol (DTT) | Euromedex | EU0006-B |
| Pepstatin A | Sigma-Aldrich | P5318 |
| **Software** | | |
| ImageJ Fiji | (Schindelin et al, 2012) | |
| Stardist | (Schmidt et al, 2018) | |
| CellProfiler | (McQuin et al, 2018) | |
| Spyder 4.1 | Python 3.7 | |
| Prism 9. | Graphpad | |
| Pymol 3.1.0 | Schrodinger LLC | |
| Zeiss Atlas software | Fibics Inc | |
| Microscopy Image Browser | (Belevich et al, 2016). | |
| IMOD | (Kremer et al, 1996) | |
| ChimeraX | (Pettersen et al, 2021) | |

| Reagent/Resource | Reference or Source | Identifier or Catalog Number |
|---|---|---|
| Dynamics v6.1 | Wyatt technology | |
| BeStSel | (Kardos et al, 2025) | |
| ZeroCostDL4Mic | (von Chamier et al, 2021) | |
| **Other** | | |
| ÄKTA Start chromatography system | Cytiva | 29022094 |
| ÄKTA go chromatography system | Cytiva | 29383015 |
| NanoDrop 2000 | Thermo Scientifics | ND-2000 |
| Cell Disruptor TS SERIES | Constant Systems Ltd. | |
| J-815 spectrometer | Jasco | |
| Cell holder | Starna Scientific Ltd. | CH/2049 |
| Quartz cell | Starna Scientific Ltd. | 20/0/Q/0.5 |
| Spinning-disk confocal microscope (CSU-X1) | Nikon | |
| Live-SR Super Resolution module | Gataca Systems | |
| SP8 UV inverted confocal microscope | Leica | |
| Elyra 7 | Zeiss | |
| HPM 10 | Abra Fluid AG | |
| CM12 transmission electron microscope | Philips | |
| Orius 1000 CCD camera | Gatan | |
| Auriga 60 FIB-SEM | Zeiss | |
| Vibra-Cell 75041 | Bioblock Scientific | |
| Wyatt DynaPro-99-E-50 System | Protein Solutions | |
| Ultra Micro Cell, Type 105.252-QS, HP Quartz Glass, path length 1.5 ×1.5 mm, center height 8.5 mm | Hellma | 105.252-QS |
| ProteomeLab XL-I analytical ultracentrifuge | Beckman Coulter | |
| Optima MAX-XP ultracentrifuge | Beckman Coulter | |
| Fusion-FX | Vilber Loumat | |
| Spectrophotometer | Safas monaco | UVmc2 |
| *E. coli* C321 ΔA bacteria | (Lajoie et al, 2013) | Addgene # 48998 |
| *E. coli* Rosetta2 (DE3) pLysS | Sigma-Aldrich | 71403 |
| *E. coli* BL21-GOLD(DE3) | Agilent | 230132 |

## Cloning and constructs

The retroviral expression vector (pQCXIP) coding for STARD3, STARD3 ΔSTART, STARD3 $S_{209}A$, STARD3 ΔFFAT, STARD3 $S_{209}D/P_{210}A$, STARD3NL, and STARD3NL ΔFFAT were previously described (Alpy et al, 2013; Di Mattia et al, 2020a; Wilhelm et al, 2017). The GFP-MOSPD2 (WT Addgene plasmid # 186467; http://n2t.net/addgene:186467; RRID:Addgene_186467 and RD/LD mutant Addgene plasmid # 186468; http://n2t.net/addgene:186468; RRID:Addgene_186468), pQCXIP mScarlet-ER [TM(SAC1)] (Addgene plasmid # 186572; http://n2t.net/addgene:186572; RRID:Addgene_186572), and Flag-tagged STARD3 expression vectors were previously described (Alpy et al, 2013; Di Mattia et al, 2018; Zouiouich et al, 2022).

The retroviral expression vector pQCXIP encoding GFP-VAP-A (WT and KD/MD mutant) and GFP-VAP-B (WT and KD/MD mutant) were constructed by cloning PCR-amplified fragments from pEGFPC1-hVAP-A (Addgene plasmid # 104447), pEGFPC1-hVAP-A KD/MD (Addgene plasmid # 104449), pEGFPC1-hVAP-B (Addgene plasmid # 104448), and pEGFPC1-hVAP-B KD/MD (Addgene plasmid # 104450). The fragments were amplified using the primer 5'-AATTG ATCCG CGGCC ACCAT GGTGA GCAAG GGCGA G-3' and either VAP-A: 5'-GCGGA ATTCC GGATC GCTAC AAGAT GAATT CCCT AGAAA GAATC CAATG-3' or VAP-B: 5'-GCGGA ATTCC GGATC GCTAC AAGGC AATCT TCCCA ATAAT TACAC-3'. Cloning was performed using the SLiCE (Seamless Ligation Cloning Extract) method (Okegawa and Motohashi, 2015) into the NotI and BamHI linearized pQCXIP vector.

The expression vectors coding for STARD3 $S_{213}A$, STARD3 $S_{217}A$, STARD3 $S_{221}A$, STARD3 $S_{213}A/S_{217}A/S_{221}A$, and STARD3 $K_{260}D/K_{281}D$ were obtained by assembling 2 PCR fragments generated using STARD3 cDNA as template, the following central primers: STARD3 $S_{213}A$: 5'-CCCCC AGAAG CCTTT GCAGG GTCTG ACAAT GAATC AG-3' and 5'-CCCTG CAAAG GCTTC TGGGG GTGAA TAGAA CTGTC CC-3'; STARD3 $S_{217}A$: 5'-CCTTT GCAGG GGATG ACAAT GAATC AGATG AAG-3' and 5'-CATCT GATTC ATTGT CATCC CCTGC AAAGG ATTCT G-3'; STARD3 $S_{221}A$: 5'-CCTTT GCAGG GGCTG ACAAT GAATC AGATG AAG-3' and 5'-ATTCA TTGTC AGCCC CTGCA AAGGA TTC-3'; STARD3 $S_{213}A/S_{217}A/S_{221}A$: 5'-CACCC CCAGA AGCCT TGCA GCGGC TGACA ATGAA GCAGA TGAAG AAGTT GCTGG GAAGA AAAG-3' and 5'-CAGCA ACTTC TTCAT CTGCT TCATT GTCAG CCGCT GCAAA GGCTT CTGGG GGTGA ATAGA ACTGT CCCTC-3', STARD3 $K_{260}D/K_{281}D$: 5'-GGACA CCGTG TACAC CATTG AAGTT CCCTT TCACG GCGAC ACGTT TATCC TGAAG ACCTT CCTGC CCTG-3' and 5'-GTACA CGGTG TCCCC ATATT CATTA TTCTT CTCAA GTCC CAGTT CTCTT CCTGG GCCAA GATCT GGTCC ACC-3' and the following peripheral primers: 5'-GGAAT TGATC CGCGG CCACC AGGAT GAGCA AGCTG CCCAG GGAGC TGACC CG-3' and 5'-GGGCG GAATT CCGGA TCACG CCCGG GCCCC CAGCT CGCTG ATGCG C-3'. PCR fragments were cloned using the SLiCE method into the NotI and BamHI linearized pQCXIP vector. STARD3 $S_{209}D/P_{210}A/S_{213}A$ expression construct in the pQCXIP vector was obtained using the same strategy as above with STARD3 $S_{209}D/P_{210}A$ cDNA as template.

STARD3 $S_{209}A$/ΔSTART expression construct in the pQCXIP vector was obtained using the same strategy as above with STARD3 ΔSTART cDNA as template, and the following central primers: 5'-GATTC TGGGG GTGCA TAGAA CTGTC CCTCG G-3' and 5'-GACAG TTCTA TGCAC CCCCA GAATC CTTTG C-3'.

The pLenti PGK Blast$^R$ vector was generated by removing the Gateway cassette from pLenti PGK Blast DEST (w524-1) vector (gift from Eric Campeau & Paul Kaufman; Addgene plasmid # 19065; http://n2t.net/addgene:19065; RRID:Addgene_19065) (Campeau et al, 2009) via SalI digestion, followed by religation. The pLenti PGK Puro$^R$ vector was constructed by replacing the Blasticidine resistance gene coding sequence (removed via XmaI and KpnI digestion) with the puromycin resistance gene, which was amplified by PCR using pQCXIP as template and the primers 5'-GCAAA AAGCT CCCGG CCTTC CATGA CCGAG TACAA GCC-3' and 5'-TCATT GGTCT TAAAG TCAGG CACCG GGCTT GCGGG TCA-3'. To construct the STARD3 expression

plasmid, the STARD3 coding sequence was inserted into AgeI/BsrGI-linearized pLenti PGK Puro$^R$ using a PCR fragment amplified with the primers: 5'-CAGGG GGATC ACCGG ACCAG GATGA GCAAG CTGCC C-3' and 5'-TCAAC CACTT TGTAC TCACG CCCGG GCCCC CAGCT C-3'. Cloning was performed using the SLiCE method.

The chimeric construct of STARD3NL and STARD3 was generated by amplifying the STARD3NL coding region via PCR with the primers 5'-GAGAG GATCC GCCGC CATGA CCAC CTGCC AGAAG AC-3' and 5'-CTGGC GGATG TACTC CCGCT CTAGT TCTAA AAGTG GTTTC TC-3'. The STARD3 START domain (residues 235–445) was amplified using the primers 5'-GAGAA ACCAC TTTTA GAACT AGAGC GGGAG TACAT CCGCC AG-3' and 5'-GAGAC AATTG TCACG CCCGG GCCCC CAGCT CG-3'. The two fragments were assembled by linked PCR, digested with BamHI and MfeI, and ligated into the BamHI- and EcoRI-linearized pQCXIP vector.

The chimeric construct of TMEM192 and STARD3 was constructed by amplifying the transmembrane coding region of TMEM192 using PCR with the primers 5'-GGAAT TGATC CGCGG CCGCG GCGGG GGGCA GGATG GAGG-3' and 5'-GGCAA GATAC CATCG TTCTT CTTCA GTAT ATCAG GCTCT GGTTT AGCTT TATTA AATCT CCGG-3', using pLJC5-Tmem192-3xHA as template (a gift from David Sabatini, Addgene plasmid # 102930; http://n2t.net/addgene:102930; RRID:Addgene_102930 (Abu-Remaileh et al, 2017)). The STARD3 sequence (START domain without an active FFAT) was amplified using the primers 5'-ATACT TGAAG AAGAA CGATG GTATC TTGCC GCCCA GGTTG CTGTT GC-3' and 5'-GGGCG GAATT CCGGA TCACG CCCGG GCCCC CAGCT CGC-3'. The two fragments were assembled and cloned into the pQCXIP vector using the SLiCE method.

The sequence encoding the 39 first residues of LAMTOR1 was amplified by PCR using the primers 5'-GGACT CAGAT CTCGA CCGGC CATGG GGTGC TGCTA CAGCA GCG-3', 5'-GTCGA CTGCA GAATT CGTTG GGCTC GGCTC CATTG AGAGC TTTGG TAGGG GG-3', and 5'-GCCAT GGGGT GCTGC TACAG CAGCG AGAAC GAGGA CTCGG ACCAG GACCG AGAGG AGCGG AAGCT GCTGC TGGAC CCTAG CAGCC CCCCT ACCAA AGCT CTCAA TGGAG CCGAG CCCAA-3'. The amplified fragment was cloned into the pEGFP-N2 vector (Clontech), linearized with XhoI and EcoRI, using the SLiCE method to generate the Lyso$_{LAMTOR1}$-EGFP plasmid. The chimeric construct between LAMTOR1 (1-39) and the START domain of STARD3 was constructed by assembling 2 PCR fragments amplified using 5'-GGAAT TGATC CGCGG CCACC AGGAT GGGGT GCTGC TACAG CAGCG AGAAC GAG-3' and 5'-CGAAG CTTGA GCTCG AGATC TTCCG GAACC GTTGG GCTCG GCTCC ATTGA GAGCT TGGT AGG-3' (LAMTOR1 fragment; Lyso$_{LAMTOR1}$-EGFP template) and 5'-CGAGC TCAAG CTTCG GCAGG GTCTG ACAAT GAATC AGATG AAGAA GTTGC TGGG-3' and 5'-GGGCG GAATT CCGGA TCACG CCCGG GCCCC CAGCT CGCTG ATGCG C-3' (STARD3 fragment) cloned using the SLiCE method into the NotI and BamHI linearized pQCXIP vector.

The plasmid encoding STARD3 (196–445) hereafter referred to as $_C$STD3, and the plasmid encoding $_C$STD3 pS$_{209}$ in which the S$_{209}$ codon was replaced by an amber codon (TAG), were previously described (Di Mattia et al, 2020a). To obtain the $_C$STD3 pS$_{213}$ and

$_C$STD3 S$_{213}$E encoding constructs, the S$_{213}$ codon was replaced by an amber and a Glu codon, respectively, by PCR using STARD3 cDNA template, the following forward primers $_C$STD3 pS$_{213}$: 5'-GGTTC CGCGT GGATC CTGCC TGTTC TCCGG TGCTC TGTCC GAGGG ACAGT TCTAT TCACC CCCAG AATAG TTTGC AGGGT CTGAC AATGA ATCAG ATG-3'; $_C$STD3 S$_{213}$E: 5'-GGTTC CGCGT GGATC CTGCC TGTTC TCCGG TGCTC TGTCC GAGGG ACAGT TCTAT TCACC CCCAG AAGAA TTCGC AGGGT CTGAC AATGA ATCAG ATG-3', and the reverse primer 5'-GTCGA CCCGG GAATT CCGGT CACGC CCGGG CCCCC AGCTC-3'. The PCR fragments were cloned by SLiCE in the plasmid encoding $_C$STD3 (Di Mattia et al, 2020a) linearized by BamHI and EcoRI.

The pET22b-START-STARD3 expression vector, a kind gift from J.H. Hurley (Tsujishita and Hurley, 2000), was opened with NdeI and NcoI, the following primers 5'-TTT AAG AAG GAG ATA TAC AT ATG GGC CAT CAT CAT CAT CAT CAC ATG GGG TCT GAC AAT GAA TC-3'; 5' GAT TCA TTG TCA GAC CCC ATG TGA TGA TGA TGA TGA TGG CCC ATA TGT ATA TCT CCT TCT TAAA-3' were hybridized and cloned by the SLiCE method to generate the pET22b-START-STARD3 V2 (216-445) vector.

The plasmid encoding EGFP-tagged TMEM192 was generated by inserting the EGFP coding sequence, amplified by PCR from pEGFP-C1 using primers 5'-CGC TAG CGC TAC CGG GCC ACC ATG GTG AGC AAG GGC GAG G-3' and 5'-AAT ACT CTG GGT GAT ATC AAG GAC CGT CTC CAT CCT GCC CCC CGC CGC AGC TCG AGA TCT GAG TCC GGA CTT GTA CAG CTC GTC CAT GC-3', using the SLiCE method into pLJC5-Tmem192-3xHA linearized with AgeI and EcoRV.

pES002 MBP-GSK3β S9A-HA-His was a gift from Jesse Zalatan (Addgene plasmid # 196184; http://n2t.net/addgene:196184; RRID:Addgene_196184) (Gavagan et al, 2023). pcDNA3 HA-GSK3β was a gift from Jim Woodgett (Addgene plasmid # 14753; http://n2t.net/addgene:14753; RRID:Addgene_14753) (He et al, 1995).

All constructs were verified by DNA sequencing (Eurofins).

## Protein production and purification

MBP-GSK3β S9A-HA-6His and the 6His-tagged START domain of STARD3 (START-STARD3 V2 216-445) were produced and purified similarly. They were expressed in *E. coli* Rosetta2 (DE3) pLysS bacteria in auto-inducible medium (Terrific Broth including Trace elements; Formedium) supplemented with ampicillin (100 μg/mL); cells were incubated at 37 °C until the optical density reached OD$_{600nm}$ = 0.5 and at 20 °C for 16 h. Bacteria were pelleted at 3500 × g for 15 min at 4 °C and resuspended in lysis buffer (50 mM Tris pH 7.5, 150 mM NaCl, 35 mM imidazole, 2 mM dithiothreitol (DTT), EDTA-free protease inhibitor tablets (cOmplete, Roche)). Cells were lysed using a Cell Disruptor TS SERIES (Constant Systems Ltd), and the lysate was first centrifuged at 3500 × g for 15 min, then at 50,000 × g for 45 min, and filtered through a 0.45 μm membrane (Millipore Express Plus). Purification was performed using an ÄKTA Start chromatography system (Cytiva) with HisPur Ni-NTA Chromatography Cartridges. Proteins were eluted with elution buffer (50 mM Tris-HCl pH 7.5, 150 mM NaCl, 2 mM DTT, 625 mM imidazole) and further purified by gel filtration (HiLoad 16/60 Superdex 200, Cytiva) in GF buffer (50 mM Tris-HCl pH 7.5, 150 mM NaCl, 2 mM DTT, 15%

glycerol). Proteins were concentrated using Amicon Ultra15 30 kDa or 10 kDa filters (Millipore), and protein concentration was determined by UV-spectroscopy.

$_C$STD3 S$_{213}$E was produced as a GST fusion protein in *E. coli* Rosetta2 (DE3) pLysS bacteria incubated in auto-inducible medium (Formedium) supplemented with ampicillin (100 µg/mL) at 37 °C overnight. Bacteria were lysed as described above in lysis buffer (50 mM Tris-HCl pH 7.5, 150 mM NaCl, 2 mM DTT, cOmplete, EDTA-free protease inhibitor tablets (Roche)). Glutathione Sepharose 4B beads (17-0756, GE Healthcare) were washed with lysis buffer, and the lysate was then applied to the beads for 2 h at 4 °C under agitation. After three washing steps with lysis buffer, the beads were incubated with thrombin at 4 °C for 16 h to cleave the GST fusion and release $_C$STD3 S$_{213}$E. The protein was recovered in the supernatant after centrifugation, and the beads were washed three times with lysis buffer. The fractions were pooled and concentrated using Amicon Ultra15 15k filters (Millipore). Protein concentration was determined by UV-spectroscopy.

$_C$STD3 S$_{209}$D/P$_{210}$A (used for circular dichroism, flotation, and aggregation assays and hereafter called $_C$STD3) and $_C$STD3 S$_{209}$D/P$_{210}$A/K$_{260}$D/K$_{281}$D ($_C$STD3 KD/KD) were expressed in *E. coli* BL21-GOLD(DE3) competent cells (Agilent) grown in Luria Bertani Broth (LB) medium at 30 °C overnight upon induction with 1 mM isopropyl β-D-1-thiogalactopyranoside (IPTG), when the optical density of the bacterial suspension, measured at 600 nm (OD$_{600}$), reached a value of 0.6–0.7. Bacterial cells were harvested and re-suspended in cold TN buffer (50 mM Tris pH 7.4, 150 mM NaCl, 2 mM DTT) supplemented with 1 mM PMSF, 1.6 µM pepstatin A and cOmplete, EDTA-free protease inhibitor tablets (Roche). Cells were lysed in a Cell Disruptor TS SERIES (Constant Systems Ltd.), and the lysate was centrifuged at 186,000 × *g* for 1 h 30. Then, the supernatant was applied to Glutathione Sepharose 4B (Cytiva) for 3 h 30 at 4 °C. The beads were then washed four times with TN buffer devoid of protease inhibitors; the beads were incubated with thrombin overnight at 4 °C to cleave off the $_C$STD3 construct from the GST domain. Each construct was recovered in the supernatant after several cycles of centrifugation and washing of the beads and concentrated. $_C$STD3 KD/KD was further loaded onto an XK-16/70 column packed with Sephacryl S-200 HR to be purified by size-exclusion chromatography. The fractions with ~100% pure $_C$STD3 construct were pooled, concentrated, and supplemented with 10% (v/v) pure glycerol (Sigma). Aliquots were prepared, flash-frozen in liquid nitrogen, and stored at −80 °C for both constructs. For some experiments, a volume of 100 µL from a stock solution of $_C$STD3 and $_C$STD3 KD/KD was applied onto a 0.5 mL Zeba spin desalting column (7 kDa molecular weight cut-off) equilibrated with freshly degassed HK buffer, according to manufacturer's indications, to remove DTT from the protein, and immediately used. The protein concentration was determined by measuring the absorbance at λ = 280 nm ($\varepsilon$ = 30,160 M$^{-1}$.cm$^{-1}$).

$_C$STD3 pS209 and $_C$STD3 pS213 were expressed in *E. coli* C321 ΔA bacteria [gift from George Church; Addgene # 48998 (Lajoie et al, 2013)] as described in Di Mattia et al, 2020a. Protein production was performed in auto-inducible medium (Formedium) supplemented with ampicillin (100 µg/mL), kanamycin (20 µg/mL), and 20 mM O-phospho-L-serine (P0878, Sigma) for 48 h at 20 °C. All purification steps were conducted as described for $_C$STD3 S213E.

MSP-VAPB$_{6HIS}$ was produced as described in (Di Mattia et al, 2020a).

## Circular dichroism

The experiments were performed on a Jasco J-815 spectrometer at room temperature with a quartz cell of 0.05 cm path length (Starna Scientific Ltd.). Each protein was dialyzed in a Slide-A-Lyzer MINI dialysis device (MWCO 3.5 kDa) three times against 20 mM Tris pH 7.4, 120 mM NaF buffer for 30 min to remove glycerol or DTT from the protein stock and to exchange buffer. Then, the samples were subjected to ultracentrifugation at 100,000 × *g* for 20 min at 20 °C to pellet protein aggregates, and the supernatant was collected. Each CD spectrum is the average of ten scans recorded from λ = 185 to 260 nm with a bandwidth of 1 nm, a step size of 0.5 nm, and a scan speed of 50 nm.min$^{-1}$. Protein concentration was determined at λ = 280 nm by spectrophotometry. A control spectrum of buffer was subtracted from each protein spectrum. The percentages of protein secondary structure were estimated by analyzing their UV CD spectrum (in the 185–260 nm range) using the BeStSel method provided online.

## Antibody purification

The anti-phospho-STARD3-pS$_{209}$ (rabbit polyclonal, 3144, IGBMC) antibody was described previously (Di Mattia et al, 2020a). The serum underwent a two-step purification: first, the serum was applied to an affinity chromatography column bearing the phosphorylated synthetic peptide (CDGQFYpSPPESEA), and the bound fraction was eluted using glycine buffer (100 mM glycine pH 2.5). Next, the eluted fraction was further purified on a column bearing the non-phosphorylated synthetic peptide (CDGQFYSP-PESEA), and the unbound fraction was collected. The buffer was exchanged for 1x TBS (24.8 mM Tris pH 7.5, 137 mM NaCl, 2.7 mM KCl) through successive dialysis steps using SnakeSkin Dialysis Tubing (Thermo Fischer Scientific, 68035). Then, the antibody was concentrated by incubating the dialysis tube with polyethylene glycol powder (Mn 20000, 81300, Sigma-Aldrich) at 4 °C during 5 h. The recovered antibody was supplemented with 50% glycerol and stored at –20 °C. To prepare the affinity chromatography columns, the peptides were coupled to SulfoLink Gel (Thermo Fisher Scientific, 20402) according to the manufacturer's instructions.

## Cell culture, transfection and infection

HeLa cells (American Type Culture Collection (ATCC) CCL-2, RRID:CVCL_0030) were maintained in DMEM (1 g/L glucose) supplemented with 5% fetal calf serum (FCS) and 40 µg/mL gentamicin. 293 T cells (ATCC CRL-3216, RRID:CVCL_0063) were cultured in DMEM (4.5 g/L glucose) supplemented with 10% FCS, 100 IU/mL penicillin, and 100 µg/mL streptomycin. MCF7 cells (ATCC-HTB-22, RRID: CVCL_0031) were cultured in DMEM (1 g/L glucose) supplemented with 10% FCS, 0.6 µg/mL insulin, and 40 µg/mL gentamicin. HCC1954 cells (ATCC CRL-2338, RRID:CVCL1259) were cultured in RPMI without HEPES supplemented with 10% FCS and 40 µg/mL gentamicin. U2OS cells (ATCC HTB-296, RRID: CVCL_0042) were cultured in DMEM (4.5 g/L glucose) supplemented with 10% FCS and 40 µg/mL gentamicin. COS7 (ATTC CRL-1651, RRID:CVCL_0224) were cultured in DMEM (1 g/L glucose) supplemented with 5% FCS and 40 µg/mL gentamicin. Cells are regularly tested for mycoplasma

infection. HeLa, 293 T, MCF7 and HCC1954 cell lines were authenticated in 2021.

Cells were transfected using X-tremeGENE 9 DNA Transfection Reagent (Roche). Lentiviral particles were generated by co-transfecting pLenti PGK Puro$^R$ vectors with three packaging plasmids (pLP1, pLP2, and pLP/VSVG from Invitrogen) into 293 T cells. Retroviral particles were generated by co-transfecting pQCXIP vectors with pCL-Ampho vector (Imgenex) into 293 T cells. Viral supernatants were filtered through a 0.45 μm membrane, supplemented with 10 μg/mL polybrene and 20 mM HEPES, and then used to infect MCF7 cells or HeLa cells, which were then selected using puromycin (0.5 μg/mL) or blasticidin (4 μg/mL).

siRNA transfections were performed using Lipofectamine RNAiMAX (Invitrogen) following the manufacturer's instructions. Control siRNA (D-001810-10) and siRNAs targeting: GSK3α (L-003009-00-0005), GSK3β (L-003010-00-0005), STARD3 (L-017665-00-0005), MOSPD2 (J-017039-09), VAP-A (L-021382-00-0010) and VAPB (L-017795-00-0010) were SMARTpool ON-TARGETplus obtained from Horizon Discovery.

When specified, cells were treated with the GSK3 inhibitor CHIR99021 (Axon Medchem BV, cat. no. Axon 1386).

## Immunofluorescence

Thirty thousand cells were grown on glass coverslips, fixed in 4% paraformaldehyde in PBS for 15 min, and permeabilized with 0.1% Triton X-100 in PBS for 10 min or with 0.5 mg/mL filipin for 20 min. After blocking with 1% bovine serum albumin in PBS (PBS-BSA), cells were incubated overnight at 4 °C with the primary antibody in PBS-BSA. The primary antibodies used were: rabbit anti-GFP (1:1000; Torrey Pine Biolabs TP401, RRI-D:AB_10013661), mouse anti-Lamp1 (1:50; DSHB H4A3, RRI-D:AB_2296838), rabbit anti-GOLGA2/GM130 (1/1000; Proteintech 11308-1-AP, RRID:AB_2115327), mouse anti-EEA1 Clone 14 (1:1000; BD Biosciences 610457, RRID:AB_397830), rabbit anti-calnexin (1:1000; Proteintech 10427-2-AP, RRID:AB_2069033) rabbit anti-STARD3 (1611; 1:2000; IGBMC), mouse anti-STARD3 (3G11; 1:100; IGBMC), rabbit anti-pS$_{209}$ STARD3 (3144; 1:200; IGBMC), and rabbit anti-STARD3NL (1545; 1:1000, IGBMC). Cells were washed twice with PBS and incubated for 30 min with Hoechst-33258 (1:10,000, B2261 Sigma-Aldrich) to stain nuclei and the appropriate secondary antibodies (AlexaFluor 488 (RRID: AB_2535792 and AB_141607), AlexaFluor 555 (RRID: AB_2762848 and AB_162543), or AlexaFluor 647 (RRID: AB_2536183 and AB_162542) from ThermoFisher Scientific). After two washes with 1x PBS, the slides were mounted in ProLong Gold (Invitrogen). Observations were made using a spinning-disk confocal microscope (CSU-X1; Nikon, 100×, NA 1.4) with a Live-SR Super Resolution module (Gataca systems). When indicated, observations were made using a Leica SP8 UV inverted confocal microscope (63×, NA 1.4). For SIM², image acquisition was done on a Zeiss Elyra 7 device (63x, NA 1.4).

## Colocalization analysis

Colocalization was quantified using the Pearson correlation coefficient, calculated using the Colocalization Threshold plugin in Fiji software.

## Quantification of endosome proximity

Thirty thousand cells were plated in 24-well plates on glass coverslips and transfected the same day with different plasmids. After 24 h, cells were treated overnight with 5 μM CHIR99021. The following day, immunofluorescence was performed as described above. Images were acquired using identical settings for all samples (laser power, number of z-slices, and exposure length). For image processing, z-stacks were projected using the maximum intensity method in ImageJ Fiji (Schindelin et al, 2012). STARD3-positive endosomes were segmented using a custom-trained Stardist model (Schmidt et al, 2018), generated on Google Colab using Zero-CostDL4Mic. Cells were manually segmented in CellProfiler (McQuin et al, 2018) to create individual cell masks, and LE/Lys were assigned to their corresponding cells. For each cell, the proportion of LE/Lys in contact with at least one other LE/Lys was measured using the MeasureOjectNeighbors module (Method: Adjacent). Data analysis was carried out using Spyder 4.1 (Python 3.7) and GraphPad Prism 9.

## Electron microscopy sample preparation

Electron microscopy was performed as previously described (Alpy et al, 2013; Zouiouich et al, 2022). Cells grown on carbon-coated sapphire disks were cryoprotected with DMEM containing 10% FCS and frozen at high pressure (HPM 10 Abra Fluid AG). Samples were then freeze-substituted and embedded in lowicryl HM20.

## Transmission electron microscopy

Thin sections were collected on formvar-/carbon-coated nickel slot grids and stained with uranyl acetate and lead citrate. Imaging was performed with a transmission electron microscope (Philips CM12) coupled to an Orius 1000 CCD camera (Gatan).

## FIB-SEM nanotomography

Resin embedded cell monolayers were prepared for FIB-SEM imaging as follows: resin blocks were mounted on SEM stubs using conductive adhesive carbon tabs (Micro to Nano 15-000409) and made conductive by applying silver paint (Agar AGG3692) on both the stub and the sides of the resin block. The top surface of the resin blocks was made conductive by sputter-coating with 12 nm of platinum (Leica EM ACE600 high vacuum sputter coater).

Samples were imaged using a Zeiss Auriga 60 FIB-SEM microscope equipped with a multi GIS system, InLens and SESI secondary electron detectors, and an ESB backscattered electron detector. During imaging, the resin blocks were tilted at 54° to allow for precise in-chamber GIS platinum deposition and perpendicular ion beam milling. Sample preparation and volume imaging were performed using Zeiss Atlas software (Fibics Inc, version 5.1). The resin blocks were imaged in continuous mode (imaging while slicing) with drift compensation, −36° tilt correction, automatic focusing and stigmation enabled. One image stack per condition was selected for further processing. The imaging parameters were as follows: HeLa/Ctrl: dwell time of 7 ms and line averaging of 3; ESB detector; The voxel size was 8 nm in x/y and 8 nm in z, with a total of 994 frames acquired, corresponding

to a volume of $5.6 \times 10^3$ µm³. HeLa/STARD3: dwell time of 0.7 ms and line averaging of 20; InLens and SESI detectors; SEM EHT set to 1.5 kV, with a 30 µm aperture; The voxel size was 5 nm in x/y and 10 nm in z, with a total of 810 frames acquired, corresponding to a volume of $3.4 \times 10^3$ µm³. HeLa/STARD3 FYAA: dwell time of 2.4 ms and line averaging of 4; ESB and SESI detectors; SEM EHT set to 1.5 kV, with a 30 µm aperture; The voxel size was 8 nm in x/y and 8 nm in z, with a total of 600 frames were acquired, corresponding to a volume of $11.8 \times 10^3$ µm³.

## FIB-SEM image processing

For HeLa/STARD3 and HeLa/STARD3 FYAA samples, images recorded using the InLens and SESI detectors were merged with a balance of 90% and 10%, respectively. A Gaussian filter with a sigma of 1 was applied with Fiji (ImageJ version 1.54p) (Schindelin et al, 2012), followed by contrast-limited adaptive histogram equalization (CLAHE), with the maximum slope adjusted between 2 and 4 depending on the condition. For the HeLa/STARD3 stack, a Fourier high-pass filter (Fiji FFT bandpass filter, large structures = 2000 pixels, small structures = 1 pixel) was additionally applied to reduce illumination gradients introduced by the InLens detector. Images were aligned using the Scale-Invariant Feature Transform (SIFT) (Lowe, 1999) algorithm implemented in Fiji. Default parameters were used, except for: the number of steps per scale octave (increased to 4), the maximum alignment error (reduced to 5.00 pixels), and the transformation type (set to rigid with interpolation enabled). Between 7 and 10 sub-stacks were extracted per condition to build the training datasets: $512 \times 512 \times 20$ voxels for HeLa/Ctrl, $480 \times 480 \times 20$ for HeLa/STARD3, and $1024 \times 1024 \times 20$ for HeLa/STARD3 FYAA. In each sub-stack, LE/Lys, ER, and mitochondria were manually segmented using Fiji's built-in Segmentation Editor. These annotated sub-stacks were used in DeepSCEM to train a U-Net-based segmentation model for each condition (Meyer et al, 2025). Each trained model was then applied to the corresponding full image stack to produce segmentation prediction maps, resulting in total segmented volumes of 1674 µm³ for HeLa/Ctrl, 1034 µm³ for HeLa/STARD3, and 626 µm³ for HeLa/STARD3 FYAA. Full segmentation prediction maps were initially curated in Microscopy Image Browser (MIB, v2.83) (Belevich et al, 2016). Maps were then smoothed in Fiji: 2D mean filters (radius 2–5 pixels) for LE/Lys and mitochondria, and 3D mean filters for ER across all conditions. ER maps were further refined in ChimeraX (v1.9rc202411111853) (Pettersen et al, 2021). using the "hide dust" tool to remove features smaller than 50 nm in any dimension. For each condition, a representative region of interest (ROI) was selected and carefully curated in MIB, with volumes of 246 µm³ (HeLa/Ctrl), 147 µm³ (HeLa/STARD3), and 112 µm³ (HeLa/STARD3 FYAA). ER ROIs were filtered again in Fiji using a 3D mean filter (radius 2–5). Segmentation files were converted from TIFF to MRC format using tif2mrc (IMOD v4.11.24) (Kremer et al, 1996) for 3D rendering in ChimeraX. LE/Lys and mitochondria surfaces were smoothed (factor 1; 100, 70, and 50 iterations for HeLa/Ctrl, HeLa/STARD3, and HeLa/STARD3 FYAA, respectively), and ER surfaces were smoothed with factor 1 and 5, 6, and 5 iterations, respectively. 3D movies were generated in ChimeraX using a command script (.cxc).

## Protein extraction

### For Western blots

Cells were washed twice with 1x TBS and scraped in ice-cold loading buffer (50 mM Tris-HCl pH 6.8, 100 mM DTT, 2% SDS, 10% glycerol, 0.1% bromophenol blue). Protein extracts were sonicated at 4 °C three times for 5 s each, with 1 sec interval pulses, at 35% amplitude (Bioblock Scientific, Vibra-Cell 75041). Finally, protein extracts were boiled for 30 s before loading onto SDS-PAGE gels.

### For GFP-trap and immunoprecipitation assays

Cells were washed twice with 1x TBS and scraped in ice-cold IP Buffer (50 mM Tris pH 7.5, 50 mM NaCl, 1% Triton X-100, 1 mM EDTA, protease inhibitor tablets (cOmplete, Roche); phosphatase inhibitor tablets (PhosSTOP, Roche)) for GFP-Trap, or in M-PER buffer (78501, ThermoFisher Scientific) supplemented with cOmplete and PhosSTOP for immunoprecipitation. Cell extracts were incubated on ice for 20 min and then centrifuged at $9500 \times g$ for 10 min at 4 °C to remove debris. The supernatant was collected, and protein concentration was determined using the BCA protein assay (ThermoFisher Scientific).

## SDS-PAGE, Western blot and Coomassie blue staining

SDS-PAGE gels were composed of a stacking gel (4.5% acrylamide/bis-acrylamide (29/1), 0.375 M Tris-HCl pH 6.8, 0.1% SDS, 0.05% ammonium persulfate (APS), 0.01% tetramethyl ethylenediamine (TEMED)) and a separating gel (10% acrylamide/bis-acrylamide (29/1), 0.375 M Tris-HCl pH 8.8, 0.1% SDS, 0.05% APS, 0.01% TEMED). Protein migration was conducted in running buffer (0.02 M Tris-base, 0.1% SDS, 0.2 M glycine) at 120 V for 1 h 20.

For Coomassie blue staining, gels were stained with PageBlue Protein Staining Solution (Thermo Fisher Scientific) according to the manufacturer's instructions.

For Western Blots, proteins were electro-transferred onto nitrocellulose sheets (Whatman Protran nitrocellulose membranes, 0.45 µm) soaked in transfer buffer (0.04 M glycine, 0.04 M Tris-base, 0.035% SDS, 20% Ethanol) for 45 min at 240 mA (BIO-RAD, Tran-Blot SD Semi-Dry Transfer Cell). Membranes were blocked for 20 min in 1x PBS containing 3% non-fat dry milk and 0.1% Tween 20, or in 1x TBS–1% BSA for standard and for phospho-specific antibodies, respectively. Membranes were then incubated with primary antibodies overnight at 4 °C. The primary antibodies used were rabbit anti-GFP (1:1000; TP401, Torrey Pine Biolabs, RRID:AB_10013661), mouse anti-Lamp1 H4A3 (1:50; DSHB, RRID:AB_2296838), mouse anti-GSK3α/β (1/1000; Santa Cruz Biotechnology; 0011-A: sc-7291, RRID:AB_2279451), mouse anti-STARD3 (3G11; 1:100; IGBMC), and rabbit anti-pS$_{209}$ STARD3 (3144; 1:200; IGBMC), rabbit anti-Tip60 (1:1000; Cell Signaling Technology 12058, RRID:AB_2797811), rabbit anti-pS$_{86}$ Tip60 (1:1000; Abcam ab73207, RRID:AB_1523845), mouse anti-VAP-A (1:1000; 4C12, Santa Cruz Biotechnology, sc-293278), rabbit anti-VAP-B [1:1,000; kind gift from Dr. L. Dupuis (Kabashi et al, 2013)], mouse anti-MOSPD2 [1:7; 1MOS-4E10, (Di Mattia et al, 2018)], rabbit anti-GAPDH (1:1000; Sigma-Aldrich G9545, RRID:AB_796208). After 3 washes in 1x PBS-0.01% Tween or 1x TBS-0.01% Tween, the blots were incubated with appropriate secondary antibodies in the same buffers. Secondary antibodies were

peroxidase-conjugated AffiniPure goat anti-rabbit or goat anti-mouse (1:10,000; Jackson ImmunoResearch, 111-035-003 RRID:AB_2313567 and 115-035-003 RRID:AB_10015289, respectively). Protein-antibody complexes were visualized by chemiluminescence (Amersham ECL start Western Blotting detection reagent, GE Healthcare, or Super-Signal West Pico Plus Chemiluminescent Substrate, 34580 Thermo-Fischer Scientific) on an Imager 600 (Cytiva).

## Mass spectrometry

Phosphorylation at pS213 on the $_C$STD3 pS$_{213}$ recombinant protein was identified using nano-LC-MS/MS, as previously described (Di Mattia et al, 2020a).

## In vitro kinase assays

Five µg of $_C$STD3 were mixed with 50 ng of GSK3β S9A in kinase buffer (50 mM Tris-HCl pH 7.5, 10 mM MgCl$_2$, 0.1 mM EDTA, 100 µM ATP) and incubated at 30 °C during 30 min. The reaction was stopped by the addition of 6x SDS loading buffer (100 mM Tris-HCl pH 6.8, 4.5% SDS, 20% glycerol, 0.1% bromophenol blue, 3 M 2-mercapto-ethanol).

## GFP-trap and immunoprecipitation assays

For GFP-Trap assays, GFP-Trap Magnetic Agarose beads (ChromoTek) were washed with 1 mL of IP buffer. For each condition, 20 µL of beads were incubated with 500 µg protein extract for 2 h or overnight, at 4 °C under agitation. Beads were then washed twice with IP buffer, and bound proteins were eluted by adding 100 µL of 6x SDS loading buffer directly on the beads, followed by boiling.

For immunoprecipitation assays, Protein G agarose beads were washed with 1 mL TEN Buffer (Tris 50 mM pH 7.5, 1 mM EDTA, 100 mM NaCl), incubated with mouse anti-STARD3 (1STAR-2G5) in TEN Buffer (Di Mattia et al, 2018), and washed with IP buffer. The subsequent steps were performed as described for the GFP-Trap assays.

## Lipids

18:1-PC (1,2-dioleoyl-*sn*-glycero-3-phosphocholine or DOPC), 18:1-PS (1,2-dioleoyl-*sn*-glycero-3-phospho-L-serine or DOPS), 16:0-18:1-PS (1-palmitoyl-2-oleoyl-*sn*-glycero-3-phospho-L-serine or POPS), 18:1 NBD-PE (1,2-dioleoyl-*sn*-glycero-3-phosphoethanolamine-N-(7-nitro-21,3-benzoxadiazol-4-yl), 18:1–12:0 NBD-PA (1-oleoyl-2-{12-[(7-nitro-2-1,3-benzoxadiazol-4-yl)amino]dodecanoyl}-*sn*-glycero-3-phosphate), and 18:1 MPB-PE (1,2-dioleoyl-*sn*-glycero-3-phosphoethanolamine-N-[4-(p-maleimidophenyl)butyramide]) were purchased from Avanti Polar Lipids.

## Liposome preparation

In glass tubes, 1 µmole of lipids stored in CHCl$_3$ or CHCl$_3$/methanol stock solutions were mixed at the desired molar ratio. The tubes were pre-warmed at 33 °C for 5 min; then, the solvent was dried under a nitrogen flux for 25 min, and the tubes were placed in a vacuum chamber for 40 min to remove the remaining solvent. The lipid film was hydrated in 1 mL of 50 mM HEPES pH 7.4, 120 mM K-Acetate (HK) buffer to obtain a suspension of multilamellar vesicles. After thorough vortexing for 3 min, the multilamellar liposomes underwent a two-step extrusion process: first, 11 passages through a polycarbonate filter (Nuclepore Track-Etched Membrane; Whatman) with a pore diameter of 1 µm, followed by a second step of 11 passages through a polycarbonate filter with a pore diameter of 0.1 µm using a mini-extruder (Avanti Polar Lipids). Alternatively, the multilamellar vesicle suspension was frozen and thawed five times and then extruded through a polycarbonate filter of 0.2 µm pore size. For some experiments, the suspension of multilamellar vesicles underwent a one-step extrusion process, and was therefore immediately frozen and thawed five times before being extruded through a polycarbonate filter of 0.2 µm pore size. Liposomes were either used immediately or stored at 4 °C in the dark and used within 2 days.

## Flotation assays

The association of DTT-free $_C$STD3 constructs with membranes was measured by mixing the protein (0.75 µM) with liposomes of the desired composition, doped with 0.2 mol% NBD-PA (750 µM lipids) for 1 h at 25 °C under constant shaking (800 rpm). Next, in all cases, the liposome/protein mixture (final volume 150 µL) was adjusted to 28% (w/w) sucrose by mixing 100 µL of a 60% (w/w) sucrose solution in HK buffer and overlaid with 200 µL of HK buffer containing 24% (w/w) sucrose and 50 µL of sucrose-free HK buffer. The sample was centrifuged at 201,600 × g (average centrifuge force) in a swing rotor (TLS 55 Beckmann) for 70 min. The bottom (250 µL), middle (140 µL), and top (110 µL) fractions were collected. The bottom and top fractions were analyzed using SDS-PAGE after staining with SYPRO Orange using a FUSION FX fluorescence imaging system.

## Liposome pull down

Liposomes pull-down assays were performed as previously described in Kassas et al (2017). Briefly, NTA-Ni$^{2+}$ beads (Pure-Proteome Nickel Magnetic Beads, LSKMAGH10; Millipore) were washed with HK buffer. Then, 90 µg of recombinant proteins (START-$_{6HIS}$or MSP-VAPB$_{6HIS}$) was added to the beads and incubated for 20 min under agitation at 4 °C. To remove the excess of proteins, beads were washed twice with HK buffer and resuspended in 1 mL of HK buffer. Afterward, 50 µL of fluorescent liposomes were added to the beads and incubated for 20 min under agitation at 4 °C. Beads were then washed three times with HK buffer and resuspended in a final volume of 30 µL of HK buffer. For imaging, 10 µL of the suspension were dropped on a glass bottom dish (MatTek) and imaged on a spinning-disk CSU-X1 (Nikon; 100× NA 1.4).

For fluorescence quantification, beads of uniform size in the same focal plane were segmented, the background fluorescence subtracted, and the mean fluorescence intensity was measured.

## Aggregation assays

The experiments were performed at 25 °C using a Wyatt DynaPro-99-E-50 System (Protein Solutions). Liposomes (50 µM total lipids) composed of DOPC, DOPC/MPB-PE (97:3), DOPC/DOPS (70:30) or DOPC/DOPS/MPB-PE (67:30:3) were mixed in 20 µL of freshly degassed HK buffer and added to the quartz cell. A first set of 12 autocorrelation curves was acquired to measure the size

distribution of the initial liposome suspension. Then, a DTT-free sample of $_C$STD3 or $_C$STD3 KD/KD constructs (500 nM final concentration) was added manually and mixed thoroughly. For all the experiments, aggregation kinetics were measured by acquiring one autocorrelation curve every 10 s for 120 min. At the end of the experiment, a set of 12 autocorrelation functions was acquired. The data were analyzed using two different algorithms provided by the Dynamics v6.1 software. The autocorrelation functions were fitted during the kinetics, assuming the size distribution is a simple Gaussian function. This mode, called the monomodal or cumulant algorithm, gives a mean hydrodynamic radius, $R_H$, and width (or polydispersity). The polydispersity is represented in the kinetics measurements by a shaded area. It can reach tremendous values because of the simultaneous presence of free liposomes and liposome aggregates of various sizes. The autocorrelation functions were also fitted before and after the aggregation process using a regularization algorithm that can resolve several populations of different sizes, such as free liposomes and liposome aggregates.

## Ultra-centrifugation analysis

SV–AUC experiments were conducted in a ProteomeLab XL-I analytical ultracentrifuge (Beckman Coulter) at 20 °C. The samples were loaded into AUC cell assemblies with 12 mm charcoal-filled Epon double-sector centerpieces. The sample cells were loaded into an 8-hole An-50 Ti rotor for temperature equilibration for 2–3 h, followed by acceleration to full speed at $201,600 \times g$. Absorbance data at 280 nm were collected at 7-min intervals for 16 h. The partial specific volumes of the proteins, buffer density and viscosity were calculated using the software SEDNTERP. Sedimentation data were time-corrected and modeled with diffusion-deconvoluted sedimentation coefficient distributions c(s) in SEDFIT 16.1c, with signal-average frictional ratio and meniscus position refined with nonlinear regression (Schuck, 2000). Sedimentation coefficient distributions were corrected to standard conditions (water at 20 °C, s20,W). The plot was created in GUSSI (Brautigam, 2015). We used the software Hullrad (Fleming and Fleming, 2018) to calculate the sedimentation coefficient from the monomeric STARD3 structural model.

## Statistical analyses

Statistical analyses were performed using the One-way ANOVA or Student's t-test parametric tests (Prism9, GraphPad). Conditions were compared with the Dunnett's or Tukey multiple comparisons tests. *P*-values < 0.05, <0.01, <0.001, and <0.0001 are identified with 1, 2, 3, and 4 asterisks, respectively. ns: $p \geq 0.05$.

# Data availability

FIB-SEM data have been deposited in EMPIAR (Iudin et al, 2023) under accession number EMPIAR-13156. https://doi.org/10.6019/EMPIAR-13156.

The source data of this paper are collected in the following database record: biostudies:S-SCDT-10_1038-S44318-026-00705-3.

# Peer review information

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

## Acknowledgements

We thank the members of the Molecular and Cellular Biology of Breast Cancer team (IGBMC) and Céline Charvet for helpful advice and discussions. J. Eichler received fellowships from EUR IMCBio funds and the Ligue Nationale contre le Cancer. M. Zouiouich received fellowships from ITMO Cancer AVIESAN (Alliance Nationale pour les Sciences de la Vie et de la Santé/National Alliance for Life Sciences and Health) within the framework of the Cancer Plan (https://itcancer.aviesan.fr/) and from Fondation ARC pour la recherche sur le cancer (ARCDOC42022010004428). S. Huver received a fellowship from the Ligue Nationale contre le Cancer. This work was supported by grants from the Agence Nationale de la Recherche (grant CONSIGN ANR-25-CE13-4997, SPHINGODYNA ANR-22-CE44-0019), and from the Ligue Contre le Cancer (Conférence de Coordination Interrégionale du Grand Est; https://www.ligue-cancer.net). We are grateful to the members of the IGBMC Imaging Center, especially Bertrand Vernay, Elvire Guiot and Erwan Grandgirard for their help. We acknowledge the IGBMC imaging center, member of the national infrastructure France-BioImaging supported by the French National Research Agency (ANR-10-INBS-04). We thank the IGBMC cell culture facility, the peptide synthesis facility (Pascal Eberling), the Flow Cytometry facility (Claudine Ebel and Muriel Philipps), Molecular Biology and Virus Service (Paola Rossolillo and Nicole Jung), the Integrated Structural Biology platform (Pierre Poussin-Courmontagne), the Mediaprep facility (Denis Fumagalli). We thank Danièle Spehner from IGBMC, and Paolo Ronchi and Yannick Schwab from the Electron Microscopy Core Facility at the European Molecular Biology Laboratory Heidelberg for their help with electron microscopy. This work of the Interdisciplinary Thematic Institute IMCBio+, as part of the ITI 2021-2028 program of the University of Strasbourg, CNRS and Inserm, was supported by IdEx Unistra (ANR-10-IDEX-0002), and by SFRI-STRAT'US project (ANR-20-SFRI-0012) and EUR IMCBio (ANR-17-EURE-0023) under the framework of the France 2030 Program. This work used the Integrated Structural Biology platform of the Strasbourg Instruct-ERIC center IGBMC-CBI supported by FRISBI (ANR-10-INBS-0005).

## Author contributions

**Julie Eichler**: Conceptualization; Data curation; Software; Formal analysis; Funding acquisition; Validation; Investigation; Visualization; Methodology; Writing—original draft; Writing—review and editing. **Corinne Wendling**: Data curation; Formal analysis; Investigation; Visualization; Methodology; Writing—original draft; Writing—review and editing. **Sophie Huver**: Data curation; Formal analysis; Investigation; Methodology; Writing—review and editing. **Mehdi Zouiouich**: Data curation; Software; Formal analysis; Validation; Investigation; Visualization; Writing—review and editing. **Victor Hanss**: Data curation; Software; Investigation; Visualization; Methodology; Writing—original draft; Writing—review and editing. **Anna Cardinal**: Data curation; Formal analysis; Investigation; Methodology; Writing—original draft; Writing—review and editing. **Victoria Fimbel**: Formal analysis; Validation; Investigation; Visualization; Methodology. **Catherine Birck**: Formal analysis; Investigation; Visualization; Methodology; Writing—original draft; Writing—review and editing. **Alastair G McEwen**: Conceptualization; Software; Formal analysis; Investigation; Writing—review and editing. **Céline Knorr**: Formal analysis; Investigation; Methodology; Writing—review and editing. **Catherine Fromental-Ramain**: Formal analysis; Investigation; Writing—review and editing. **Maxime Boutry**: Formal analysis; Investigation; Writing—review and editing. **Marie-Pierre Chenard**: Resources; Writing—review and editing. **Guillaume Drin**: Conceptualization; Data curation; Formal analysis; Supervision; Funding acquisition; Validation; Investigation; Visualization; Methodology; Writing—original draft; Project administration; Writing—review and editing. **Catherine Tomasetto**: Conceptualization; Data curation; Formal analysis; Supervision; Funding acquisition; Validation; Investigation; Visualization; Methodology; Writing—original draft; Project administration; Writing—review and editing. **Fabien Alpy**: Conceptualization; Resources; Data curation; Software; Formal analysis; Supervision; Funding acquisition; Validation; Investigation; Visualization; Methodology; Writing—original draft; Project administration; Writing—review and editing.

Source data underlying figure panels in this paper may have individual authorship assigned. Where available, figure panel/source data authorship is

*Julie Eichler et al*                                                                          *The EMBO Journal*

listed in the following database record: biostudies:S-SCDT-10_1038-S44318-026-00705-3.

## Disclosure and competing interests statement

The authors declare no competing interests.

ns licence, unless indicated otherwise in a credit line to the material. If material is not included in the article's Creative Commons licence and your intended use is not permitted by statutory regulation or exceeds the permitted use, you will need to obtain permission directly from the copyright holder. To view a copy of this licence, visit http://creativecommons.org/licenses/by/4.0/. Creative Commons Public Domain Dedication waiver http://creativecommons.org/public-domain/zero/1.0/ applies to the data associated with this article, unless otherwise stated in a credit line to the data, but does not extend to the graphical or creative elements of illustrations, charts, or figures. This waiver removes legal barriers to the re-use and mining of research data. According to standard scholarly practice, it is recommended to provide appropriate citation and attribution whenever technically possible.

© The Author(s) 2026

© The Author(s)                                              *The EMBO Journal* Volume 45 | Issue 7 | April 2026 | 2239 – 2277    **2269**

# Expanded View Figures

**Figure EV1.   Dose and time-dependent effects of GSK3 inhibition on STARD3 phosphorylation.**

(A) Western blot analysis of MCF7 cells overexpressing TIP60 treated or not with the GSK3 inhibitor CHIR99021 (5 μM; overnight). TIP60 protein levels (Total) and $S_{586}$ phosphorylation ($pS_{586}$) were analyzed. (B) Western blot analysis of HeLa/STARD3 cells treated overnight with different concentration of CHIR99021 (1, 2 and 5 μM) or left untreated. STARD3 protein level (Total) and $S_{209}$ phosphorylation ($pS_{209}$) were analyzed. (C) (a) Western blot analysis of HCC1954 cells treated with 5 μM of CHIR99021 for 2, 4, 6, 8 and 16 h, or left untreated. STARD3 protein levels (Total) and $S_{209}$ phosphorylation ($pS_{209}$) were analyzed. (b) Quantification of relative $S_{209}$ phosphorylation levels. Means ± SD. One-way ANOVA with Dunnett's multiple comparison test (*, $P < 0.05$; ***, $P < 0.001$; $n = 3$ independent experiments; 0 h vs 2 h, $P = 0.33$; 0 h vs 4 h, $P = 0.06$; 0 h vs 6 h, $P = 2.3 \times 10^{-2}$; 0 h vs 8 h, $P = 1.9 \times 10^{-2}$; 0 h vs 16 h, $P = 8 \times 10^{-4}$). (c) Representative images of HCC1954 cells expressing WT STARD3 and treated with the GSK3 inhibitor CHIR99021 for 2, 4, 6, 8 and 16 h, or left untreated. Cells were labeled with an anti-LAMP1 antibody (magenta), an anti-STARD3 antibody (green) and Hoechst for nuclei (blue). (D, E) Quantification of relative GSK3α and GSK3β in HCC1954 (D) and MCF7/STARD3 (E) cells transfected with control siRNAs (siCtrl) or siRNAs targeting GSK3α (siGSK3α), GSK3β (siGSK3β), or both (siGSK3α + siGSK3β) (see Fig. 1E,F). Means ± SD. One-way ANOVA with Dunnett's multiple comparison test (*, $P < 0.05$; **, $P < 0.01$; ****, $P < 0.0001$, $n = 3$-4 independent experiments; D, GSK3α: siCtrl vs WT, $P = 0.76$; GSK3α, $P = 9 \times 10^{-3}$; siCtrl vs siGSK3β, $P = 0.52$; siCtrl vs siGSK3α + siGSK3β, $P = 1.1 \times 10^{-2}$; (D), GSK3 β: siCtrl vs WT, $P = 0.5$; GSK3α, $P = 0.99$; siCtrl vs siGSK3β, $P = 3.6 \times 10^{-3}$; siCtrl vs siGSK3α + siGSK3β, $P = 1.6 \times 10^{-3}$; (E), GSK3α: siCtrl vs WT, $P = 0.94$; GSK3α, $P = 1.3 \times 10^{-2}$; siCtrl vs siGSK3β, $P = 0.97$; siCtrl vs siGSK3α + siGSK3β, $P = 8 \times 10^{-3}$; (E), GSK3 β: siCtrl vs WT, $P = 0.93$; GSK3α, $P = 0.28$; siCtrl vs siGSK3β, $P < 10^{-4}$; siCtrl vs siGSK3α + siGSK3β, $P < 10^{-4}$). Source data are available online for this figure.

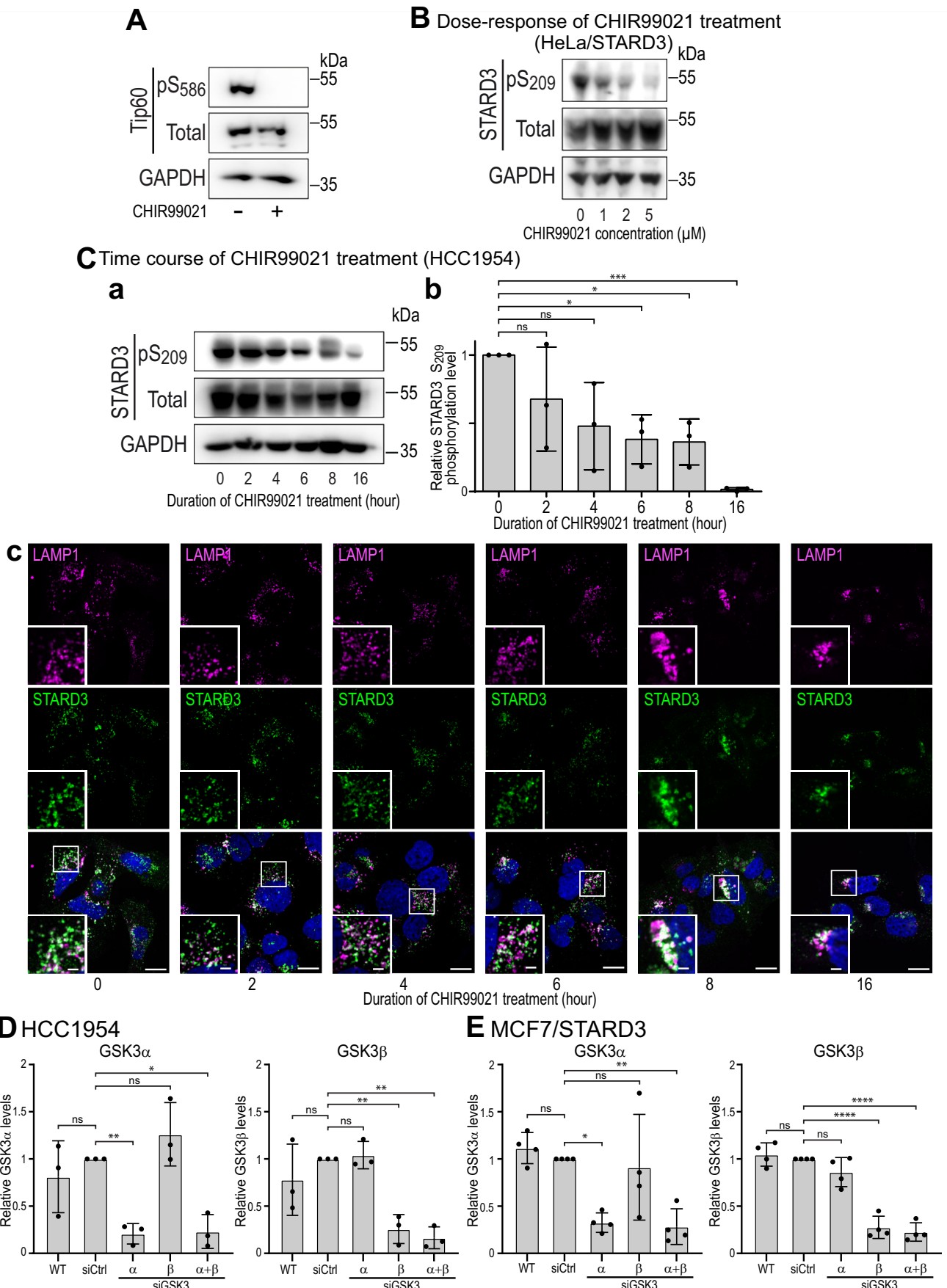

**A**

Tip60 | pS586
Total

GAPDH

CHIR99021 − +

kDa
—55
—55
—35

**B** Dose-response of CHIR99021 treatment (HeLa/STARD3)

STARD3 | pS209
Total

GAPDH

CHIR99021 concentration (μM)
0 1 2 5

kDa
—55
—55
—35

**C** Time course of CHIR99021 treatment (HCC1954)

**a**

STARD3 | pS209
Total

GAPDH

Duration of CHIR99021 treatment (hour)
0 2 4 6 8 16

kDa
—55
—55
—35

**b**

Relative STARD3 S209 phosphorylation level

ns * * ***

Duration of CHIR99021 treatment (hour)
0 2 4 6 8 16

**c**

LAMP1   LAMP1   LAMP1   LAMP1   LAMP1   LAMP1

STARD3   STARD3   STARD3   STARD3   STARD3   STARD3

Duration of CHIR99021 treatment (hour)
0   2   4   6   8   16

**D** HCC1954

GSK3α

Relative GSK3α levels

ns ns ** * 

WT siCtrl α β α+β
siGSK3

GSK3β

Relative GSK3β levels

ns ns ** ** 

WT siCtrl α β α+β
siGSK3

**E** MCF7/STARD3

GSK3α

Relative GSK3α levels

ns ns * ** 

WT siCtrl α β α+β
siGSK3

GSK3β

Relative GSK3β levels

ns ns **** **** 

WT siCtrl α β α+β
siGSK3

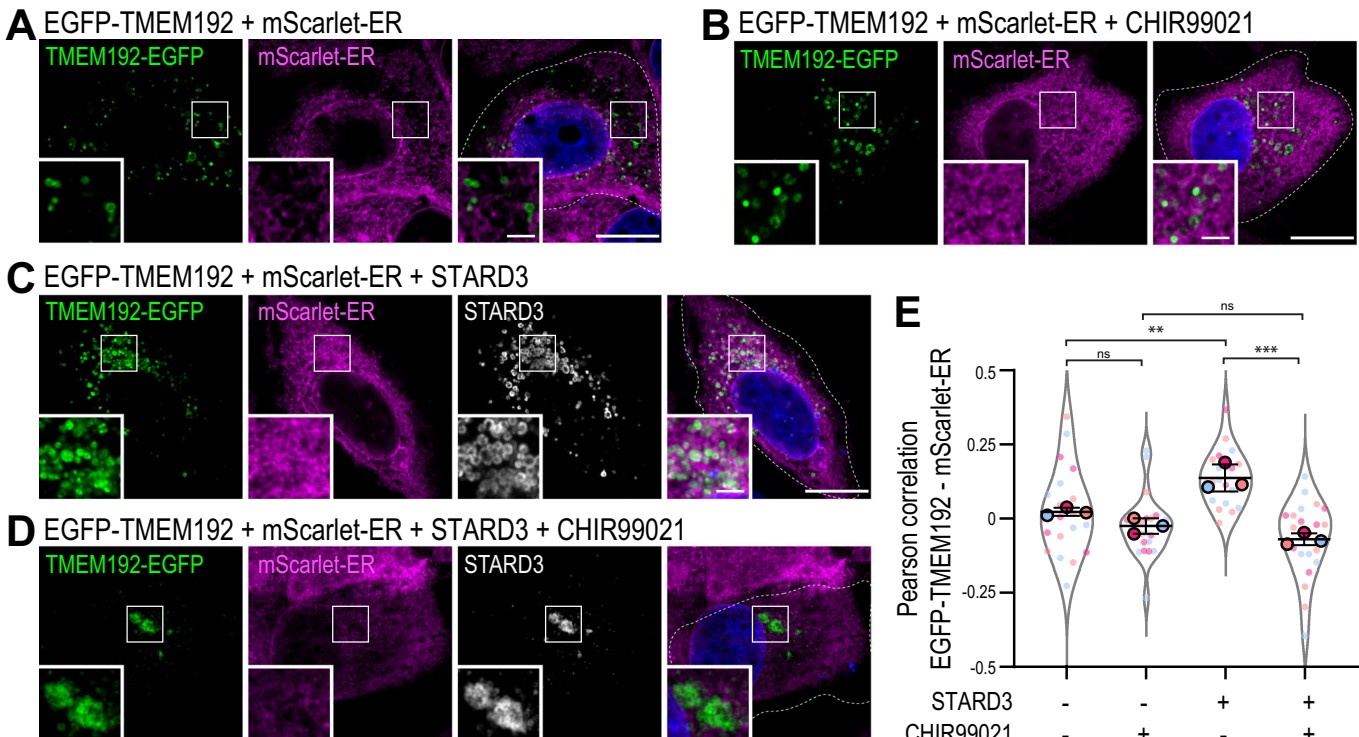

**Figure EV2.  In vivo, GSK3 activity governs the establishment of STARD3-mediated ER–endosome contacts.**

(**A–D**) MCF7 cells expressing mScarlet-ER (magenta) were transfected with EGFP-TMEM192 (green) (**A–D**) and STARD3 WT (**C, D**). Cells were left untreated (**A, C**) or treated with CHIR99021 (**B, D**). STARD3 was labeled using anti-STARD3 antibodies (gray), and nuclei were stained with Hoechst (blue). Insets show higher magnification images of the areas outlined in white. Scale bars: 10 µm. Inset scale bars: 2 µm. Overlay panels show merged green, magenta and blue channels. In (**A, B**), endogenous STARD3 levels were too low to be detected with anti-STARD3 antibodies. (**E**) Pearson's correlation coefficients between EGFP-TMEM192 and mScarlet-ER in cells with or without STARD3 expression and with or without CHIR99021 treatment. Data are displayed as Superplots showing the clustering index per cell (small dots) and its mean per independent experiment (large dots). Number of cells: mScarlet-ER / EGFP-TMEM192: 21; mScarlet-ER / EGFP-TMEM192 treated with CHIR99021: 22; mScarlet-ER / EGFP-TMEM192 / STARD3: 22, mScarlet-ER / EGFP-TMEM192 / STARD3 treated with CHIR99021: 22, from three independent experiments). Means and error bars (SD) are shown. ANOVA with Tukey's multiple comparison test (\*\*, $P < 0.01$; \*\*\*, $P < 0.001$; mScarlet-ER / EGFP-TMEM192 vs mScarlet-ER / EGFP-TMEM192-CHIR99021: $P = 0.26$; mScarlet-ER / EGFP-TMEM192 vs mScarlet-ER / EGFP-TMEM192 / STARD3: $P = 6 \times 10^{-3}$; mScarlet-ER / EGFP-TMEM192-CHIR99021 vs mScarlet-ER / EGFP-TMEM192 / STARD3-CHIR99021: $P = 0.31$; mScarlet-ER / EGFP-TMEM192 / STARD3 vs mScarlet-ER / EGFP-TMEM192 / STARD3-CHIR99021: $P = 1.9 \times 10^{-4}$). Source data are available online for this figure.

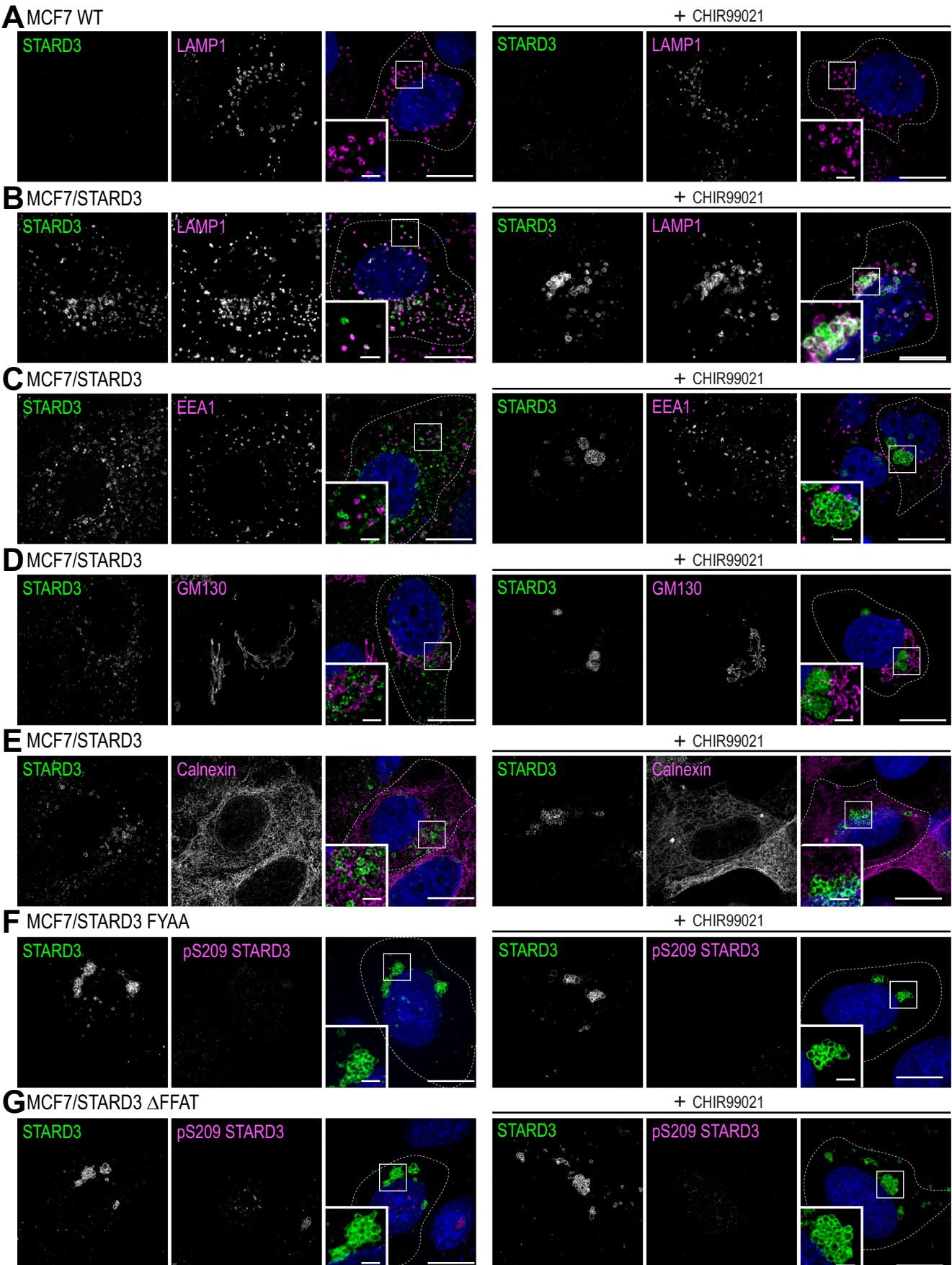

◀ **Figure EV3.  STARD3 mediates the formation of LE/Lys clusters but does not affect other organelles.**

WT MCF7 cells (**A**), MCF7 cells expressing STARD3 WT (**B**–**F**) or FFAT-motif deficient mutants (STARD3 FYAA (**F**), STARD3 ΔFFAT (**G**)) were left untreated (left) or treated with CHIR99021 (5 μM, overnight; right). Cells were labeled with anti-STARD3 antibodies (green) (**A**–**G**) and in magenta with: anti-LAMP1 antibodies (**A**, **B**) to label LE/Lys, anti-EEA1 antibodies (**C**) to label early endosomes, anti-GM130 antibodies (**D**) to label the Golgi apparatus, anti-calnexin antibodies (**E**) to label the ER or phospho-specific antibodies (pS$_{209}$ STARD3, **F**, **G**). Nuclei were stained with Hoechst (blue). Subpanels show higher magnification images of the area outlined in white. Scale bars: 10 μm. Inset scale bars: 2 μm. Source data are available online for this figure.

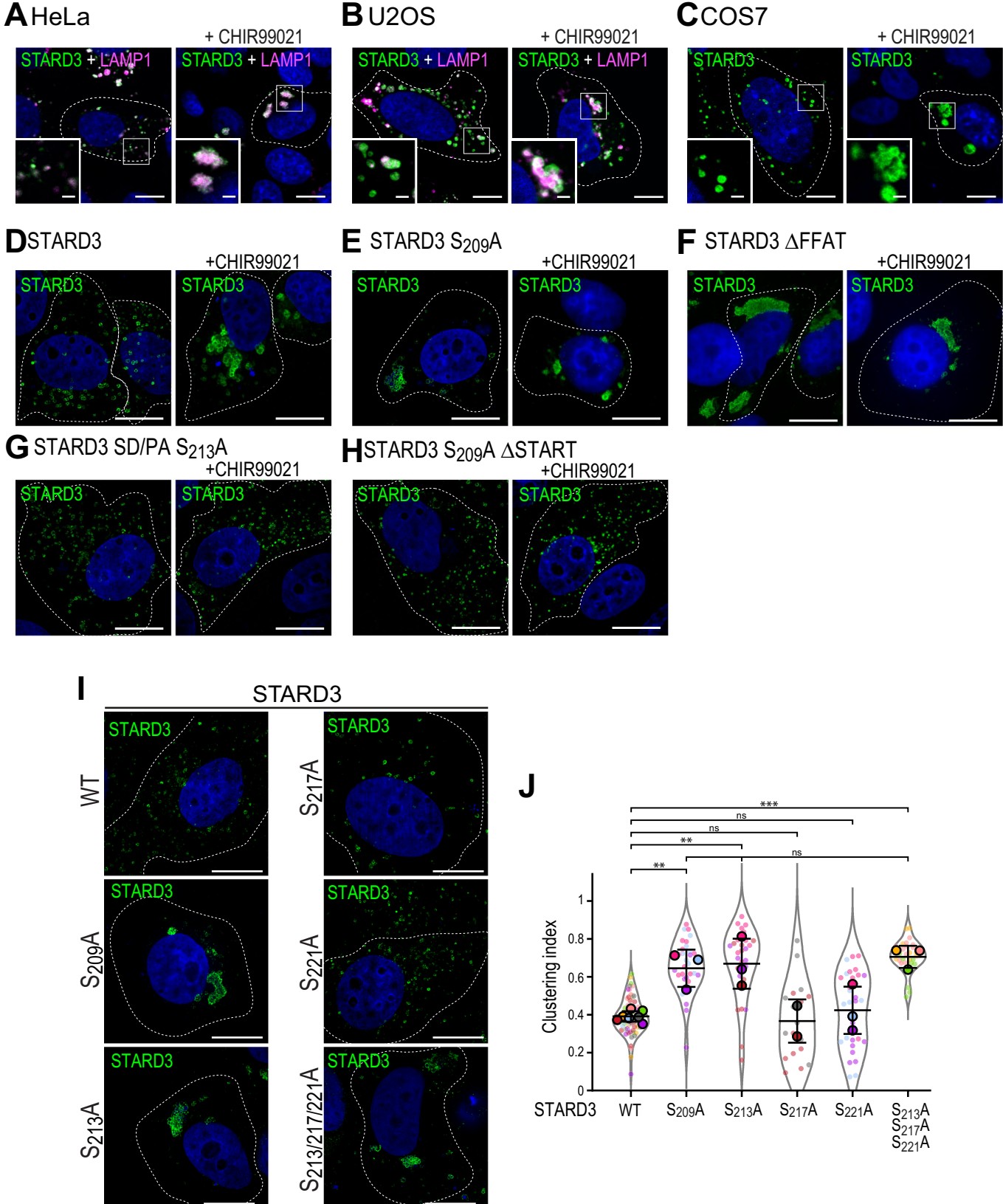

**Figure EV4.  Characterization of STARD3-induced LE/Lys clusters.**

(A–C) HeLa cells (A), U2OS cells (B) and COS-7 cells (C) expressing STARD3 WT were left untreated (left) or treated with CHIR99021 (5 µM, overnight) (right). Cells were labeled with anti-STARD3 antibodies (green) and anti-LAMP1 antibody (magenta). Nuclei were stained with Hoechst (blue). Subpanels show higher magnification images of the area outlined in white. Images were acquired with a SP8-UV confocal microscope. Scale bars: 10 µm. Inset scale bars: 2 µm. (D–H) MCF7 (A) expressing STARD3 WT (D), STARD3 $S_{209}$A (E), STARD3 ΔFFAT (F), STARD3 SD/PA $S_{213}$A (G) or STARD3 $S_{209}$A ΔSTART (H) were left untreated (left) or treated with CHIR99021 (5 µM, overnight; right). Cells were labeled with anti-STARD3 antibodies (green). Nuclei were stained with Hoechst (blue). Scale bars: 10 µm. Inset scale bars: 2 µm. (I, J) Representative images of MCF7 cells expressing STARD3 WT, STARD3 $S_{209}$A, STARD3 $S_{213}$A, STARD3 $S_{217}$A, STARD3 $S_{221}$A, or STARD3 $S_{213}$A-$S_{217}$A-$S_{221}$A. Cells were labeled with anti-STARD3 antibodies (green) and Hoechst (blue). Scale bars: 10 µM. (E) Quantification of LE/Lys clustering in cells shown in (D). Data are displayed as Superplots showing the clustering index per cell (small dots) and its mean per independent experiment (large dots). Number of cells: MCF7-STARD3: 70, MCF7-STARD3 $S_{209}$A: 31, MCF7-STARD3 $S_{213}$A: 31, MCF7-STARD3 $S_{217}$A: 18, MCF7-STARD3 $S_{221}$A: 32, MCF7-STARD3 $S_{213}$A-$S_{217}$A-$S_{221}$A: 31, from seven independent experiments). Independent experiments are color-coded. Means and error bars (SD) are shown as black bars. One-way ANOVA with Tukey's multiple comparison test (**, $P < 0.01$; ***, $P < 0.001$; $n = 3$–7 independent experiments; WT vs $S_{209}$A, $P = 6.8 \times 10^{-3}$; WT vs $S_{213}$A, $P = 3 \times 10^{-3}$; WT vs $S_{217}$A, $P = 0.99$; WT vs $S_{221}$A, $P = 0.99$; WT vs $S_{213}$A-$S_{217}$A-$S_{221}$A, $P = 10^{-3}$; $S_{209}$A vs $S_{213}$A, $P = 0.99$; $S_{209}$A vs $S_{213}$A-$S_{217}$A-$S_{221}$A, $P = 0.95$; $S_{213}$A vs $S_{213}$A-$S_{217}$A-$S_{221}$A, $P = 0.99$). Source data are available online for this figure.

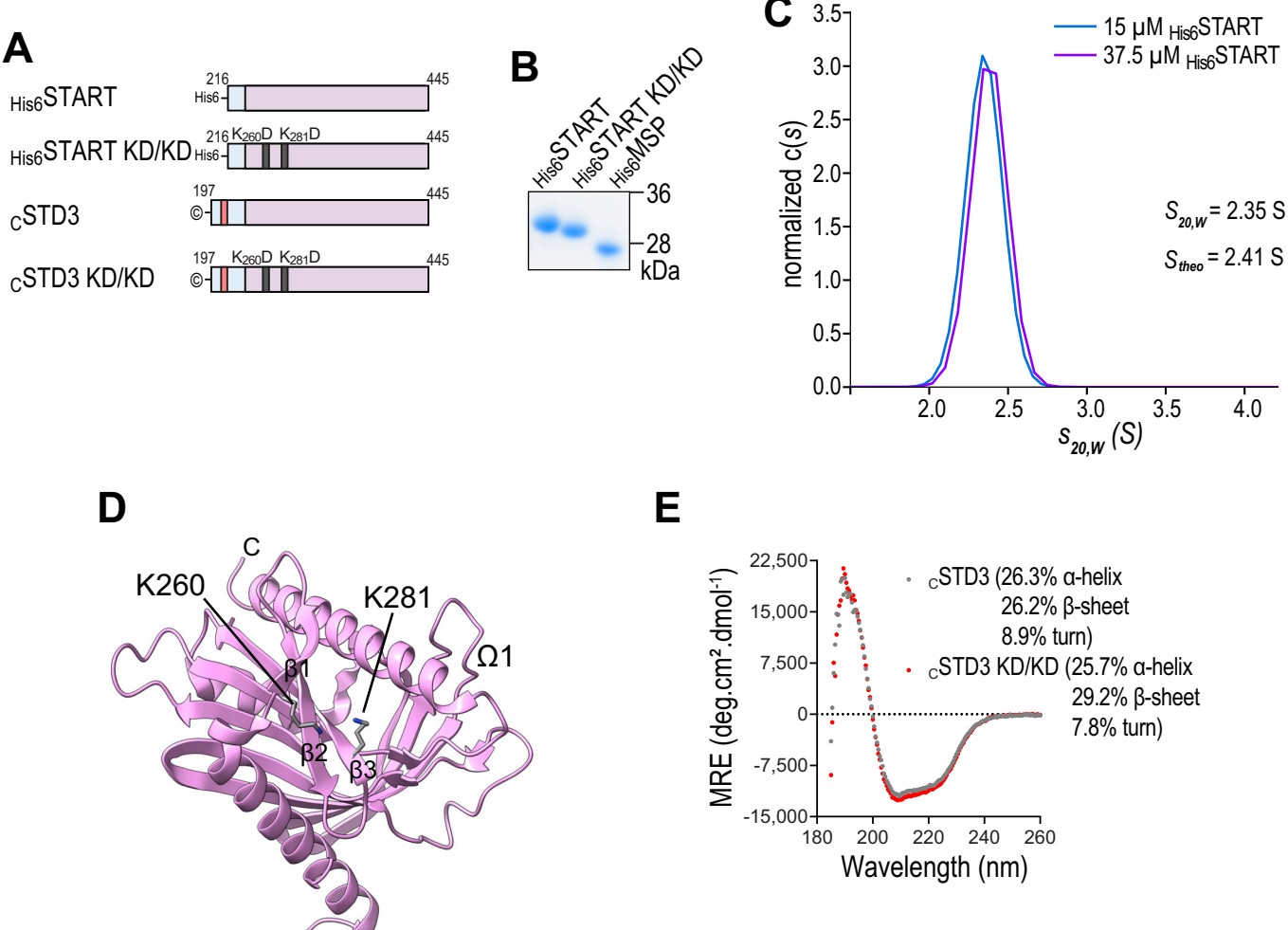

**Figure EV5. The START domain is a monomer in solution and contains positively charged surface patches.**

(A) Schematic representation of the different recombinant proteins used. (B) Coomassie blue staining of the recombinant START domain of STARD3 (WT and KD/KD mutant) and the MSP domain of VAP-B proteins after SDS-PAGE. (C) Superimposition of sedimentation coefficient distributions c(s) obtained from sedimentation velocity (SV) experiments with either 15 µM (blue) or 37.5 µM (purple) of the recombinant START domain of STARD3. The experimental average sedimentation coefficient ($S_{20,w}$) of 2.35 S is close to the theoretical value ($S_{theo}$) of 2.41 S, indicating that the protein remains in a stable monomeric form at both concentrations. (D) Ribbon diagram of the START domain of STARD3. The positions of the two mutated residues, K260 and K281, are highlighted. Key structural features, including omega loop 1 (Ω1) and three beta strands (β1, β2, β3), are also indicated (PDB ID: 1EM2) (Tsujishita and Hurley, 2000). (E) Far-UV CD spectrum of purified $_C$STD3 and $_C$STD3 KD/KD constructs in 20 mM Tris pH 7.4, 120 mM NaF buffer at room temperature. Source data are available online for this figure.

