## [Peer Review File · The EMBO Journal]

STARD3 regulates lysosome positioning and contacts via a GSK3-controlled phosphorylation switch

Julie Eichler, Corinne Wendling, Sophie Huver, Mehdi Zouiouich, Victor Hanss, Anna Cardinal, Victoria Fimbel, Catherine Birck, Alastair McEwen, Céline Knorr, Catherine Fromental-Ramain, Maxime Boutry, Marie-Pierre Chenard, Guillaume Drin, Catherine Tomasetto, and Fabien Alpy

Corresponding authors: Fabien Alpy (Fabien.Alpy@igbmc.fr) , Catherine Tomasetto (cat@igbmc.fr)

Review Timeline:

Submission Date:	13th Jun 25
Editorial Decision:	21st Jul 25
Revision Received:	20th Oct 25
Editorial Decision:	4th Dec 25
Revision Received:	14th Dec 25
Accepted:	8th Jan 26

Editor: William Teale

Transaction Report:

Dear Dr. Alpy,

Thank you again for the submission of your manuscript entitled "STARD3 regulates ER-endosome contacts and positioning via a GSK3-controlled phosphorylation switch" and for your patience during the review process. Your manuscript was sent to two referees; we have now received the reports from both of them, which I copy below.

As you can see from their comments, both referees considered the study to be well-performed and timely. That said, they point out some issues that will require your attention before your manuscript can be published in The EMBO Journal.

Based on the overall interest expressed in the reports, I would like to invite you to address the comments of all referees in a revised version of the manuscript. I should add that it is The EMBO Journal policy to allow only a single major round of revision and that it is therefore important to resolve the main concerns at this stage. I believe the concerns of the referees are reasonable and addressable, but please contact me if you have any questions, need further input on the referee comments or if you anticipate any problems in addressing any of their points. I am available to discuss the reports over Zoom at any time. Please, follow the instructions below when preparing your manuscript for resubmission.

I would also like to point out that as a matter of policy, competing manuscripts published during this period will not be taken into consideration in our assessment of the novelty presented by your study ("scooping" protection). We have extended this 'scooping protection policy' beyond the usual 3 month revision timeline to cover the period required for a full revision to address the essential experimental issues. Please contact me if you see a paper with related content published elsewhere to discuss the appropriate course of action.

Again, please contact me at any time during revision if you need any help or have further questions.

Thank you very much again for the opportunity to consider your work for publication. I look forward to your revision.

Best regards,

William

William Teale, Ph.D.
Editor
The EMBO Journal

When submitting your revised manuscript, please carefully review the instructions below and include the following items:

- 1) a .docx formatted version of the manuscript text (including legends for main figures, EV figures and tables). Please make sure that the changes are highlighted to be clearly visible.
- 2) individual production quality figure files as .eps, .tif, .jpg (one file per figure).
- 3) a .docx formatted letter INCLUDING the reviewers' reports and your detailed point-by-point response to their comments. As part of the EMBO Press transparent editorial process, the point-by-point response is part of the Review Process File (RPF), which will be published alongside your paper.
- 4) a complete author checklist, which you can download from our author guidelines ([https://wol-prod-cdn.literatumonline.com/pb-assets/embo-site/Author Checklist%20-%20EMBO%20J-1561436015657.xlsx](https://wol-prod-cdn.literatumonline.com/pb-assets/embo-site/Author%20Checklist%20-%20EMBO%20J-1561436015657.xlsx)). Please insert information in the checklist that is also reflected in the manuscript. The completed author checklist will also be part of the RPF.
- 5) Please note that all corresponding authors are required to supply an ORCID ID for their name upon submission of a revised manuscript.
- 6) We require a 'Data Availability' section after the Materials and Methods. Before submitting your revision, primary datasets produced in this study need to be deposited in an appropriate public database, and the accession numbers and database listed under 'Data Availability'. Please remember to provide a reviewer password if the datasets are not yet public (see <https://www.embopress.org/page/journal/14602075/authorguide#datadeposition>). If no data deposition in external databases is

needed for this paper, please then state in this section: This study includes no data deposited in external repositories. Note that the Data Availability Section is restricted to new primary data that are part of this study.

Note - All links should resolve to a page where the data can be accessed.

8) For data quantification: please specify the name of the statistical test used to generate error bars and P values, the number (n) of independent experiments (specify technical or biological replicates) underlying each data point and the test used to calculate p-values in each figure legend. The figure legends should contain a basic description of n, P and the test applied. Graphs must include a description of the bars and the error bars (s.d., s.e.m.).

9) We would also encourage you to include the source data for figure panels that show essential data. Numerical data can be provided as individual .xls or .csv files (including a tab describing the data). For 'blots' or microscopy, uncropped images should be submitted (using a zip archive or a single pdf per main figure if multiple images need to be supplied for one panel). Additional information on source data and instruction on how to label the files are available at .

10) We replaced Supplementary Information with Expanded View (EV) Figures and Tables that are collapsible/expandable online (see examples in <https://www.embopress.org/doi/10.15252/embj.201695874>). A maximum of 5 EV Figures can be typeset. EV Figures should be cited as 'Figure EV1, Figure EV2" etc. in the text and their respective legends should be included in the main text after the legends of regular figures.

12) Our journal encourages inclusion of *data citations in the reference list* to directly cite datasets that were re-used and obtained from public databases. Data citations in the article text are distinct from normal bibliographical citations and should directly link to the database records from which the data can be accessed. In the main text, data citations are formatted as follows: "Data ref: Smith et al, 2001" or "Data ref: NCBI Sequence Read Archive PRJNA342805, 2017". In the Reference list, data citations must be labeled with "[DATASET]". A data reference must provide the database name, accession number/identifiers and a resolvable link to the landing page from which the data can be accessed at the end of the reference. Further instructions are available at .

13) In order to increase the reproducibility and reach of your work, The EMBO Journal includes a table of reagents that were used in the study. Please provide this along with your revisions.

Further instructions for preparing your revised manuscript:

We realize that it is difficult to revise to a specific deadline. In the interest of protecting the conceptual advance provided by the work, we recommend a revision within 3 months (19th Oct 2025). Please discuss the revision progress ahead of this time with the editor if you require more time to complete the revisions.

Referee #1:

Reviewer Comments

Eichler et al. have elucidated the molecular mechanism by which the intracellular localization of late endosomes and lysosomes is regulated through the activity of a tethering protein. Specifically, they demonstrate that phosphorylation of STARD3 at Ser209 by GSK3 α and GSK3 β activates STARD3-mediated contacts between the endoplasmic reticulum (ER) and late endosomes/lysosomes. This finding indicates that such phosphorylation events control endosomal localization within cells and sheds light on previously uncharacterized molecular mechanisms involving endosomal dynamics. Conventionally, the localization of these organelles was believed to be regulated primarily via motor proteins such as kinesin and dynein, and their interactions with microtubules. This study highlights an alternative mechanism, thus representing a significant advance in the field. Below, I provide several points that I believe require revision.

Major Points

1. Inconsistency in cell lines used across experiments:

The authors use various cell lines in different figures. For example, HCC1954 and MCF7 cells are used in Figure 1 (phosphorylation of STARD3 by GSK3 β), MCF7 in Figures 2, 5, 6, and 8, and HeLa cells in Figure 3 (immunoprecipitation and immunostaining experiments on the STARD3-VAP interaction) as well as in Figure S6 (electron microscopy). While the main figures predominantly use MCF7 cells, the rationale for employing HeLa cells specifically in the experiments shown in Figure 3 and Figure S6 is unclear. The authors should provide a justification for this choice.

2. Panel J in Figure 1:

In contrast to Panel I, which shows selective knockdown of GSK3 β in HCC1954 cells, Panel J shows a marked decrease in both GSK3 β and GSK3 α expression levels, even with GSK3 β knockdown alone, suggesting that knockdown of both isoforms may have occurred. However, the corresponding bar graph beneath this panel shows no substantial difference in STARD3 S209 phosphorylation levels between GSK3 β and GSK3 α knockdowns, while an additive effect is observed in the double knockdown. This apparent discrepancy requires clarification.

Minor Points

1. Schematic illustration in Figure 9:

The large oval-shaped structure enclosed within the cell appears to represent the nucleus, but this is not explicitly indicated in the figure. To avoid any potential confusion, the authors should clearly label or annotate this structure.

2. Electron microscopy data in Figure S6:

As endosomes and lysosomes can be identified by electron microscopy, it may be possible to perform a more quantitative analysis (e.g., quantifying the number of membrane contact sites per endosome per cell). The authors are encouraged to consider whether such an analysis could be incorporated to support the qualitative observations.

Referee #2:

This paper from Eichler and colleagues dissects an elegant mechanism of regulation of ER-LE/Lysosome tethering, linked with LE/Lysosomal clustering. Using a combination of in vitro and cellular work in multiple different mammalian cell lines (including kinase inhibitors and silencing/mutational studies) the work identifies GSK3 as the kinase which phosphorylates the lysosomal protein STARD3 on a specific serine residue in its FFAT motif. This phosphorylation then regulates interaction with the ER tethering mediators VAPA/B/MOSPD2 to control connections between the ER and LE/Lysosomes. A similar regulatory mechanism has already been shown for mitochondria-ER and peroxisome-ER contacts and so this fits with an emerging theme of phosphorylation of FFAT motif regulation and places GSK3 beta at the centre of this. The authors then observed that uncoupling from VAP appears to allow the lipid binding START domain of STARD3 to then instead induce lysosomal clustering, potentially by a direct membrane binding mechanism. Based on this they propose an overall mechanism (Fig 9) that GSK3 activity either allows ER connection and the even distribution of LE/Lysos in the cell or loss of ER connection, resulting in clustering at the perinuclear region.

The study is exceptionally well performed, with multiple control experiments, different cell lines utilised, complementary in vitro and cellular work and mostly very clear and convincing data.

A weakness would be, as no clear function is attributed to this clustering/evenly distributed switch, the purpose of this mechanism and where it is applied is unclear. It could be considered that this reduces the importance of the study for a wider readership. However, from a cell-biology perspective, this is an important study which contributes to our understanding of a current hot topic in the field, regulation of membrane contact sites.

Overall, the vast majority of the data is highly convincing, and despite the lack of a clear function for the switching mechanism this study clearly warrants publication.

I have just a few minor suggestions for improvement:

Line 138-139 - specify which cells were used

Figure 1J - MCF7 GSK3 silencing - I find this a little unconvincing, whilst the data for the HCC1954 cells is clear - do the authors have a blot which is not over-exposed for GAPDH and STARD3 so the reader can judge the change for themselves. It also appears that GSK3-alpha is reduced when beta is silenced in these cells? Is this the case - or just a loading discrepancy? If so what does this mean for this protein? It cannot bind GSK3 presumably?

Line 207 -clarify use of HeLa cells vs HCC1954 and MCF7 cells used in Figure 1. What is the reasoning here?

The data on the disruption of lysosome-ER membrane contacts with GSK3 modulation (lines 224-242), using as an assay the overexpression of STARD3 driving overexpressed VAPs into STARD3 positive puncta, is somewhat convincing. However, it could be argued that this is not measuring contact sites, it is only measuring STARD3-GSK3 co-localisation. Another type of assay, or more context on this assay to show this would strengthen this conclusion/ Could other markers be used, another ER marker and another lysosome marker to show that there is a shift of ER or ER-associated domains (analogous to the MAM) to lysosomes upon GSK3 inhibition and not just a VAP-STARD3 phenomenon? Or could TEM (as used later in the study) be employed here?

Figure 3 - Why not quantify the IP data in fig 3D?

The data that the expression of a non GSK3-phosphorylatable (S209) STARD3 mutant or a FFAT mutants also triggers the clustering of LE/Lyso's (most notably in cells expressing STARD3) is clear. Although, and this is just a comment, the use of the triple silencing VAPA/B/MOSPD2 - which presumably has massive impact on ER network and overall cell morphology I am not so convinced is that helpful or adds much to the story. However, the FIB-SEM and Super Res analyses are highly convincing.

The in vitro data clearly suggests liposome binding for the START domain dependent on 2 specific lysines, this is then supported by cellular work showing that when mutated these two residues block the clustering. However, the title of Figure 8 "The START domain interacts with membranes" should be clarified - this may be the case but this is not what this data in this figure directly shows.

Line 322 - "as well as transmission electron microscopy (Fig. 6A-C) on control HeLa cells and on cells expressing". This should be Supp Fig6?

Discussion - what is not discussed here are three things which are fairly key to the mechanism -

1) Why does the START domain not bind to LE/Lyso membranes when the FFAT motif is phosphorylated? What is the authors model here? That VAP binding to the FFAT motif somehow precludes the START domain from positioning itself though these are large separate domains? Does this make sense in terms of structure - could the authors attempt to model this in AlphaFold?

2) The authors mention in places that STARD3 levels may differ between different cell types, hence different use of cell types for certain experiments. Here, it would be useful to have some commentary on expected STARD3 levels and also likely different GSK3 levels/activity and different levels of S209 phosphorylation, in different cells and how these factors impact on the model. Is STARD3 always expressed? Why would some cells have much less - this would potentially mean less ER association? How would this fit with the role of such cells?

3) How could any of the regulation seen here fit with GSK3 signalling!?

I think, at the authors discretion, that these 3 points would be worth discussing.

Fabien ALPY

Tél. 03 88 65 35 19

Email Fabien.Alpy@igbmc.fr

Dr William Teale
The EMBO Journal
 Meyerhofstrasse 1
 69117 Heidelberg
 Germany

Illkirch, October 16th 2025

Dear Dr Teale,

Please find a revised version of our manuscript entitled “STARD3 regulates ER-endosome contacts and positioning via a GSK3-controlled phosphorylation switch” (EMBOJ-2025-121626).

We would like to thank the reviewers for their careful evaluation of the manuscript. The referees' concerns (boxed text) are discussed in detail below. The corresponding changes that we have made in the text are also indicated, as well as being printed in red in the revised manuscript. We feel that these changes improve the manuscript. Notably, we confirmed the role of GSK3 in the interaction and co-localization between STARD3 and VAP-A, VAP-B, and MOSPD2 in MCF7 cells (now presented in Figure 3) and clarified the use of different cell lines in the text.

Point by point answers to the reviewers

Referee #1:

Reviewer Comments

Eichler et al. have elucidated the molecular mechanism by which the intracellular localization of late endosomes and lysosomes is regulated through the activity of a tethering protein. Specifically, they demonstrate that phosphorylation of STARD3 at Ser209 by GSK3 α and GSK3 β activates STARD3-mediated contacts between the endoplasmic reticulum (ER) and late endosomes/lysosomes. This finding indicates that such phosphorylation events control endosomal localization within cells and sheds light on previously uncharacterized molecular mechanisms involving endosomal dynamics. Conventionally, the localization of these organelles was believed to be regulated primarily via motor proteins such as kinesin and dynein, and their interactions with microtubules. This study highlights an alternative mechanism, thus representing a significant advance in the field. Below, I provide several points that I believe require revision.

Major Points

1. Inconsistency in cell lines used across experiments:

The authors use various cell lines in different figures. For example, HCC1954 and MCF7 cells are used in Figure 1 (phosphorylation of STARD3 by GSK3 β), MCF7 in Figures 2, 5, 6, and 8, and HeLa cells in Figure 3 (immunoprecipitation and immunostaining experiments on the STARD3-VAP interaction) as well as in Figure S6 (electron microscopy). While the main figures predominantly use MCF7 cells, the rationale for employing HeLa cells specifically in the experiments shown in Figure 3 and Figure S6 is unclear. The authors should provide a justification for this choice.

For this study, we used representative human cell lines. Primarily, we worked with the breast cancer cell line HCC1954, where STARD3 is expressed at endogenous high levels and can be easily detected. A limitation of HCC1954 cells is their heterogeneous morphology, which makes imaging difficult. To overcome this issue, we also used another breast cancer cell line with a stable morphology called MCF7 that expresses low endogenous levels of STARD3. In MCF7 cells, we generated stable cell lines with ectopic STARD3 expression, allowing high-quality imaging, as shown in Figures 3, 4E–F, 5B–D, 6, 8, and EV2–4. The rationale for cell line choice is explained in the text (Line 138-143):

“STARD3 is a ubiquitous protein expressed at low basal levels, except in breast cancer cells with genomic alterations in the HER2 (Human Epidermal Growth Factor Receptor 2) locus that lead to high expression levels (Lodi et al., 2023; Tomasetto et al., 1995; Voilquin et al., 2019). To study the regulation of STARD3 phosphorylation, we selected two breast cancer cell lines, HCC1954 and MCF7, which naturally express high and low endogenous levels of STARD3, respectively. MCF7 cells served as a model for ectopic STARD3 expression.”

In addition, we replicated key experiments in HeLa cells, which express low endogenous levels of STARD3 because they are good models for studying endosomal traffic and membrane contact sites. These duplicated experiments are now presented in the supplementary figures, while the main figures show data from HCC1954 and MCF7 cells (details below). A strength of this study is that consistent results were obtained across different cell lines, supporting the notion that the functions described here are relevant to most cells.

One exception is the ultrastructural studies, which were all performed in HeLa cells. This choice was based on practical considerations: over the years, we have established in HeLa cells a robust high pressure freezing / freeze substitution protocol optimized for the detection of inter-organelle contacts, which is not easily transferable to other cell types.

To gain in consistency with the cells used, we repeated the experiments originally made in HeLa cells shown in Figure 3 (immunoprecipitation and immunostaining of the STARD3-VAP interaction) in MCF7 cells, which are now presented in the main Figure 3. The corresponding experiments performed in HeLa cells are now provided as a supplementary figure (Appendix Figure S2). These results confirm that the relationship between STARD3 and GSK3 is conserved across different cell types and can therefore be generalized. Line 224: *“Similar results were obtained in HeLa cells, showing that GSK3 activity is required across cell types (Fig. S2 A-C)”*.

2. Panel J in Figure 1:

In contrast to Panel I, which shows selective knockdown of GSK3 β in HCC1954 cells, Panel J shows a marked decrease in both GSK3 β and GSK3 α expression levels, even with GSK3 β knockdown alone, suggesting that knockdown of both isoforms may have occurred. However, the corresponding bar graph beneath this panel shows no substantial difference in STARD3 S209 phosphorylation levels between GSK3 β and GSK3 α knockdowns, while an additive effect is observed in the double knockdown. This apparent discrepancy requires clarification.

We thank the reviewer for this observation.

To determine whether silencing one GSK3 isoform affects the protein level of the other, we quantified GSK3 α and GSK3 β in HCC1954, and also in MCF7/STARD3 cells from three and four independent experiments, respectively. All raw blots from these experiments are provided as source data, and one representative blot is shown in Fig. 1E and 1F. Quantification of GSK3 α and GSK3 β protein levels is now shown in Fig. EV1D and EV1E. These data show that silencing one isoform does not alter the expression of the other, indicating that the blot initially presented in Fig. 1J was not representative of the overall data with respect to GSK3 isoform levels.

Minor Points

1. Schematic illustration in Figure 9:

The large oval-shaped structure enclosed within the cell appears to represent the nucleus, but this is not explicitly indicated in the figure. To avoid any potential confusion, the authors should clearly label or annotate this structure.

The nucleus is now labeled in the schematic illustration Figure 9.

2. Electron microscopy data in Figure S6:

As endosomes and lysosomes can be identified by electron microscopy, it may be possible to perform a more quantitative analysis (e.g., quantifying the number of membrane contact sites per endosome per cell). The authors are encouraged to consider whether such an analysis could be incorporated to support the qualitative observations.

In this work, we used two complementary electron microscopy approaches: transmission electron microscopy (TEM) and focused ion beam scanning electron microscopy (FIB-SEM). TEM allows precise imaging of thin sections, but it does not provide three-dimensional information. As a result, quantitative measures such as the number of contact sites *per* endosome per cell cannot be reliably obtained with this method. Previously, we performed quantitative yet time-consuming analyses on TEM sections by measuring the proportion of the endosomal perimeter in contact with the ER. We found that in control cells, about 5% of the total endosomal surface was covered by the ER. In contrast, in cells overexpressing

STARD3, more than 15% of the endosomal surface was covered (Alpy et al., 2013), demonstrating that STARD3 expression promotes ER–LE/Lys contacts.

The novelty of the present study lies in the characterization of the LE/Lys clustering phenotype. To document this, we provide FIB-SEM movies (Movie EV1–EV3) that illustrate the three-dimensional organization of endosomes, the ER, and mitochondria in control cells, in cells expressing STARD3 (showing increased ER–LE/Lys contacts), and in cells expressing a non-phosphorylatable STARD3 mutant (inducing LE/Lys clustering). FIB-SEM offers the advantage of providing three-dimensional information, but it remains a low-throughput approach. For this reason, we were limited in our ability to process enough samples to perform robust statistical analyses, and therefore, chose to present these data qualitatively. We would like to stress that however these FIB-SEM data are backed by statistically significant quantifications of LE/Lys clustering obtained by light microscopy.

Referee #2:

This paper from Eichler and colleagues dissects an elegant mechanism of regulation of ER-LE/Lysosome tethering, linked with LE/Lysosomal clustering. Using a combination of in vitro and cellular work in multiple different mammalian cell lines (including kinase inhibitors and silencing/mutational studies) the work identifies GSK3 as the kinase which phosphorylates the lysosomal protein STARD3 on a specific serine residue in its FFAT motif. This phosphorylation then regulates interaction with the ER tethering mediators VAPA/B/MOSPD2 to control connections between the ER and LE/Lysosomes. A similar regulatory mechanism has already been shown for mitochondria-ER and peroxisome-ER contacts and so this fits with an emerging theme of phosphorylation of FFAT motif regulation and places GSK3 beta at the centre of this. The authors then observed that uncoupling from VAP appears to allow the lipid binding START domain of STARD3 to then instead induce lysosomal clustering, potentially by a direct membrane binding mechanism. Based on this they propose an overall mechanism (Fig 9) that GSK3 activity either allows ER connection and the even distribution of LE/Lysos in the cell or loss of ER connection, resulting in clustering at the perinuclear region.

The study is exceptionally well performed, with multiple control experiments, different cell lines utilised, complementary in vitro and cellular work and mostly very clear and convincing data.

A weakness would be, as no clear function is attributed to this clustering/evenly distributed switch, the purpose of this mechanism and where it is applied is unclear. It could be considered that this reduces the importance of the study for a wider readership. However, from a cell-biology perspective, this is an important study which contributes to our understanding of a current hot topic in the field, regulation of membrane contact sites.

Overall, the vast majority of the data is highly convincing, and despite the lack of a clear function for the switching mechanism this study clearly warrants publication.

I have just a few minor suggestions for improvement:

Line 138-139 - specify which cells were used

We thank the reviewer for their positive feedback on our manuscript.

The model cell line used in Fig. 1C-D is now indicated in the text. In addition, we have provided the rationale for the use of different cell lines throughout the manuscript (See answer to Reviewer 1 Point 1).

Figure 1J - MCF7 GSK3 silencing - I find this a little unconvincing, whilst the data for the HCC1954 cells is clear - do the authors have a blot which is not over-exposed for GAPDH and STARD3 so the reader can judge the change for themselves. It also appears that GSK3-alpha is reduced when beta is silenced in these cells? Is this the case - or just a loading discrepancy? If so what does this mean for this protein? It cannot bind GSK3 presumably?.

This point has already been addressed in our response to Reviewer 1, point 2. In brief, the experiments were reproduced, and the new data are now presented in Fig. 1J, along with a quantification in Fig. EV1D and EV1E. These results confirm that GSK3- α protein levels are not affected by GSK3- β silencing.

Line 207 -clarify use of HeLa cells vs HCC1954 and MCF7 cells used in Figure 1. What is the reasoning here?.

This point concurs with reviewer 1's first concern, which we have addressed above in our response to Reviewer 1, point 1.

The data on the disruption of lysosome-ER membrane contacts with GSK3 modulation (lines 224-242), using as an assay the overexpression of STARD3 driving overexpressed VAPs into STARD3 positive puncta, is somewhat convincing. However, it could be argued that this is not measuring contact sites, it is only measuring STARD3-GSK3 co-localisation. Another type of assay, or more context on this assay to show this would strengthen this conclusion/ Could other markers be used, another ER marker and another lysosome marker to show that there is a shift of ER or ER-associated domains (analogous to the MAM) to lysosomes upon GSK3 inhibition and not just a VAP-STARD3 phenomenon? Or could TEM (as used later in the study) be employed here?

To answer this question, we employed an alternative approach to visualize the proximity of the ER and LE/Lys using a fluorescent marker of the ER (mScarlet-ER) and LE/Lys (EGFP-TMEM182), as suggested by this reviewer. These new data are now shown in Figure EV2. Expression of STARD3 increased ER–LE/Lys co-localization compared with controls, as indicated by a higher Pearson correlation coefficient between the two markers, demonstrating that STARD3-dependent contacts can be visualized with this approach. Upon GSK3 inhibition, the correlation returned to basal levels in STARD3-negative cells, showing that ER–LE/Lys proximity depends on GSK3 activity. These results confirm that GSK3 is required for STARD3-mediated ER–LE/Lys tethering.

Figure 3 - Why not quantify the IP data in fig 3D?

The IP data in Fig. 3D are now quantified and shown as panel 3E.

The data that the expression of a non GSK3-phosphorylatable (S209) STARD3 mutant or a FFAT mutants also triggers the clustering of LE/Lys's (most notably in cells expressing STARD3) is clear. Although, and this is just a comment, the use of the triple silencing VAPA/B/MOSPD2 - which presumably has massive impact on ER network and overall cell morphology I am not so convinced is that helpful or adds much to the story. However, the FIB-SEM and Super Res analyses are highly convincing.

The LE/Lys clustering phenotype induced by the expression of STARD3, whose Phospho-FFAT was mutated and unable to interact with VAPA/VAPB/MOSPD2, suggests that the lack of this interaction is responsible for the phenotype. In other words, when STARD3 is unable to associate with any of the three VAP proteins, it induces LE/Lys clustering. To further test this possibility, we put up an alternative experimental strategy reasoning that removing all VAPs would disrupt the interaction between STARD3 and the ER. Consistently, the overexpression of STARD3 in cells lacking VAPA/VAPB/MOSPD2 leads to the formation of LE/Lys clusters. We acknowledge, as rightly pointed out by the reviewer, that the depletion of all three VAPs does have an impact on cells, particularly with respect to their viability. Importantly, however, depletion of VAPA/VAPB/MOSPD2 alone does not trigger the clustering phenotype; this effect specifically arises in the presence of STARD3, underscoring that the phenotype is not a general consequence of VAP depletion but a STARD3-dependent process.

The in vitro data clearly suggests liposome binding for the START domain dependent on 2 specific lysines, this is then supported by cellular work showing that when mutated these two residues block the clustering. However, the title of Figure 8 "The START domain interacts with membranes" should be clarified - this may be the case but this is not what this data in this figure directly shows.

We appreciate the reviewer's comment, as the original title did not fully reflect the data presented. We have revised the title to: "The ability of the START domain to bind membranes is required for LE/Lys clustering induced by STARD3".

Line 322 - "as well as transmission electron microscopy (Fig. 6A-C) on control HeLa cells and on cells expressing". This should be Supp Fig6?

Thank you for pointing this out. It has been corrected to Appendix Fig. S5 (the numbering of supplementary figures has been updated in the revised version).

Discussion - what is not discussed here are three things which are fairly key to the mechanism -

1) Why does the START domain not bind to LE/Lyso membranes when the FFAT motif is phosphorylated? What is the authors model here? That VAP binding to the FFAT motif somehow precludes the START domain from positioning itself though these are large separate domains? Does this make sense in terms of structure - could the authors attempt to model this in AlphaFold?

2) The authors mention in places that STARD3 levels may differ between different cell types, hence different use of cell types for certain experiments. Here, it would be useful to have some commentary on expected STARD3 levels and also likely different GSK3 levels/activity and different levels of S209 phosphorylation, in different cells and how these factors impact on the model. Is STARD3 always expressed? Why would some cells have much less - this would potentially mean less ER association? How would this fit with the role of such cells?

3) How could any of the regulation seen here fit with GSK3 signalling!?

I think, at the authors discretion, that these 3 points would be worth discussing.

The different points raised by the reviewer are now discussed in the Discussion section:

Point 1 (page 15)

The inability of phosphorylated STARD3 to generate homotypic interactions likely reflects differences in binding strength, with its stronger interaction with VAPs prevailing over its weaker interaction with membranes. A comparable mechanism likely occurs for the mitochondria-bound protein MIGA2, which forms mitochondria-ER contacts via VAP binding when its phospho-FFAT is active, but, when not phosphorylated, engages mitochondria-LD contacts by directly binding to the LD surface (Freyre et al., 2019).

Point 2 and 3 are discussed in the same paragraph (page 16):

This balance is likely influenced by STARD3 expression levels and GSK3 activity. STARD3 is expressed ubiquitously at basal levels and is overexpressed in HER2-positive breast cancers (Lodi et al., 2023; Tomasetto et al., 1995). GSK3 activity is subject to complex regulation such as through inhibitory phosphorylation mediated by multiple kinases, and through its subcellular localization (Beurel et al., 2015). An important remaining question is to determine the relative abundance of phosphorylated versus unphosphorylated STARD3 in different cell types and in cancer, and how this distribution shapes the balance between heterotypic and homotypic interactions. Notably, GSK3 activity is linked to lysosome biology: GSK3 regulates mammalian target of rapamycin (mTOR) activity, a central kinase that integrates nutrient availability with cell growth. mTOR is localized to lysosomes when inactive, and its activation is associated with changes in lysosome positioning (Jia and Bonifacino, 2019; Korolchuk et al., 2011). The connection between these pathways and STARD3 function deserves further investigations.

I thank you for your attention.

Sincerely yours

Dear Fabien,

We have now received re-review reports both referees, which I have included below. As you will see, you have addressed their concerns satisfactorily. Before I can finally accept the manuscript, there are some remaining editorial points which need to be addressed. In this regard would you please:

- note that employment in a biotech company should be stated in the "Disclosure and competing interests statement",
- acknowledge funding from EUR IMCBio (ANR-17-EURE-0023) (funding information included in the Comments box could not be extracted by our production team, all funders should therefore be added to our "More Funders" list),
- include up to five keywords,
- change the title of the conflict of interests statement to "Disclosure and competing interests statement",
- remove the AC/CrediT section from the text,
- rephrase the sentence in the discussion on motif analysis (on page 15) to allow you to remove '(not shown)' from the text,
- be reminded that there is the opportunity to attach any lab protocols to the manuscript that you think might be helpful to the community,
- correct the callout to Table 1; this table has not been included,
- ensure all callouts are listed sequentially,
- update source file name, title, legend and manuscript callout to Dataset EV1 instead of EV Table 1; the legend should be removed from the main manuscript and uploaded as a separate tab/sheet in the Excel file,
- include "Appendix for STARD3 regulates ER-endosome contacts and positioning via a GSK3-controlled phosphorylation switch" on the Appendix title page,
- remove Appendix figure legends from the main manuscript file,
- remove the Reagents and Tools table from the main manuscript file and upload it as a separate file using the template from our guide to authors,
- supply Source Data for Figures 2C, 3E, 5E-G,
- provide p values in the legends of figures 1B, D, E, F, J; 2C, F; 3E, K; 4D, G; 5C, 6F, K, O; 7E, G; 8F, EV1 C, D, E; EV2 E, EV4 J,
- remove movie legends from the main manuscript and zip with each movie file, and
- correct the section order as follows: Title page - Abstract - Keywords - Introduction - Results - Discussion - Methods - Data Availability - Acknowledgements - Disclosure and Competing Interests Statement - References - Figure Legends - Table(s) - Expanded View Figure Legends.

I am looking forward to receiving your revised manuscript.

EMBO Press is an editorially independent publishing platform for the development of EMBO scientific publications.

Best wishes,

William

William Teale, PhD
Editor
The EMBO Journal
w.teale@embojournal.org

Read our guidance for manuscript revisions and related editorial policies: <https://link.springer.com/journal/44318/submission-guidelines#cms-Revised-submissions>

<https://media.springernature.com/original/springer-cms/rest/v1/content/27825798/data/v1>

- a point-by-point response to the referees' comments, with a detailed description of the changes made (as a word file).
- a word file of the manuscript text.

- individual production quality figure files (one file per figure)
- a complete author checklist
- Expanded View files (replacing Supplementary Information)
- a Reagents and Tools Table as part of the Methods section

Please remember: Digital image enhancement is acceptable practice, as long as it accurately represents the original data and conforms to community standards. If a figure has been subjected to significant electronic manipulation, this must be noted in the figure legend or in the 'Methods' section. The editors reserve the right to request original versions of figures and the original images that were used to assemble the figure.

We realize that it is difficult to revise to a specific deadline. In the interest of protecting the conceptual advance provided by the work, we recommend a revision within 3 months (4th Mar 2026). Please discuss the revision progress ahead of this time with the editor if you require more time to complete the revisions.

Referee #1:

The authors have addressed the comments with sincerity, and the additional experiments have been conducted appropriately. They have also adequately responded to the major concern regarding the differences in cell types. Therefore, I have no further comments, and I consider the manuscript to be acceptable.

Referee #2:

All my comments have been addressed and I suggest publication.

Fabien ALPY

Tél. 03 88 65 35 19

Email Fabien.Alpy@igbmc.fr

Dr William Teale
The EMBO Journal
Meyerhofstrasse 1
69117 Heidelberg
Germany

Illkirch, December 13th 2025

Dear Dr Teale,

Please find the revised version of our manuscript entitled "STARD3 regulates ER-endosome contacts and positioning via a GSK3-controlled phosphorylation switch" (EMBOJ-2025-121626).

The editorial changes were performed as indicated.

Point by point answer

- note that employment in a biotech company should be stated in the "Disclosure and competing interests statement",

No co-author was employed by a company during this project.

- acknowledge funding from EUR IMCBio (ANR-17-EURE-0023) (funding information included in the Comments box could not be extracted by our production team, all funders should therefore be added to our "More Funders" list),

The list of funders was updated on the website.

- include up to five keywords,

Done. *Membrane contact site. Endoplasmic reticulum. Endosome. Lipid transfer protein. Phosphorylation*

- change the title of the conflict of interests statement to "Disclosure and competing interests statement",

Done.

- remove the AC/CrediT section from the text,

Done.

- rephrase the sentence in the discussion on motif analysis (on page 15) to allow you to remove '(not shown)' from the text,

The sentence was rephrased.

- be reminded that there is the opportunity to attach any lab protocols to the manuscript that you think might be helpful to the community,

Ok.

- correct the callout to Table 1; this table has not been included,

Table 1 is now Dataset EV1 and is called page 15.

- ensure all callouts are listed sequentially,

Done.

- update source file name, title, legend and manuscript callout to Dataset EV1 instead of EV Table 1; the legend should be removed from the main manuscript and uploaded as a separate tab/sheet in the Excel file,

Done.

- include "Appendix for STARD3 regulates ER-endosome contacts and positioning via a GSK3-controlled phosphorylation switch" on the Appendix title page,

Done.

- remove Appendix figure legends from the main manuscript file,

Done.

- remove the Reagents and Tools table from the main manuscript file and upload it as a separate file using the template from our guide to authors,

Done.

- supply Source Data for Figures 2C, 3E, 5E-G,

Source Data for Figures 2C and 3E have been uploaded on the submission website.

Source Data for Figures 5E-G were deposited on EMPIAR. DOI: 10.6019/EMPIAR-13156 and will be publicly available in the next few days.

- provide p values in the legends of figures 1B, D, E, F, J; 2C, F; 3E, K; 4D, G; 5C, 6F, K, O; 7E, G; 8F, EV1 C, D, E; EV2 E, EV4 J,

Done.

- remove movie legends from the main manuscript and zip with each movie file, and

Done.

- correct the section order as follows: Title page - Abstract - Keywords - Introduction - Results - Discussion - Methods - Data Availability - Acknowledgements - Disclosure and Competing Interests Statement - References - Figure Legends - Table(s) – Expanded

Done.

I thank you for your attention.

Sincerely yours

Dear Fabien,

I am pleased to inform you that your manuscript has been accepted for publication in the EMBO Journal.

Congratulations to you and your team!

You may qualify for financial assistance for your publication charges - either via a Springer Nature fully open access agreement or an EMBO initiative. Check your eligibility: <https://link.springer.com/journal/44318/how-to-publish-with-us>

Yours sincerely,

William

William Teale, PhD
Editor
The EMBO Journal
w.teale@embojournal.org

Please note that it is The EMBO Journal policy for the transcript of the editorial process (containing referee reports and your response letters) to be published as an online supplement to each paper. If you should prefer removal of any referee-only figures included in the point-by-point response(s), e.g. because they may still be used for future publication or because they have been reproduced from published work by others, please do let us know immediately via response email.

More information is available here: <https://link.springer.com/partners/embo-press/editorial-policies#Peer%20review>